# Sketched Gaussian Mechanism for Private Federated Learning

**Qiaobo Li**
Siebel School of Computing and Data Science
University of Illinois Urbana-Champaign
`qiaobol2@illinois.edu`

**Zhijie Chen**
Siebel School of Computing and Data Science
University of Illinois Urbana-Champaign
`lucmon@illinois.edu`

**Arindam Banerjee**
Siebel School of Computing and Data Science
University of Illinois Urbana-Champaign
`arindamb@illinois.edu`

## Abstract

Communication cost and privacy are two major considerations in federated learning (FL). For communication cost, gradient compression by sketching the clients' transmitted model updates is often used for reducing per-round communication. For privacy, the Gaussian mechanism (GM), which consists of clipping updates and adding Gaussian noise, is commonly used to guarantee client-level differential privacy. Existing literature on private FL analyzes privacy of sketching and GM in an isolated manner, illustrating that sketching provides privacy determined by the sketching dimension and that GM has to supply any additional desired privacy.

In this paper, we introduce the Sketched Gaussian Mechanism (SGM), which directly combines sketching and the Gaussian mechanism for privacy. Using Rényi-DP tools, we present a joint analysis of SGM's overall privacy guarantee, which is significantly more flexible and sharper compared to isolated analysis of sketching and GM privacy. In particular, we prove that the privacy level of SGM for a fixed noise magnitude is proportional to $1/\sqrt{b}$, where $b$ is the sketching dimension, indicating that (for moderate $b$) SGM can provide much stronger privacy guarantees than the original GM under the same noise budget. We demonstrate the application of SGM to FL with either gradient descent or adaptive server optimizers, and establish theoretical results on optimization convergence, which exhibits only a logarithmic dependence on the number of parameters $d$. Experimental results confirm that at the same privacy level, SGM based FL is at least competitive with non-sketching private FL variants and outperforms them in some settings. Moreover, using adaptive optimization at the server improves empirical performance while maintaining the privacy guarantees.

## 1 Introduction

Federated learning (FL) [44] is a widely used machine learning framework where a shared global model is trained under the coordination of a central server using data distributed across various clients. In FL, each client carries out training steps on its local data and sends only the model updates back to the server, which then refines the global model by suitably aggregating these local updates. Because the data remains on the clients, FL can offer enhanced privacy protection compared to traditional centralized learning. Nonetheless, FL faces two significant challenges: (1) it lacks a rigorous privacy guarantee (e.g., differential privacy (DP)) [17] and has been shown to be susceptible to various

39th Conference on Neural Information Processing Systems (NeurIPS 2025).

inference attacks, leading to local information leakage during training [49, 50, 75, 83, 84]; and (2) it usually requires a high communication overhead due to the frequent communication between the server and the client [35]. Many recent advances in FL have been motivated by these two challenges.

Towards reducing communication costs, a key goal has been to reduce the communication cost per round by compressing local updates from clients to a lower dimension [30, 36], including sparsification [40, 70, 5], quantization [4, 41, 52], and sketching [59, 54, 28]. Among these approaches, sketching methods stand out for its simplicity, making it easier to integrate with existing FL methods. As an unbiased compressor, it does not require bias correction using error feedback mechanisms, unlike sparsification, which incurs additional memory costs [57, 60]. As a linear operation, sketching ensures the geometric properties are approximately preserved after compression [13] as opposed to quantization distorting the inner product structure, potentially slowing convergence [4].

Towards preserving privacy, two DP definitions are commonly considered in FL algorithm design: sample-level privacy and client-level privacy, where client-level privacy is the stricter guarantee in the sense that it ensures the output remains statistically indistinguishable when an entire client's dataset is altered, rather than merely hiding the inclusion or exclusion of individual data samples as sample-level privacy [81]. Various FL algorithms [21, 71, 63, 66] have focused on client-level privacy by adapting standard differential privacy techniques from centralized training (i.e., clipping gradients and adding Gaussian noise to the clipped values [1, 19]) to the federated learning setting.

Though privacy and communication-efficiency have mostly been studied independently, there have been some efforts towards solving these two challenges together. $D^2P$-FED[71] combines stochastic quantization and random rotation [44] techniques to lower communication costs, to the discrete Gaussian mechanism for privacy [9], along with secure aggregation [7] to mitigate noise magnitude. Alternatively, DPSFL [80] directly applies the standard Gaussian mechanism to the communication-efficient federated learning algorithm FetchSGD [54]. However, these methods merely select individual compression and privacy components and stitch them together, treating communication-efficiency and differential privacy as separate concerns rather than examining their intrinsic interplay.

[37] is one of the very few works to investigate the underlying relationship between differential privacy and communication efficiency in distributed learning. It demonstrates that the Count Sketch algorithm [10], despite being originally developed for communication efficiency, inherently satisfies a form of differential privacy for distributed learning algorithms. However, their results have several major limitations. First, when the privacy level $\epsilon$ provided by the Count Sketch mechanism falls short of the requirement, additional Laplacian or Gaussian noise must be injected, but there is no precise characterization of the amount of noise required. Second, their theoretical guarantees require arguably impractical assumptions, including the input (client gradients) to follow a Gaussian distribution and the sketching dimension $b \lesssim \sqrt{d}$ where $d$ is the input (gradient) dimension, which contradicts typical empirical configurations in sketched distributed learning, where $b$ is often a fixed fraction of $d$, e.g., $b = d/100$, to avoid derailing the optimization.

In this paper, we introduce the Sketched Gaussian Mechanism (SGM), which combines an isometric Gaussian sketching transform [59] with the classical Gaussian mechanism. Leveraging tools from Rényi differential privacy (RDP) [46] with the subsampling, post-processing, and composition theorems of differential privacy [18], we derive a tight upper bound on the overall privacy level $\epsilon$ of SGM. Concretely, with all other hyperparameters held fixed, $\epsilon = O(\frac{1}{\sqrt{b}\sigma_g^2})$, where $b$ denotes the sketch dimension and $\sigma_g^2$ the variance of the added Gaussian noise. Our new result for SGM **establishes a clear dependence of** $\epsilon$ **on both** $b$ **and** $\sigma_g^2$. Unlike prior work, our result imposes no restrictive upper limit on $b$, thereby covering practical regimes in which $b$ is a fixed fraction of the ambient dimension $d$. More importantly, this bound implies that for suitably large $b$, SGM achieves strictly stronger privacy guarantees than the standard Gaussian mechanism, demonstrating that the sketching operation itself confers inherent privacy benefits.

We further integrate SGM into a federated learning (FL) framework, referred to as Fed-SGM, supporting flexible choices of server optimizers to match practical deployment needs. We prove that this Fed-SGM satisfies client-level privacy guarantees. Moreover, by fully leveraging the fast-decaying spectrum of the deep-learning loss Hessian [77, 82, 76], which implies a small absolute intrinsic dimension [26], we establish rigorous optimization convergence bounds that scale only logarithmically in the ambient dimension $d$ and linearly in this absolute intrinsic dimension, ensuring scalability to high-dimensional problems.

We empirically validate our approach on deep learning models for both vision and language benchmarks. Across these tasks, federated SGM requires strictly less Gaussian noises than the standard DP-FedAvg algorithm [81] to achieve the same privacy guarantee, and consistently delivers comparable or even superior model accuracy. Furthermore, when we replace vanilla gradient descent on the server with an adaptive optimizer, we observe additional accuracy gains, highlighting the practical benefit of incorporating adaptive optimization into our privacy-preserving federated learning framework.

The remainder of the paper is organized as follows. Section 2 provides the description of our SGM and establishes the privacy guarantee. Section 3 introduces Fed-SGM, the application of SGM into federated learning setting, and establish optimization analysis of our framework. In Section 4, we present experimental evaluations that assess performances of our Fed-SGM and compare them with existing approaches. Finally, Section 5 concludes our contributions and highlight potential future directions. Due to space limit, we present our discussion on related works in Appendix A.

## 2 Privacy Guarantee of SGM

In this section, we will introduce the Sketched Gaussian Mechanism (SGM), and provide the privacy guarantee with two different kinds of analysis.

**Definition 2.1** (Sketched Gaussian Mechanism (SGM)). For any statistic $\theta(D) \in \mathbb{R}^d$ of the dataset $D$, the Sketched Gaussian Mechanism outputs $\mathcal{SG}(\theta; R, \xi) = R\theta + \xi$, in which $R \in \mathbb{R}^{b \times d}$ is a Gaussian sketching matrix with each entry sampled i.i.d. from $\mathcal{N}(0, \frac{1}{\sqrt{b}})$ and $\xi \in \mathbb{R}^b$ follows the Gaussian distribution $\mathcal{N}\left(0, \sigma_g^2 \mathbb{I}_b\right)$.

---

**Algorithm 1** Sketched Gaussian Mechanism

---

**Hyperparameters:** learning rate $\eta_t$, noise scale $\sigma_g$, clipping threshold $\tau$, number of iterations $T$.
**Inputs:** Examples $D = \{x_1, \ldots, x_n\}$, loss function $\mathcal{L}(\theta) = \frac{1}{n} \sum_{i=1}^{n} \mathcal{L}(\theta, x_i)$.
  **Initialize** $\theta_0$ randomly.
  **for** $t = 0, \ldots, T-1$ **do**
    Take a mini-batch $B_t$ of $m$ samples with sampling probability $q = \frac{m}{n}$
    **Computer gradient:** For each $i \in B_t$, $g_t(x_i) \leftarrow \nabla \mathcal{L}(\theta_t, x_i)$
    **Clip the gradient:** $\hat{g}_t(x_i) = \text{clip}(g_t(x_i), \tau) = g_t(x_i) \cdot \min\left\{1, \frac{\tau}{\|g_t(x_i)\|_2}\right\}$
    **Apply SGM:** For each $i \in B_t$, $\tilde{g}_t(x_i) = \mathcal{SG}(\hat{g}_t(x_i); R_t, \xi_{t,i})$
    **Aggregate:** $\bar{\tilde{g}}_t = \frac{1}{m} \sum_{i \in B_t} \tilde{g}_t(x_i)$
    **Update parameter:** $\theta_{t+1} \leftarrow \text{OPT}\left(\theta_t, \bar{\tilde{g}}_t, \eta_t\right)$
  **end for**
**Outputs:** final model $\theta_T$

---

Algorithm 1 outlines a standard application of SGM in training a model with parameter $\theta$ in a similar manner to the standard DP-SGD [1]. At each step of training, we compute the gradient for a random subset of examples, apply clipping and SGM to each gradient, and update the parameter with the optimizer OPT using the aggregated gradient.

Denote $\gamma_t(D) = \sum_{i \in B_t} \hat{g}_t(x_i)$, then $\|\gamma_t(D)\|_2 \leq m\tau$. Notice that sketching with the random matrix $R_t$ is a linear operation, so we can rewrite the aggregated gradient $\bar{\tilde{g}}_t$ as:

$$\bar{\tilde{g}}_t = \frac{1}{m} \sum_{i \in B_t} \tilde{g}_t(x_i) = \frac{1}{m} \sum_{i \in B_t} \mathcal{SG}(\hat{g}_t(x_i); R_t, \xi_{t,i}) = \frac{1}{m} \sum_{i \in B_t} (R_t \hat{g}_t(x_i) + \xi_{t,i})$$

$$= \frac{1}{m} \left(R_t \left(\sum_{i \in B_t} \hat{g}_t(x_i)\right) + \sum_{i \in B_t} \xi_{t,i}\right) = \frac{1}{m} \left(R_t \gamma_t(D) + \sum_{i \in B_t} \xi_{t,i}\right) = \frac{\mathcal{SG}(\gamma_t(D); R_t, \xi_t)}{m}$$

where $\xi_t = \sum_{i \in B_t} \xi_{t,i}$ is a Gaussian vector with covariance matrix $m\sigma_g^2 \mathbb{I}_b$. Therefore, the aggregated gradient $\bar{\tilde{g}}_t$ is also an output of SGM based on the examples.

We study the privacy guarantee of Algorithm 1 subject to the rigorous privacy guarantees of Differential Privacy (DP)[17], whose formal definition is given below.

**Definition 2.2.** A randomized algorithm $\mathcal{M}$ is $(\epsilon, \delta)$-differentially private if for any pair of datasets $D, D'$ differ in exactly one data point and for all event $\mathcal{Y}$ in the output range of $\mathcal{M}$, we have

$$\mathbb{P}\{\mathcal{M}(D) \in \mathcal{Y}\} \leq e^\epsilon \mathbb{P}\{\mathcal{M}(D') \in \mathcal{Y}\} + \delta$$

where the probability is taken over the randomness of $\mathcal{M}$.

## 2.1 Warm Up: Privacy Analysis by Moments Accountant

Compared to the standard Gaussian mechanism [18], SGM incorporates an additional sketching operation. By the Johnson–Lindenstrauss lemma [13], sketching can be interpreted as a distance-preserving embedding that approximates the original high-dimensional update in a lower-dimensional subspace. Under this viewpoint of sketching, we obtain the privacy guarantee for Algorithm 1 by an extension of the moments accountant analysis of Gaussian mechanism [1]. We state the main theorem below with the full proof in Appendix C.1.

**Theorem 2.1.** *There exists constants $c_1$ and $c_2$ so that given the sampling probability $q = \frac{m}{n}$ and the number of steps $T$, for any $\epsilon_p < c_1 q^2 T$, Algorithm 1 is $(\epsilon_p, \delta_p)$-differentially private for any $\delta_p > 0$ if we choose*

$$\sigma_g \geq c_2 \frac{\tau \sqrt{\left(1 + \frac{\log^{1.5}(2mT/\delta_p)}{\sqrt{b}}\right) mT \log(2/\delta_p)}}{n\epsilon_p} . \tag{1}$$

**Remark 2.1.** This privacy bound decreases monotonically in the sketching dimension $b$. However, since the dependence is of the form $(1 + O(1/\sqrt{b}))$, even for large $b$, the requisite Gaussian noise variance $\sigma_g^2$ remains asymptotically at par with that of standard DP-SGD [1]. Here, the analysis attributes the entire privacy solely to the Gaussian mechanism, treating sketching purely as an approximation and omitting any potential privacy that the sketching step might confer. $\square$

## 2.2 Main Result: Privacy Analysis by Rényi Differential Privacy

To more precisely characterize the privacy guarantees of the Sketched Gaussian Mechanism and to investigate any privacy contributions imparted by the sketching step, we develop the following novel privacy guarantee of Algorithm 1.

**Theorem 2.2.** *There exists constants $c_3$ and $c_4$ so that given the sampling probability $q = \frac{m}{n}$ and the number of steps $T$, for any $\epsilon_p \leq c_3 q \sqrt{T}$, Algorithm 1 is $(\epsilon_p, \delta_p)$-differentially private for any $\delta_p > 0$ if we choose*

$$\sigma_g^2 \geq \frac{c_4 q \tau^2 \sqrt{T} \log(2qT/\delta_p)}{\sqrt{b}\epsilon_p} . \tag{2}$$

**Remark 2.2.** In Theorem 2.2, for any fixed privacy target $\epsilon_p$, the required Gaussian noise variance $\sigma_g^2$ is a monotonically decreasing function of the sketch dimension $b$, so sketching to moderately high dimensions $b$ provably reduces the marginal variance of the Gaussian mechanism. Moreover, compared to Theorem 2.1, one attains the same privacy guarantee with strictly less noise once the sketch dimension satisfies $b \geq \Omega\left(\frac{(n\epsilon_p \log(2pT/\delta_p))^2}{T \log^2(1/\delta_p)}\right)$. $\square$

We refer readers to Appendix C.2 for a detailed proof, and provide a high-level sketch here. Our analysis leverages tools from the Rényi Differential Privacy (RDP) framework [46].

**Definition 2.3** (Rényi divergence [53])**.** For two probability distributions $P$ and $Q$ defined over $\mathcal{R}$, the Rényi divergence of order $\alpha > 1$ is $D_\alpha(P\|Q) \triangleq \frac{1}{\alpha-1} \log \mathbb{E}_{x \sim Q}\left(\frac{P(x)}{Q(x)}\right)^\alpha$.

From the definition of SGM, $\mathcal{SG}(\gamma_t(D); R_t, \xi_t) \sim \mathcal{N}\left(0, \left(\frac{\|\gamma_t(D)\|^2}{b} + m\sigma_g^2\right) \mathbb{I}_b\right)$, which can be used to show the following result:

**Lemma 2.1.** *The Rényi divergence between $\mathcal{SG}(\gamma_t; R_t, \xi_t)$ for neighboring datasets $D, D'$ is*

$$D_\alpha\left(\mathcal{SG}(\gamma_t(D); R_t, \xi_t)\|\mathcal{SG}(\gamma_t(D'); R_t, \xi_t)\right) = bf_\alpha\left(\sqrt{\frac{\|\gamma_t(D')\|^2 + mb\sigma_g^2}{\|\gamma_t(D)\|^2 + mb\sigma_g^2}}\right) ,$$

*where $f_\alpha(x) = \log x + \frac{1}{2(\alpha-1)} \log \frac{x^2}{\alpha x^2 + 1 - \alpha}$.*

Since this divergence purely depends on the ratio $\sqrt{\frac{\|\gamma_t(D')\|^2 + mb\sigma_g^2}{\|\gamma_t(D)\|^2 + mb\sigma_g^2}}$, we define the ratio sensitivity of the statistic analogous to the classical sensitivity measure in the Gaussian mechanism [18].

**Definition 2.4** (Ratio Sensitivity). For any constant $c \geq 0$, define the ratio sensitivity of $\theta$ as

$$\text{rsens}_c(\theta) = \sup_{D, D'} \sqrt{\frac{\|\theta(D')\|^2 + c^2}{\|\theta(D)\|^2 + c^2}} \, , \tag{3}$$

where the supremum is over all neighboring datasets $D, D'$.

From the definition, a direct analysis shows

$$\sqrt{1 - \frac{2\tau^2}{b\sigma_g^2}} \leq \frac{1}{\text{rsens}_{\sqrt{mb}\sigma_g}(\gamma_t)} \leq 1 \leq \text{rsens}_{\sqrt{mb}\sigma_g}(\gamma_t) \leq \sqrt{1 + \frac{2\tau^2}{b\sigma_g^2}} \, . \tag{4}$$

Since $f_\alpha(x)$ is monotonically decreasing for $x \leq 1$ and increasing for $x \geq 1$, we can obtain the following bound on the Rényi divergence.

**Lemma 2.2.** *For any neighboring datasets $D, D'$,*

$$D_\alpha\left(\mathcal{SG}(\gamma_t(D)); R_t, \xi_t) \| \mathcal{SG}(\gamma_t(D')); R_t, \xi_t)\right) \leq b \max\left\{ f_\alpha\left(\sqrt{1 + \frac{2\tau^2}{b\sigma_g^2}}\right), f_\alpha\left(\sqrt{1 - \frac{2\tau^2}{b\sigma_g^2}}\right) \right\} \leq \frac{\alpha^2 \tau^4}{(\alpha - 1)b\sigma_g^4}$$

We are now ready to analyze the privacy of SGM using Rényi Differential Privacy (RDP).

**Definition 2.5** ($(\alpha, \epsilon)$-RDP [46]). A randomized mechanism $f : \mathcal{D} \to \mathcal{R}$ is said to have $\epsilon$-Rényi differential privacy of order $\alpha$, or $(\alpha, \epsilon)$-RDP for short, if for any adjacent $D, D' \in \mathcal{D}$, it holds that $D_\alpha(f(D)\|f(D')) \leq \epsilon$.

Recall that RDP can be transformed into the standard $(\epsilon, \delta)$-DP.

**Lemma 2.3** (Relationship with $(\epsilon, \delta)$-DP [46]). *If $f$ is an $(\alpha, \epsilon)$-RDP mechanism, it also satisfies $\left(\epsilon + \frac{\log 1/\delta}{\alpha - 1}, \delta\right)$-differential privacy for any $0 < \delta < 1$.*

Based on Lemma 2.2 and Lemma 2.3, we immediately have the RDP and DP result for SGM.

**Lemma 2.4.** *SGM on $\gamma_t$ is $(\alpha, \frac{\alpha^2 \tau^4}{(\alpha-1)b\sigma_g^4})$-RDP, therefore $(\frac{\alpha^2 \tau^4}{(\alpha-1)b\sigma_g^4} + \frac{\log(1/\delta)}{\alpha-1}, \delta)$-DP.*

Optimizing over the Rényi order $\alpha$ (in Lemma 2.4) recovers the optimal $(\epsilon, \delta)$-DP guarantee for SGM at each step for the entire dataset $D$. Subsequently, by invoking the subsampling lemma, the post-processing invariance, and the sequential composition theorem from [18], we can establish the complete privacy guarantee of Algorithm 1.

**Remark 2.3.** Theorem 2.2 shows that, holding all other parameters fixed, the privacy level $\epsilon_p$ is monotonically decreasing with the sketching dimension $b$. This can appear counterintuitive at first, so we outline the intuition. According to the previous calculation, $\mathcal{SG}(\gamma_t(D); R_t, \xi_t) \sim \mathcal{N}\left(0, \left(\frac{\|\gamma_t(D)\|^2}{b} + m\sigma_g^2\right)\mathbb{I}_b\right)$. As $b$ grows, the signal term $\frac{\|\gamma_t(D)\|^2}{b}$ vanishes and the noise term $m\sigma_g^2$ dominates, making the output distribution increasingly independent of $D$. Consequently, for any two neighboring datasets $D$ and $D'$, the distributions $\mathcal{SG}(\gamma_t(D); R_t, \xi_t)$ and $\mathcal{SG}(\gamma_t(D'); R_t, \xi_t)$ become increasingly similar, since both are essentially noise–driven. Lemma 2.2 formalizes this: for a fixed noise multiplier $\sigma_g/\tau$, the $\alpha$-Rényi divergence between these two distributions is bounded by $\frac{\alpha^2 \tau^4}{(\alpha-1)b\sigma_g^4}$, so that larger $b$ strictly decreases the divergence. Therefore, increasing $b$ makes it harder to distinguish which dataset produces the sketch, which implies a higher privacy level.

**Remark 2.4.** For the original Gaussian mechanism, i.e., $\mathcal{G}(\gamma_t(D)) = \gamma_t(D) + \xi'_t$ where $\xi'_t \sim \mathcal{N}\left(0, \sigma_g^2 \mathbb{I}_d\right)$, $\mathcal{G}(\gamma_t(D))$ and $\mathcal{G}(\gamma_t(D'))$ are Gaussian distributions with the same variance $\sigma_g^2$, but different means $\gamma_t(D)$ and $\gamma_t(D')$ respectively. So their Rényi divergence is $\frac{\alpha^2 \|\gamma_t(D) - \gamma_t(D')\|^2}{2\sigma^2} \leq \frac{\alpha^2 \tau^2}{2\sigma^2}$, which is fixed for any dimensions. In contrast, for the SGM, $\mathcal{SG}\left(\gamma_t(D)\right)$ and $\mathcal{SG}\left(\gamma_t(D')\right)$ are two Gaussians with the same mean of $0$ but different variances $\frac{\|\gamma_t(D)\|_2^2}{b} + m\sigma_g^2$ and $\frac{\|\gamma_t(D')\|_2^2}{b} + m\sigma_g^2$, and their Rényi divergence is proportional to $O(\frac{1}{b})$ according to Lemma 2.2. This offers an intuitive justification for the additional $1/\sqrt{b}$ factor in Theorem 2.2's privacy bound, as compared to the privacy guarantee of the standard Gaussian mechanism [1]. $\qquad\square$

---

**Algorithm 2** Fed-SGM

---

**Hyperparameters:** server learning rate $\eta_{\text{global}}$, local learning rate $\eta_{\text{local}}$, noise scale $\sigma_g$, clipping threshold $\tau$, number of rounds $T$.
**Inputs:** local datasets $D_c = \{(x_{i,c}, y_{i,c})\}_{i=1}^{n_c}$ and loss function $\mathcal{L}_c(\theta) = \frac{1}{n_c} \sum_{i=1}^{n_c} \ell\big(\theta, (x_{i,c}, y_{i,c})\big)$ for clients $c \in [C]$.

  **Initialize** $\theta_0$ randomly.
  **for** $t = 0, \dots, T-1$ **do**
    Take a subset $\mathcal{C}_t$ of $N$ clients with sampling probability $q = \frac{N}{C}$
    **On Client Nodes:**
    **for** $c \in \mathcal{C}_t$ **do**
      **Assign local initialization:** $\theta_{c,t,0} \leftarrow \theta_t$
      **Local update:**
      **for** $k = 1, \dots, K$ **do**
        $\theta_{c,t,k} \leftarrow \theta_{c,t,k-1} - \eta_{\text{local}} \cdot g_{c,t,k}$
      **end for**
      **Compute update:** $\Delta_{c,t} \leftarrow \theta_t - \theta_{c,t,K}$
      **Clip the update:** $\hat{\Delta}_{c,t} = \text{clip}\left(\frac{\Delta_{c,t}}{\eta_{\text{local}}}, \tau\right)$
      **Apply SGM:** $\tilde{\Delta}_{c,t} = \eta_{\text{local}} \mathcal{SG}(\hat{\Delta}_{c,t}(x_i); R_t, \mathbf{z}_{c,t})$
      Send $\tilde{\Delta}_{c,t}$ to the server
    **end for**
    **On Server Node:**
    **Aggregate:** $\bar{\tilde{\Delta}}_t = \frac{1}{N} \sum_{c \in \mathcal{C}_t} \tilde{\Delta}_{c,t}$
    Broadcast $\bar{\tilde{\Delta}}_t$ to the clients
    **On Client Nodes:**
    **Update parameter:** $\theta_{t+1} \leftarrow \texttt{GLOBAL\_OPT}\left(\theta_t, R_t^\top \bar{\tilde{\Delta}}_t, \eta_{\text{global}}\right)$
  **end for**
**Outputs:** final model $\theta_T$

---

## 3 Application in FL: Fed-SGM

In this section, we introduce Fed-SGM, the application of the SGM within a federated learning framework that simultaneously achieves communication efficiency and differential privacy guarantees. We then establish its client-level privacy bounds and derive optimization convergence results.

We consider a federated learning setting with $C$ clients indexed by $c \in [C]$. Each client $c$ holds a local dataset $\mathcal{D}_c = \{(x_{i,c}, y_{i,c})\}_{i=1}^{n_c}$ of size $n_c$ and defines its empirical loss by $\mathcal{L}_c(\theta) = \frac{1}{n_c} \sum_{i=1}^{n_c} \ell\big(\theta; (x_{i,c}, y_{i,c})\big)$, where $\theta \in \mathbb{R}^d$ is the model parameter and $\ell$ the per-example loss. The goal of the algorithm is to minimize the average empirical loss over clients, i.e., $\mathcal{L}(\theta) := \frac{1}{C} \sum_{c=1}^C \mathcal{L}_c(\theta)$.

Algorithm 2 formalizes the Fed-SGM algorithm. At each communication round, the server samples a subset of $N$ clients uniformly at random without replacement. Each selected client $c$ then:

1. Performs local stochastic gradient descent (SGD) on its local dataset to obtain an update $\Delta_{c,t}$.

2. Clips $\frac{\Delta_{c,t}}{\eta_{\text{local}}}$ with respect to the clipping threshold $\tau$.

3. Applies the SGM to the clipped update and transmits the resulting sketch to the server.

Upon receiving client sketches, the server computes their aggregate and broadcasts this aggregated sketch back to every client. Each client inverts the sketch to recover the aggregated update in the original ambient space, and then updates the global model parameters in a single step via the $\texttt{GLOBAL\_OPT}$ operator. In this paper, we study the case with gradient descent (GD) and AMSGrad as $\texttt{GLOBAL\_OPT}$.

**Remark 3.1.** In the parameter-update step, we lift the aggregated sketch back to the ambient space using the transpose $R_t^\top$, rather than the least-squares estimator $R_t^\dagger$, i.e., the Moore–Penrose pseudo-inverse. This preference is motivated by the following points. First, since each entry of the sketching matrix $R_t \in \mathbb{R}^{b \times d}$ is drawn i.i.d. Gaussian with mean 0 and variance $\frac{1}{\sqrt{b}}$, so $\mathbb{E}\left[R_t^\top R_t\right] = \mathbb{I}_d$. Therefore, for any vector $g$, $\mathbb{E}\left[R_t^\top R_t g\right] = g$, which shows that $R_t^\top$ can recover $g$ in expectation,

implying $R_t^\top$ is a near-optimal desketching operator for our use. Second, computing $R_t^\dagger$ each round requires solving a least-squares system, typically via QR/SVD, with per-round cost on the order of $\mathcal{O}\left(b^2 d\right)$ plus the overhead of forming and inverting a Gram matrix. In contrast, the transpose map $R_t^\top$ is a single matrix–vector multiplication with cost $\mathcal{O}(bd)$, which is substantially cheaper.

## 3.1 Privacy Guarantee

As a direct application of Theorem 2.2, we can obtain the following client-level privacy guarantee of Algorithm 2.

**Theorem 3.1.** *There exists constants $c_3$ and $c_4$ so that given the sampling probability $q = \frac{N}{C}$ and the number of communication rounds $T$, for any $\epsilon_p \leq c_3 q \sqrt{T}$, Algorithm 2 is $(\epsilon_p, \delta_p)$-client-level private for any $\delta_p > 0$ if we choose*

$$\sigma_g^2 \geq \frac{c_4 q \tau^2 \sqrt{T} \log(2qT/\delta_p)}{\sqrt{b}\epsilon_p} . \tag{5}$$

## 3.2 Convergence Analysis

In this section, we analyze the optimization performance of Fed-SGM as specified in Algorithm 2. Section 3.2.1 introduces our assumptions on the loss-gradient bounds and Hessian spectrum, and justifies their empirical validity via prior work. Section 3.2.2 then establishes convergence guarantees for Fed-SGM when employing AMSGrad as the `GLOBAL_OPT`, with the corresponding gradient-descent analysis deferred to Appendix D.

### 3.2.1 Assumptions

We begin by presenting a set of standard assumptions that are widely adopted in the literature on first-order stochastic methods.

**Assumption 1** (Bounded Loss Gradients). *There exists a constant $G \geq 0$, such that for every $\theta \in \mathbb{R}^p$ and $c \in [C]$, $\|\nabla \mathcal{L}_c(\theta)\|_2 \leq G$.*

According to the definition, $\nabla \mathcal{L}(\theta) = \frac{1}{C} \sum_{c=1}^C \nabla \mathcal{L}_c(\theta)$, so we can directly see that $\|\nabla \mathcal{L}(\theta)\|_2 \leq G$. Therefore, $G$ is the upper bound for both the client and global gradient norms. We also assume the stochastic noise from mini-batches is sub-Gaussian, which is widely adopted in first-order optimization [25, 48].

**Assumption 2** (Sub-Gaussian Noise). *The stochastic noise $\|\nabla \mathcal{L}_c(x) - g_c(x)\|_2$ at each client is a $\sigma_s$-sub-Gaussian random variable, i.e. $\mathbb{P}(\|\nabla \mathcal{L}_c(x) - g_c(x)\|_2 \geq a) \leq 2\exp(-a^2/\sigma_s^2)$, for all $a \geq 0$.*

Besides, we also assume a special structure on the Hessian eigenspectrum $\{\lambda_i, v_i\}_{i=1}^d$ of the loss function $\ell$.

**Assumption 3** (Bounded Loss Hessian Eigenvalues). *For each $c \in [C]$, the smoothness of the loss function $\mathcal{L}_c$, i.e. the largest eigenvalue of the loss Hessian $H_{\mathcal{L}_c}$ is bounded by $L$.*

This local smoothness assumption is commonly used in many federated learning analysis [55, 20]. Due to a similar reason as Assumption 1, this assumption also indicates the $L$-smoothness of the average loss function $\mathcal{L}$.

**Assumption 4** (Loss Hessian Eigenspetrum). *Denote $\{\lambda_i\}_{i=1}^d$ the eigenvectors of the Hessian matrix $H_{\mathcal{L}}$ of the average loss function $\mathcal{L}$. Then we assume the absolute intrinsic dimension of $H_{\mathcal{L}}$ is bounded, i.e., $\frac{\sum_{i=1}^d |\lambda_i|}{\max_i \lambda_i} \leq \mathcal{I}$.*

The absolute intrinsic dimension considered here, $\frac{\sum_{i=1}^d |\lambda_i|}{\max_i \lambda_i}$, is close to the concept of intrinsic dimension proposed in [26], and the difference is that we consider absolute values of eigenvalues. A growing amount of empirical research suggests that the absolute intrinsic dimension of the Hessian in deep learning can be significantly smaller than the ambient dimension $d$. Studies by [22, 38, 42] indicate that the eigenspectrum undergoes a sharp decay in magnitude. Additionally, research by [56, 39] demonstrates that a substantial portion of the eigenspectrum is concentrated near zero. Further investigations by [76, 82] reveal that the eigenvalues follow a power-law distribution, suggesting that in such cases, the absolute intrinsic dimension remains a much smaller order than $d$.

### 3.2.2 Optimization Analysis

In this section, we will demonstrate the optimization result of Algorithm 2 with AMSGrad as `GLOBAL_OPT`. We refer the reader to Appendix E for a more formal statement with the full analysis.

**Theorem 3.1.** [Informal version of Theorem E.1] Suppose $\{\theta_t\}_{t=0}^T$ is generated by Algorithm 2 with AMSGrad as GLOBAL_OPT. Denote $\mathcal{L}^*$ the minimum of the average empirical loss. Under Assumption 1-4, with learning rate $\eta = \eta_{\text{global}}\eta_{\text{local}}$, we have that with probability at least $1 - \Theta(\delta)$,

$$\frac{1}{T}\sum_{t=1}^T \|\nabla\mathcal{L}(\theta_t)\|_2^2 \le E_s^{\text{AMSGrad}} + E_c^{\text{AMSGrad}} + E_g^{\text{AMSGrad}}$$

in which $E_s^{\text{AMSGrad}}$, $E_c^{\text{AMSGrad}}$ and $E_g^{\text{AMSGrad}}$ denote terms from sketching-based FedAvg algorithm, caused by clipping, and caused by Gaussian noises. Specifically, with $\tau_{K,G} = \min\{\tau, KG\}$,

$$E_s^{\text{AMSGrad}} = \mathcal{O}\left(\frac{\mathcal{L}(\theta_0)-\mathcal{L}^*}{\eta TK}\right) + \tilde{\mathcal{O}}\left(\frac{1}{\sqrt{NT}}\right) + \tilde{\mathcal{O}}\left(\eta_{\text{local}}K\right) + \tilde{\mathcal{O}}\left(\frac{\tau_{K,G}}{\sqrt{b}TK}\right) + \tilde{\mathcal{O}}\left(\frac{(\eta_{\text{local}}+\eta\mathcal{I})\tau_{K,G}^2}{K}\right) + \tilde{\mathcal{O}}\left(\frac{1}{\eta TK}\right)$$

$$E_c^{\text{AMSGrad}} = \max\left\{0, \frac{\left(\epsilon + \tilde{\mathcal{O}}(\eta_{\text{local}})\right)G(KG-\tau)}{K\epsilon}\right\}$$

$$E_g^{\text{AMSGrad}} = \tilde{\mathcal{O}}\left(\frac{(\eta_{\text{local}}+\eta\mathcal{I})\sigma_g^2}{NK}\right)$$

**Remark 3.2.** Regarding the term $E_s^{\text{AMSGrad}}$, most of the optimization results for sketching-based algorithms are expectation results [59, 54], while ours is a high-probability optimization bound for the sketching-based FedAvg algorithm. In addition, previous optimization results in expectation either have a linear dependence on the ambient dimension $d$ [59], or get around the dependence by using Top-k components of the gradient vector with rely on heavy-hitter assumptions [54, 80]. As a contrast, our result $E_s^{\text{AMSGrad}}$ only has a logarithmic dimensional dependence from the need of high probability with no additional mechanisms.

**Remark 3.3.** When there is no clipping activated, i.e., $\tau \ge KG$, with learning rates $\eta_{\text{local}} = \mathcal{O}\left(\frac{\sqrt{\mathcal{I}}}{\sqrt{T}K}\right)$, $\eta = \mathcal{O}\left(\frac{1}{\sqrt{T\mathcal{I}}K}\right)$, we can get a the term $E_s^{\text{AMSGrad}}$ at the order of $\tilde{\mathcal{O}}\left(\sqrt{\frac{\mathcal{I}}{T}}\right)$. When we clip with a constant, i.e., $\tau = \mathcal{O}(1)$, by setting $\eta_{\text{local}} = \mathcal{O}\left(\frac{1}{\sqrt{NT}K}\right)$, $\eta = \mathcal{O}\left(\frac{1}{\sqrt{T\mathcal{I}}}\right)$, the order of $E_s^{\text{AMSGrad}}$ becomes $\tilde{\mathcal{O}}\left(\frac{1}{\sqrt{NT}} + \frac{\sqrt{\mathcal{I}}}{\sqrt{T}K}\right)$.

**Remark 3.4.** Regarding the term $E_c^{\text{AMSGrad}}$ caused by clipping, it is a reasonable term in the sense that it is monotonically decreasing with the clipping threshold $\tau$. In the extreme case, when $\tau = 0$, all the clipped updates will become 0, so that there is no training, then $E_c^{\text{AMSGrad}} \approx G^2$ is a natural bound of $\frac{1}{T}\sum_{t=1}^T \|\nabla\mathcal{L}(\theta_t)\|_2^2 = \|\nabla\mathcal{L}(\theta_0)\|_2^2$ with Assumption 1; when $\tau \ge KG$, there is no clipping activated during training, which is aligned with the fact that $E_c^{\text{AMSGrad}} = \max\left\{0, \frac{(\epsilon+\tilde{\mathcal{O}}(\eta_{\text{local}}))G(KG-\tau)}{K\epsilon}\right\} = 0$.

**Remark 3.5.** Regarding the term $E_g^{\text{AMSGrad}}$ caused by Gaussian noises, with the learning rates mentioned in Remark 3.3, $E_g^{\text{AMSGrad}} = \tilde{\mathcal{O}}\left(\frac{\sqrt{\mathcal{I}}\sigma_g^2}{N\sqrt{T}K}\right)$. Besides, $E_g^{\text{AMSGrad}}$ does not have any dependence on either the ambient dimension $d$ or the sketching dimension $b$.

**Remark 3.6.** From the proof in Appendix E, we can see that our analysis on AMSGrad can be generalized to any adaptive optimizers involved with first-order and second-order moments, including Adam [32], Yogi [78], etc.

## 4 Experiments

We conduct empirical evaluations of Fed-SGM on neural network training for both vision and language benchmarks. Across various privacy budgets $\epsilon_p$, we compare Fed-SGM using GD and Adam as the global optimizer against their unsketched counterparts—DP-FedAvg [81] for the GD variant and a matched unsketched Adam implementation, and additionally benchmark against DiffSketch [37]. The results are presented after a detailed description of the experimental setup.

**Datasets and Network Structure.** We adopt two different experiment settings including both vision and language tasks. For the vision task, We use the full EMNIST ByClass dataset, which comprises 814K training samples and 140K testing samples across 62 classes, representing the complete set of

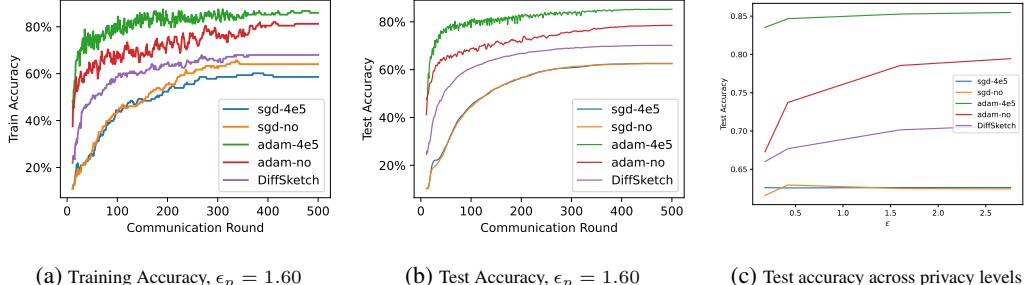

(a) Training Accuracy, $\epsilon_p = 1.60$  (b) Test Accuracy, $\epsilon_p = 1.60$  (c) Test accuracy across privacy levels

Figure 1: (a)(b) Comparison of Fed-SGM with ADAM, Fed-SGM with GD, DP-FedAvg and its Adam variant of ResNet101 trained on EMNIST with $\epsilon_p = 1.6$. The X-axis is the number of communication rounds $T$, and the Y-axis is the train/test accuracy. (c) The trend of test accuracy over privacy levels. The X-axis is the $\epsilon_p$, and the Y-axis is the test accuracy. In all three subfigures, 'sgd' and 'adam' denote the selection of GD and Adam as the global optimizers, respectively; '4e5' signifies a sketching dimension of $4 \times 10^5$, and 'no' indicates that no sketching is applied.

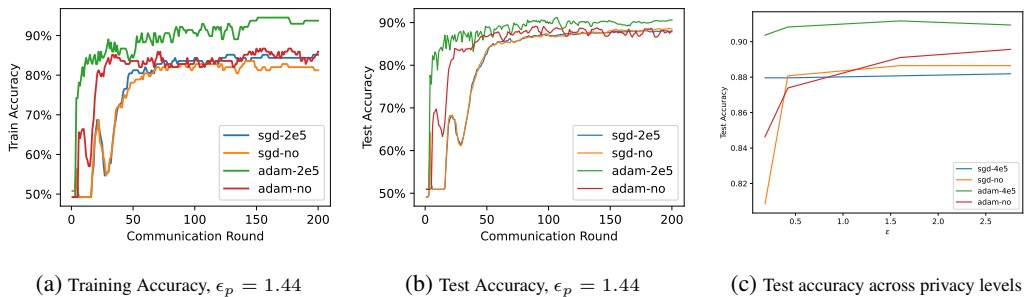

(a) Training Accuracy, $\epsilon_p = 1.44$  (b) Test Accuracy, $\epsilon_p = 1.44$  (c) Test accuracy across privacy levels

Figure 2: (a)(b) Comparison of Fed-SGM with ADAM, Fed-SGM with GD, DP-FedAvg and its Adam variant of Bert finetuned on SST-2 with $\epsilon_p = 1.44$. The X-axis is the number of communication rounds $T$, and the Y-axis is the train/test accuracy. (c) The trend of test accuracy over privacy levels. The X-axis is the $\epsilon_p$, and the Y-axis is the test accuracy. In all three subfigures, 'sgd' and 'adam' denote the selection of GD and Adam as the global optimizers, respectively; '2e5' signifies a sketching dimension of $2 \times 10^5$, and 'no' indicates that no sketching is applied.

handwritten characters. We conduct experiments on ResNet101 [74] with a total of 42M parameter. For the language task, we use the SST-2 dataset from the GLUE benchmark [69], which comprises 67349 training samples and 1821 test samples across two sentiment classes. We finetune a BERT-Base model [14], comprising approximately 100M parameters.

**Parameter Setting** We deploy $C = 625$ clients in total, sampling $N = 4$ clients uniformly at random in each communication round. Each selected client executes $K = 18$ local SGD updates on mini-batches of size 64, with gradient clipping threshold $\tau = 1$. Sketching dimension and total rounds are chosen per task: for the vision task, we set $b = 4 \times 10^5$ (approximately 1% compression rate) and run $T = 500$ communication rounds; for the language task, we use $b = 2 \times 10^5$ (approximately 0.2% compression rate) and $T = 200$ rounds.

**Privacy Level and Noise calculation:** For privacy, we fix the parameter $\delta_p = 10^{-5}$. For both tasks, we consider noise scales $\sigma_g \in \{0.8, 1, 2, 4\}$ for the unsketched algorithms. By employing the Moments Accountant method [1, 8], we compute the cumulative privacy loss and obtain approximately $\{2.75, 1.60, 0.42, 0.18\}$ for the vision task and $\{2.45, 1.44, 0.35, 0.12\}$ for the language task, each corresponding to the respective noise scales. For Fed-SGM, at each privacy level, we compute the minimal noise scale $\sigma_g$ by calculating the RDP and minimizing over the Rényi order $\alpha$ numerically [46]. This yields the noise levels approximately $\{0.0883, 0.1013, 0.1588, 0.2265\}$ for the vision task and $\{0.0948, 0.1071, 0.1664, 0.2580\}$ for the language task. Across all considered privacy budgets in both learning tasks, Fed-SGM attains the same privacy guarantee with strictly lower Gaussian noise variance than the corresponding unsketched algorithm.

**Experimental Results** We report our experimental results in Figure 1 and 2. Additional figures and ablation studies are presented in Appendix F. For each task, Figures 1a and 2a depict the training accuracy of the five algorithms at a fixed privacy budget, whereas Figures 1b and 2b show the corresponding test accuracy. Additionally, Figures 1c and 2c present a trend of the test accuracies of these algorithms under various privacy levels, providing deeper insight into their performance across different privacy constraints. Based on these figures, the following observations can be made:

- Regardless of the choice of global optimizer, Fed-SGM consistently matches or surpasses its unsketched counterpart, confirming the effectiveness of sketching within a differentially private FL framework. Notably, when Adam is employed, Fed-SGM even outperforms the non-sketching baseline. This improvement arises because Fed-SGM always requires a strictly lower Gaussian noise variance than the unsketched mechanism to attain the same privacy level. The resulting reduction in the amount of injected noise may compensate for any performance degradation typically associated with sketching.

- Irrespective of whether sketching is employed, the variant using Adam consistently outperforms the ones using gradient descent and the baseline DiffSketch. Notably, even the Adam variant with sketching surpasses the gradient descent variant without sketching, and this trend is maintained across different privacy levels. These results demonstrate the performance enhancement achieved by employing Adam.

**Remark 4.1.** For experiment results in Figure 1 and Figure 2, we use an IID split of the training dataset across all 625 clients. To assess robustness under heterogeneity, we rerun our image-classification experiments with a non-IID partition using a Dirichlet distribution with concentration parameter 0.05. Table 1 compares test accuracy for models with and without sketching, under both SGD and AMAGrad global optimizers. As expected, overall accuracy decreases under non-IID settings; however, AMSGrad variants still outperforms SGD counterparts, and sketching methods remains competitive with and sometimes outperforms non-sketching ones.

Table 1: Test accuracy under IID vs. Non-IID data.

|          | GD, Sketching | GD, Non-Sketching | Adam, Sketching | Adam, Non-Sketching |
|----------|---------------|-------------------|-----------------|---------------------|
| IID      | 62.98%        | 63.34%            | 85.09%          | 78.60%              |
| Non-IID  | 40.71%        | 42.31%            | 52.55%          | 51.42%              |

**Remark 4.2.** We use Adam as a representative adaptive optimizer in our experiments. To assess sensitivity to the optimizer choice, we repeated the image-classification experiments with AMSGrad under the same settings as Figure 1. As shown in Table 2, the results with AMSGrad are comparable to those with Adam.

Table 2: Adam vs. AMSGrad as the adaptive global optimizer.

|         | Sketching | Non-Sketching |
|---------|-----------|---------------|
| Adam    | 85.09%    | 78.60%        |
| AMSGrad | 85.04%    | 78.10%        |

## 5 Conclusion

In conclusion, we introduced the Sketched Gaussian Mechanism (SGM), which combines an isometric sketching transform with the classical Gaussian mechanism, and showed via Rényi differential privacy—together with subsampling, post-processing, and composition theorems—that its privacy loss satisfies $\epsilon = O\left(\frac{1}{\sqrt{b}\sigma_g^2}\right)$ without imposing any restrictive upper bound on the sketching dimension $b$, thereby demonstrating inherent privacy amplification from sketching. We then embed SGM into a federated learning framework (Fed-SGM) supporting arbitrary server optimizers and prove convergence guarantees that grow only logarithmically in the ambient dimension $d$ and linearly in the empirically small absolute intrinsic Hessian dimension $\mathcal{I}$. Empirical results on vision and language benchmarks confirm that Fed-SGM attains a fixed privacy budget with strictly less noise than unsketched DP-FedAvg, consistently matches or exceeds model accuracy, and benefits further from adaptive optimization. As a direction for future work, we note that our analysis is currently limited to isotropic Gaussian sketching matrices. It therefore remains to establish whether comparable privacy and convergence guarantees hold for more general classes of sketching transforms.

## Acknowledgments

The work was supported by the National Science Foundation (NSF) through awards IIS 21-31335, OAC 21-30835, DBI 20-21898, as well as a C3.ai research award. Compute support for the work was provided by the National Center for Supercomputing Applications (NCSA) and the Illinois Campus Cluster Program (ICCP).

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

# A Related Work

**Communication-Efficient Distributed Learning.** The substantial cost of transmitting model updates between clients and the central server has driven recent efforts to enhance communication efficiency in both distributed and federated learning. One widely adopted method, FedAvg [44], reduces communication frequency by allowing clients to conduct multiple local updates within each training round before syncing with the server. Another prevalent approach is compressing model updates before transmission, lowering the communication burden per round. These compression strategies generally fall into five categories: sparsification [3, 70, 40], quantization [61, 4, 73], low-rank factorization [67, 47, 68], sketching [54, 59, 29], and sparse subnetwork training [27, 33, 34]. While some of these techniques, such as certain quantization methods [4], naturally maintain unbiasedness, many introduce bias and require additional mechanisms to mitigate it for improved convergence [40, 54]. Another important characteristic is linearity, which guarantees that the geometric properties remain largely intact after compressing [13]. Among these techniques, sketching is particularly notable for its simplicity as a linear and unbiased transformation, allowing computations to be performed in the lower-dimensional space before reconstruction via desketching.

**Privacy in Federated Learning.** Differential Privacy (DP) [17] is the commonly used rigorous privacy guarantees in machine learning. In centralized training, the standard approach for ensuring DP follows a simple procedure of applying a clipping operation to the stochastic gradient, and introducing random Gaussian noise to the clipped gradient [1]. The clipping step plays a crucial role in enforcing DP, as the required noise variance is directly influenced by the chosen clipping threshold. [18]. Privacy mechanisms involving clipping are also widely applied in federated learning scenarios, but various requirements and factors result in different clipping operations. For sample-level privacy, clipping and injecting noise to every local update is proposed [65] while causing noticeable decline in performance. For client-level privacy, local models are clipped before transmission and perturbed bounded parameters [72, 66]. Later the mechanism to clip local updates instead was raised and turns out to have better numerical performance than model clipping [21, 71, 63].

**Sketching.** For decades, sketching has served as a core tool across various applications, predating the rise of deep learning in the 2010s [12, 23, 31]. It has been widely used in tasks such as low-rank approximation [64], graph sparsification [2], and least squares regression [15]. More recently, sketching has been increasingly employed in distributed and federated learning to compress model updates, thereby improving communication efficiency [29, 28, 54, 59, 24, 58]. Sketching-based frameworks have also been seamless integrated with secure aggregation and differential privacy mechanisms [11, 59, 45, 85, 6]. However, most of these approaches restrict their privacy analysis to the randomness injected after sketching and do not quantify any privacy amplification inherent to the sketching transformation itself. [37] is among the few to investigate this intrinsic privacy contribution of sketching.

**Adaptive Optimizers** [32] introduced Adam, an optimizer that has demonstrated rapid convergence and robustness to hyper-parameter choices. Adagrad [16] and RMSprop [62] update parameters using the gradient directly rather than relying on momentum. Additionally, Adadelta [79] modifies Adam's variance term to follow a non-decreasing update rule, and AdaBound [43] introduces both upper and lower bounds for this variance component. Most of these adaptive optimizers rely on first-order and second-order moment estimates, which is central to how these adaptive methods balance rapid progress during early training with more stable convergence later on [32]. [51] introduces a general federated optimization framework integrating adaptive server optimizers, demonstrating improved convergence rates and empirical performance in heterogeneous federated learning settings.

# B Sketching Guarantee

We have the following lemma of sketching guarantee from [59] for Gaussian sketching matrix.

**Lemma B.1** (Lemma D.24 from [59]). Let $R \in \mathbb{R}^{b \times d}$ denote a random Gaussian matrix . Then for any fixed vector $h \in \mathbb{R}^d$ and any fixed vector $g \in \mathbb{R}^d$, the following property holds:

$$\Pr_{R \sim \Pi} \left[ |(g^\top R^\top Rh) - (g^\top h)| > \frac{\log^{1.5}(d/\delta)}{\sqrt{b}} \|g\|_2 \|h\|_2 \right] \leq \Theta(\delta).$$

## C  Privacy Analysis

In this section, we will provide the proof of Theorem 2.1 and Theorem 2.2 on the privacy guarantee of Algorithm 1.

### C.1  Proof of Theorem 2.1

Following the analysis of [1], we provide a proof of Theorem 2.1. We restate the theorem first.

**Theorem C.1.** There exists constants $c_1$ and $c_2$ so that given the sampling probability $q = \frac{m}{n}$ and the number of steps $T$, for any $\varepsilon_p < c_1 q^2 T$, Algorithm 1 is $(\epsilon_p, \delta_p)$-differentially private for any $\delta_p > 0$ if we choose

$$\sigma_g \geq c_2 \frac{\tau \sqrt{\left(1 + \frac{\log^{1.5}(2mT/\delta_p)}{\sqrt{b}}\right) mT \log(2/\delta_p)}}{n\varepsilon_p}.$$

*Proof.* Since the process after getting $\bar{\bar{g}}_t$ can be viewed as post-processing, and due to the post-processing property of differential privacy, we only need to explore the privacy guarantee till the aggregation step in Algorithm 1.

According to Lemma B.1, for each $t \in [T]$, $i \in L_t$, we have that

$$\mathbb{P}\left[\|R_t \hat{g}_t(x_i)\|^2 \geq \left(1 + \frac{\log^{1.5}(2mT/\delta_p)}{\sqrt{b}}\right)\|\hat{g}_t(x_i)\|^2\right] \leq \frac{\delta_p}{2mT}$$

Therefore, with probability at least $1 - \frac{\delta_p}{2}$, for all $t \in [T]$ and $i \in L_t$,

$$\|R_t \hat{g}_t(x_i)\| \leq \sqrt{1 + \frac{\log^{1.5}(2mT/\delta_p)}{\sqrt{b}}} \|\hat{g}_t(x_i)\| \leq \tau \sqrt{1 + \frac{\log^{1.5}(2mT/\delta_p)}{\sqrt{b}}}$$

Denote $\mathcal{E} = \left\{\|R_t \hat{g}_t(x_i)\| \leq \tau\sqrt{1 + \frac{\log^{1.5}(2mT/\delta_p)}{\sqrt{b}}}, \forall t \in [T], \forall i \in L_t\right\}$, then we have $\mathbb{P}(\mathcal{E}) \geq 1 - \frac{\delta_p}{2}$. There are two difference cases:

**Case 1:** When $\mathcal{E}$ happens, the function $f$ mentioned in Lemma 3 in [1] is still bounded by a constant. Therefore, we can still follow the Moments Accountant (MA) method in Lemma 3, Theorem 1, and Theorem 2 in [1] to get that Algorithm 1 is $\left(\varepsilon_p, \frac{\delta_p}{2}\right)$-differentially private for any $\delta_p > 0$ if

$$\sigma_g \geq c_3 \frac{\tau\sqrt{\left(1 + \frac{\log^{1.5}(2mT/\delta_p)}{\sqrt{b}}\right) mT \log\left(\frac{2}{\delta_p}\right)}}{n\varepsilon_p}$$

**Case 2:** When $\mathcal{E}$ does not happen, we have $\mathbb{P}(\mathcal{E}^c) \leq \frac{\delta_p}{2}$.

Combining these two cases, so we can get that Algorithm 1 is $(\varepsilon_p, \delta_p)$-differentially private with

$$\sigma_g \geq c_3 \frac{\tau\sqrt{\left(1 + \frac{\log^{1.5}(2mT/\delta_p)}{\sqrt{b}}\right) mT \log\left(\frac{2}{\delta_p}\right)}}{n\varepsilon_p}.$$

Then we finish the proof. □

### C.2  Proof of Theorem 2.2

We will establish our analysis in the framework of [46]. We will restate the concepts and proposition here.

**Definition C.1** (Rényi divergence). For two probability distributions $P$ and $Q$ defined over $\mathcal{R}$, the Rényi divergence of order $\alpha > 1$ is

$$D_\alpha(P\|Q) \triangleq \frac{1}{\alpha - 1} \log \mathbb{E}_{x \sim Q}\left(\frac{P(x)}{Q(x)}\right)^\alpha.$$

**Definition C.2** $((\alpha, \epsilon)$-RDP$)$. A randomized mechanism $f : \mathcal{D} \to \mathcal{R}$ is said to have $\epsilon$-Rényi differential privacy of order $\alpha$, or $(\alpha, \epsilon)$-RDP for short, if for any adjacent $D, D' \in \mathcal{D}$, it holds that

$$D_\alpha\left(f(D)\|f(D')\right) \le \epsilon.$$

**Lemma C.2** (Lemma 2.1).

$$D_\alpha\left(\mathcal{SG}(\gamma_t(D); R_t, \xi_t)\|\mathcal{SG}(\gamma_t(D'); R_t, \xi_t)\right) = bf_\alpha\left(\sqrt{\frac{\|\gamma_t(D')\|^2 + mb\sigma_g^2}{\|\gamma_t(D)\|^2 + mb\sigma_g^2}}\right)$$

where

$$f_\alpha(x) = \log x + \frac{1}{2(\alpha - 1)}\log\frac{x^2}{\alpha x^2 + 1 - \alpha}.$$

*Proof.* According to the definition of SGM,

$$\mathcal{SG}(\gamma_t(D); R_t, \xi_t) \sim \mathcal{N}\left(0, \left(\frac{\|\gamma_t(D)\|^2}{b} + m\sigma_g^2\right)\mathbb{I}_b\right);$$

$$\mathcal{SG}(\gamma_t(D'); R_t, \xi_t) \sim \mathcal{N}\left(0, \left(\frac{\|\gamma_t(D')\|^2}{b} + m\sigma_g^2\right)\mathbb{I}_b\right)$$

Denote

$$\sigma_1^2 \;=\; \frac{\|\gamma_t(D)\|_2^2}{b} + m\sigma_g^2, \qquad \sigma_2^2 \;=\; \frac{\|\gamma_t(D')\|_2^2}{b} + m\sigma_g^2.$$

We need to compute the order-$\alpha$ Rényi divergence between $P = \mathcal{N}(0, \sigma_1^2 I_b)$ and $Q = \mathcal{N}(0, \sigma_2^2 I_b)$. According to definition of multivariate Gaussian distribution, for $x \in \mathbb{R}^b$, we can write density functions

$$P(x) = (2\pi\sigma_1^2)^{-b/2}\exp\left(-\frac{\|x\|_2^2}{2\sigma_1^2}\right),$$

$$Q(x) = (2\pi\sigma_2^2)^{-b/2}\exp\left(-\frac{\|x\|_2^2}{2\sigma_2^2}\right).$$

Hence

$$\begin{aligned}
\mathbb{E}_{x\sim Q}\left[\left(\frac{P(x)}{Q(x)}\right)^\alpha\right] &= \int_{\mathbb{R}^b} Q(x)\cdot\left(\frac{P(x)}{Q(x)}\right)^\alpha dx \\
&= \int_{\mathbb{R}^b} P(x)^\alpha Q(x)^{1-\alpha}dx \\
&= \int_{\mathbb{R}^b}\left((2\pi\sigma_1^2)^{-\frac{b}{2}}\exp\left(-\frac{\|x\|_2^2}{2\sigma_1^2}\right)\right)^\alpha\cdot\left((2\pi\sigma_2^2)^{-\frac{b}{2}}\exp\left(-\frac{\|x\|_2^2}{2\sigma_2^2}\right)\right)^{1-\alpha}dx \\
&= (2\pi\sigma_1^2)^{-\frac{\alpha b}{2}}(2\pi\sigma_2^2)^{-\frac{(1-\alpha)b}{2}}\int_{\mathbb{R}^b}\exp\left(-\frac{\|x\|_2^2}{2}\left(\frac{\alpha}{\sigma_1^2}+\frac{1-\alpha}{\sigma_2^2}\right)\right)dx \\
&\stackrel{(i)}{=} (2\pi\sigma_1^2)^{-\frac{\alpha b}{2}}(2\pi\sigma_2^2)^{-\frac{(1-\alpha)b}{2}}\cdot\left(\frac{\pi}{\frac{1}{2}\left(\frac{\alpha}{\sigma_1^2}+\frac{1-\alpha}{\sigma_2^2}\right)}\right)^{\frac{b}{2}} \\
&= \sigma_1^{-\alpha b}\sigma_2^{-(1-\alpha)b}\left(\frac{\alpha}{\sigma_1^2}+\frac{1-\alpha}{\sigma_2^2}\right)^{-\frac{b}{2}} \\
&= \left(\alpha\sigma_2^2 + (1-\alpha)\sigma_1^2\right)^{-\frac{b}{2}}\sigma_2^{\alpha b}\sigma_1^{(1-\alpha)b},
\end{aligned}$$

where (1) uses the standard Gaussian integral $\int_{\mathbb{R}^b} e^{-a\|x\|_2^2/2}dx = (2\pi/a)^{b/2}$ for $a > 0$. Therefore, following the definition of Rényi divergence in Definition C.1,

$$D_\alpha\left(P\|Q\right) = \frac{1}{\alpha - 1}\log\mathbb{E}_{x\sim Q}\left[\left(\frac{P(x)}{Q(x)}\right)^\alpha\right]$$

$$= \frac{1}{\alpha - 1} \log \left( \left( \alpha \sigma_2^2 + (1 - \alpha) \sigma_1^2 \right)^{-\frac{b}{2}} \sigma_2^{\alpha b} \sigma_1^{(1-\alpha)b} \right)$$

$$= \frac{1}{\alpha - 1} \left( -\frac{1}{2} \log \left( \alpha \sigma_2^2 + (1 - \alpha) \sigma_1^2 \right)^b + \alpha \log \sigma_2^b + (1 - \alpha) \log \sigma_1^b \right)$$

$$= \frac{1}{\alpha - 1} \left( -\frac{1}{2} \log \left( \alpha \sigma_2^2 + (1 - \alpha) \sigma_1^2 \right)^b + \log \sigma_2^b + (\alpha - 1) \log \sigma_2^b - (\alpha - 1) \log \sigma_1^b \right)$$

$$= \frac{1}{2(\alpha - 1)} \left( \log \left( \sigma_2^2 \right)^b - \log \left( \alpha \sigma_2^2 + (1 - \alpha) \sigma_1^2 \right)^b \right) + \left( \log \sigma_2^b - \log \sigma_1^b \right)$$

$$= \log \left( \frac{\sigma_2}{\sigma_1} \right)^b + \frac{1}{2(\alpha - 1)} \log \left( \frac{\sigma_2^2}{\alpha \sigma_2^2 + (1 - \alpha) \sigma_1^2} \right)^b$$

$$= \log \left( \frac{\sigma_2}{\sigma_1} \right)^b + \frac{1}{2(\alpha - 1)} \log \left( \frac{\frac{\sigma_2^2}{\sigma_1^2}}{\alpha \frac{\sigma_2^2}{\sigma_1^2} + 1 - \alpha} \right)^b$$

Substituting $\sigma_1^2 = \frac{\|\gamma_t(D)\|_2^2}{b} + m\sigma_g^2$ and $\sigma_2^2 = \frac{\|\gamma_t(D')\|_2^2}{b} + m\sigma_g^2$ into the above equation, we can obtain that

$$D_\alpha \left( \mathcal{SG} \left( \gamma_t(D); R_t, \xi_t \right) \| \mathcal{SG} \left( \gamma_t(D'); R_t, \xi_t \right) \right)$$

$$= D_\alpha \left( \mathcal{N} \left( 0, \frac{\|\gamma_t(D)\|_2^2}{b} + m\sigma_g^2 \right) \| \mathcal{N} \left( 0, \frac{\|\gamma_t(D')\|_2^2}{b} + m\sigma_g^2 \right) \right)$$

$$= \log \left( \frac{\sqrt{\frac{\|\gamma_t(D')\|_2^2}{b} + m\sigma_g^2}}{\sqrt{\frac{\|\gamma_t(D)\|_2^2}{b} + m\sigma_g^2}} \right)^b + \frac{1}{2(\alpha - 1)} \log \left( \frac{\frac{\frac{\|\gamma_t(D')\|_2^2}{b} + m\sigma_g^2}{\frac{\|\gamma_t(D)\|_2^2}{b} + m\sigma_g^2}}{\alpha \frac{\frac{\|\gamma_t(D')\|_2^2}{b} + m\sigma_g^2}{\frac{\|\gamma_t(D)\|_2^2}{b} + m\sigma_g^2} + 1 - \alpha} \right)^b$$

$$= b \log \sqrt{\frac{\|\gamma_t(D')\|^2 + mb\sigma_g^2}{\|\gamma_t(D)\|^2 + mb\sigma_g^2}} + \frac{b}{2(\alpha - 1)} \log \frac{\|\gamma_t(D')\|^2 + mb\sigma_g^2}{\|\gamma_t(D)\|^2 + mb\sigma_g^2}$$

$$= b f_\alpha \left( \sqrt{\frac{\|\gamma_t(D')\|^2 + mb\sigma_g^2}{\|\gamma_t(D)\|^2 + mb\sigma_g^2}} \right)$$

with

$$f_\alpha(x) = \log x + \frac{1}{2(\alpha - 1)} \log \frac{x^2}{\alpha x^2 + 1 - \alpha}.$$

$\square$

**Definition C.3** (Ratio Sensitivity). For any constant $c \geq 0$, define the ratio sensitivity of $\theta$ as

$$\text{rsens}_c(\theta) = \sup_{D, D'} \sqrt{\frac{\|\theta(D')\|^2 + c^2}{\|\theta(D)\|^2 + c^2}}$$

where the supremum is over all neighboring datasets $D, D'$.

Now we prove the monotonicity of $f_\alpha$ and the bound on $\text{rsens}_{\sqrt{mb}\sigma_g}(\gamma_t)$.

**Lemma C.3** (Monotonicity of $f_\alpha$). For any $(x, \alpha)$ such that $f_\alpha$ is well-defined, $f_\alpha(x)$ is monotonically decreasing with respect to $x$ for $x \leq 1$ and increasing for $x \geq 1$;

*Proof.* In fact,

$$f_\alpha'(x) = \left( \log x + \frac{1}{2(\alpha - 1)} \log \frac{x^2}{\alpha x^2 + 1 - \alpha} \right)'$$

$$= (\log x)' + \frac{1}{2(\alpha - 1)} \left( \log \frac{x^2}{\alpha x^2 + 1 - \alpha} \right)'$$

$$= \frac{1}{x} + \frac{1}{2(\alpha - 1)} \frac{\alpha x^2 + 1 - \alpha}{x^2} \cdot \left( \frac{x^2}{\alpha x^2 + 1 - \alpha} \right)'$$

$$= \frac{1}{x} + \frac{1}{2(\alpha - 1)} \frac{\alpha x^2 + 1 - \alpha}{x^2} \cdot \frac{2x \cdot (\alpha x^2 + 1 - \alpha) - 2\alpha x \cdot x^2}{(\alpha x^2 + 1 - \alpha)^2}$$

$$= \frac{1}{x} + \frac{1}{2(\alpha - 1)} \frac{\alpha x^2 + 1 - \alpha}{x^2} \cdot \frac{2x(1 - \alpha)}{(\alpha x^2 + 1 - \alpha)^2}$$

$$= \frac{1}{x} - \frac{1}{x(\alpha x^2 + 1 - \alpha)}$$

$$= \frac{1}{x} \left( 1 - \frac{1}{\alpha x^2 + 1 - \alpha} \right)$$

$$= \frac{\alpha(x-1)(x+1)}{x(\alpha x^2 + 1 - \alpha)}$$

Therefore, $f'_\alpha \geq 0$ for $x \geq 1$ and $f'_\alpha \leq 0$ for $x \leq 1$. $\qquad\square$

**Lemma C.4** (Bound on $\mathrm{rsens}_{\sqrt{mb}\sigma_g}(\gamma_t)$)**.**

$$\sqrt{1 - \frac{2\tau^2}{b\sigma_g^2}} \leq \frac{1}{\mathrm{rsens}_{\sqrt{mb}\sigma_g}(\gamma_t)} = \inf_{D,D'} \sqrt{\frac{\|\gamma_t(D')\|^2 + mb\sigma_g^2}{\|\gamma_t(D)\|^2 + mb\sigma_g^2}} \leq 1$$

$$\leq \mathrm{rsens}_{\sqrt{mb}\sigma_g}(\gamma_t) = \sup_{D,D'} \sqrt{\frac{\|\gamma_t(D')\|^2 + mb\sigma_g^2}{\|\gamma_t(D)\|^2 + mb\sigma_g^2}} \leq \sqrt{1 + \frac{2\tau^2}{b\sigma_g^2}}$$

*Proof.* According to Definition C.3,

$$\sup_{D,D'} \sqrt{\frac{\|\gamma_t(D')\|^2 + mb\sigma_g^2}{\|\gamma_t(D)\|^2 + mb\sigma_g^2}} = \mathrm{rsens}_{\sqrt{mb}\sigma_g}(\gamma_t) \geq 1;$$

$$\inf_{D,D'} \sqrt{\frac{\|\gamma_t(D')\|^2 + mb\sigma_g^2}{\|\gamma_t(D)\|^2 + mb\sigma_g^2}} = \frac{1}{\sup_{D,D'} \sqrt{\frac{\|\gamma_t(D')\|^2 + mb\sigma_g^2}{\|\gamma_t(D)\|^2 + mb\sigma_g^2}}} = \frac{1}{\mathrm{rsens}_{\sqrt{mb}\sigma_g}(\gamma_t)} \leq 1.$$

In addition,

$$\mathrm{rsens}_{\sqrt{mb}\sigma_g}(\gamma_t) = \sup_{D,D'} \sqrt{\frac{\|\gamma_t(D')\|^2 + mb\sigma_g^2}{\|\gamma_t(D)\|^2 + mb\sigma_g^2}}$$

$$\leq \sup_{D,D'} \sqrt{1 + \frac{\left| \|\gamma_t(D)\|^2 - \|\gamma_t(D')\|^2 \right|}{\|\gamma_t(D)\|^2 + mb\sigma_g^2}}$$

$$= \sup_{D,D'} \sqrt{1 + \frac{(\|\gamma_t(D)\| + \|\gamma_t(D')\|) \left| \|\gamma_t(D)\| - \|\gamma_t(D')\| \right|}{\|\gamma_t(D)\|^2 + mb\sigma_g^2}}$$

$$\leq \sup_{D,D'} \sqrt{1 + \frac{(\|\gamma_t(D)\| + \|\gamma_t(D')\|) \|\gamma_t(D) - \gamma_t(D')\|}{\|\gamma_t(D)\|^2 + mb\sigma_g^2}}$$

$$\leq \sqrt{1 + \frac{(m\tau + m\tau) \cdot \tau}{mb\sigma_g^2}} = \sqrt{1 + \frac{2\tau^2}{b\sigma_g^2}}$$

and we can get the lower bound

$$\frac{1}{\text{rsens}_{\sqrt{mb}\sigma_g}(\gamma_t)} = \inf_{D,D'} \sqrt{\frac{\|\gamma_t(D')\|^2 + mb\sigma_g^2}{\|\gamma_t(D)\|^2 + mb\sigma_g^2}} \geq \sqrt{1 - \frac{2\tau^2}{b\sigma_g^2}}$$

with a similar calculation. □

Now we can obtain the upper bound of $D_\alpha\left(\mathcal{SG}(\gamma_t(D)); R_t, \xi_t \| \mathcal{SG}(\gamma_t(D')); R_t, \xi_t\right)$.

**Lemma C.5** (Lemma 2.2). For any neighboring datasets $D, D'$,

$$D_\alpha\left(\mathcal{SG}(\gamma_t(D)); R_t, \xi_t \| \mathcal{SG}(\gamma_t(D')); R_t, \xi_t\right)$$

$$\leq b \max\left\{ f_\alpha\left(\sqrt{1 + \frac{2\tau^2}{b\sigma_g^2}}\right), f_\alpha\left(\sqrt{1 - \frac{2\tau^2}{b\sigma_g^2}}\right) \right\} \leq \frac{\alpha^2 \tau^4}{(\alpha - 1) b\sigma_g^4}$$

*Proof.* According to Lemma C.2,

$$D_\alpha\left(\mathcal{SG}(\gamma_t(D); R_t, \xi_t) \| \mathcal{SG}(\gamma_t(D'); R_t, \xi_t)\right) = b f_\alpha\left(\sqrt{\frac{\|\gamma_t(D')\|^2 + mb\sigma_g^2}{\|\gamma_t(D)\|^2 + mb\sigma_g^2}}\right)$$

From Lemma C.4, we have

$$\sqrt{1 - \frac{2\tau^2}{b\sigma_g^2}} \leq \inf_{D,D'} \sqrt{\frac{\|\gamma_t(D')\|^2 + mb\sigma_g^2}{\|\gamma_t(D)\|^2 + mb\sigma_g^2}} \leq \sqrt{\frac{\|\gamma_t(D')\|^2 + mb\sigma_g^2}{\|\gamma_t(D)\|^2 + mb\sigma_g^2}} \leq \sup_{D,D'} \sqrt{\frac{\|\gamma_t(D')\|^2 + mb\sigma_g^2}{\|\gamma_t(D)\|^2 + mb\sigma_g^2}} \leq \sqrt{1 + \frac{2\tau^2}{b\sigma_g^2}}.$$

Based on the monotonicity of $f_\alpha$ in Lemma C.3,

$$D_\alpha\left(\mathcal{SG}(\gamma_t(D); R_t, \xi_t) \| \mathcal{SG}(\gamma_t(D'); R_t, \xi_t)\right) = b f_\alpha\left(\sqrt{\frac{\|\gamma_t(D')\|^2 + mb\sigma_g^2}{\|\gamma_t(D)\|^2 + mb\sigma_g^2}}\right)$$

$$\leq b \max\left\{ f_\alpha\left(\sqrt{1 + \frac{2\tau^2}{b\sigma_g^2}}\right), f_\alpha\left(\sqrt{1 - \frac{2\tau^2}{b\sigma_g^2}}\right) \right\}$$

In addition,

$$f_\alpha\left(\sqrt{1 + \frac{2\tau^2}{b\sigma_g^2}}\right) = \frac{1}{2}\log\left(1 + \frac{2\tau^2}{b\sigma_g^2}\right) + \frac{1}{2(\alpha - 1)}\log\frac{1 + \frac{2\tau^2}{b\sigma_g^2}}{\alpha \cdot \left(1 + \frac{2\tau^2}{b\sigma_g^2}\right) + 1 - \alpha}$$

$$= \frac{1}{2}\log\left(1 + \frac{2\tau^2}{b\sigma_g^2}\right) + \frac{1}{2(\alpha - 1)}\log\frac{1 + \frac{2\tau^2}{b\sigma_g^2}}{1 + \frac{2\alpha\tau^2}{b\sigma_g^2}}$$

$$= \frac{1}{2}\log\left(1 + \frac{2\tau^2}{b\sigma_g^2}\right) + \frac{1}{2(\alpha - 1)}\left(\log\left(1 + \frac{2\tau^2}{b\sigma_g^2}\right) - \log\left(1 + \frac{2\alpha\tau^2}{b\sigma_g^2}\right)\right)$$

$$= \frac{1}{2(\alpha - 1)}\left(\alpha\log\left(1 + \frac{2\tau^2}{b\sigma_g^2}\right) - \log\left(1 + \frac{2\alpha\tau^2}{b\sigma_g^2}\right)\right)$$

$$\leq \frac{1}{2(\alpha - 1)}\left(\alpha \cdot \frac{2\tau^2}{b\sigma_g^2} - \left(\frac{2\alpha\tau^2}{b\sigma_g^2} - \frac{1}{2}\left(\frac{2\alpha\tau^2}{b\sigma_g^2}\right)^2\right)\right)$$

$$= \frac{\alpha^2\tau^4}{(\alpha - 1)b^2\sigma_g^4}$$

And similarly we can also get that $f_\alpha\left(\sqrt{1 - \frac{2\tau^2}{b\sigma_g^2}}\right) \leq \frac{\alpha^2\tau^4}{(\alpha - 1)b^2\sigma_g^4}$. Therefore,

$$D_\alpha\left(\mathcal{SG}(\gamma_t(D); R_t, \xi_t) \| \mathcal{SG}(\gamma_t(D'); R_t, \xi_t)\right)$$

$$\leq b \max \left\{ f_\alpha \left( \sqrt{1 + \frac{2\tau^2}{b\sigma_g^2}} \right), f_\alpha \left( \sqrt{1 - \frac{2\tau^2}{b\sigma_g^2}} \right) \right\} \leq \frac{\alpha^2 \tau^4}{(\alpha - 1) b\sigma_g^4}.$$

$\square$

**Definition C.4** (($\alpha, \epsilon$)-RDP [46]). A randomized mechanism $f : \mathcal{D} \to \mathcal{R}$ is said to have $\epsilon$-Rényi differential privacy of order $\alpha$, or ($\alpha, \epsilon$)-RDP for short, if for any adjacent $D, D' \in \mathcal{D}$, it holds that

$$D_\alpha \left( f(D) \| f(D') \right) \leq \epsilon.$$

And RDP can be transformed into the standard ($\epsilon, \delta$)-DP.

**Lemma C.6** (Relationship with ($\epsilon, \delta$)-DP [46]). If f is an ($\alpha, \epsilon$)-RDP mechanism, it also satisfies $\left( \epsilon + \frac{\log 1/\delta}{\alpha - 1}, \delta \right)$-differential privacy for any $0 < \delta < 1$.

So we immediately have the RDP and DP result of SGM from Lemma C.5 and Lemma C.6.

**Lemma C.7.** SGM on $\gamma_t$ is $(\alpha, \frac{\alpha^2 \tau^4}{(\alpha - 1) b\sigma_g^4})$-RDP, therefore $(\frac{\alpha^2 \tau^4}{(\alpha - 1) b\sigma_g^4} + \frac{\log(1/\delta)}{\alpha - 1}, \delta)$-DP.

Finally we can prove Theorem 2.2.

**Theorem C.8.** There exists constants $c_3$ and $c_4$ so that given the sampling probability $q = \frac{m}{n}$ and the number of steps $T$, for any $\epsilon_p \leq c_3 q \sqrt{T}$, Algorithm 1 is ($\epsilon_p, \delta_p$)-differentially private for any $\delta_p > 0$ if we choose

$$\sigma_g^2 \geq \frac{c_4 q \tau^2 \sqrt{T} \log(2qT/\delta_p)}{\sqrt{b}\epsilon_p}. \tag{6}$$

*Proof.* According to Lemma 2.4, $\mathcal{SG}(\gamma_t; R_t; \xi_t)$ is $\left( \frac{\alpha^2 \tau^4}{(\alpha - 1) b\sigma_g^4} + \frac{\log(1/\delta_0)}{\alpha - 1}, \delta_0 \right)$-DP. By taking the derivative, we can get the optimal choice $\alpha = 1 + \sqrt{1 + \frac{b\sigma_g^4 \log(1/\delta_0)}{\tau^4}}$, we can get that

$$\frac{\alpha^2 \tau^4}{(\alpha - 1) b\sigma_g^4} + \frac{\log(1/\delta_0)}{\alpha - 1} = \frac{2\tau^4}{b\sigma_g^4} \left( 1 + \sqrt{1 + \frac{b\sigma_g^4 \log(1/\delta_0)}{\tau^4}} \right)$$

in the case when $b \geq c_0 \max \left\{ \frac{\tau^4}{\sigma_g^4 \log(1/\delta_0)}, \frac{\tau^4 \log(1/\delta_0)}{\sigma_g^4} \right\}$, then $\frac{b\sigma_g^4 \log(1/\delta_0)}{\tau^4} \geq c_0$, we have that

$$\frac{2\tau^4}{b\sigma_g^4} \left( 1 + \sqrt{1 + \frac{b\sigma_g^4 \log(1/\delta_0)}{\tau^4}} \right) \leq \frac{2\tau^4}{b\sigma_g^4} \left( \sqrt{\frac{1}{c_0} \frac{b\sigma_g^4 \log(1/\delta_0)}{\tau^4}} + \sqrt{\frac{1}{c_0} \frac{b\sigma_g^4 \log(1/\delta_0)}{\tau^4} + \frac{b\sigma_g^4 \log(1/\delta_0)}{\tau^4}} \right)$$

$$= \frac{2 \left( 1 + \sqrt{c_0 + 1} \right) \tau^2 \sqrt{\log(1/\delta_0)}}{\sqrt{c_0}} \frac{\tau^2 \sqrt{\log(1/\delta_0)}}{\sqrt{b}\sigma_g^2} := c_1 \frac{\tau^2 \sqrt{\log(1/\delta_0)}}{\sqrt{b}\sigma_g^2}$$

so we get that $M$ is ($\epsilon_0, \delta_0$)-DP with $\epsilon_0 = c_1 \frac{\tau^2 \sqrt{\log(1/\delta_0)}}{\sqrt{b}\sigma_g^2}$.

Now we list the subsampling and composition properties of ($\epsilon, \delta$)-DP.

**Lemma C.9** (Sub-sampling of ($\epsilon, \delta$)-DP). If $M$ is ($\epsilon, \delta$)-DP, then $M' = M \circ \text{Sample}_m$ obeys ($\epsilon', \delta'$)-DP with $\epsilon' = \log(1 + p(e^\epsilon - 1))$ and $\delta' = p\delta$, in which $p = \frac{m}{n}$ is the sampling ratio.

**Lemma C.10** (Strong composition of ($\epsilon, \delta$)-DP). For all $\epsilon, \delta, \delta' \geq 0$, the class of ($\epsilon, \delta$)-differentially private mechanisms satisfies ($\epsilon', k\delta + \delta'$)-differential privacy under $k$-fold adaptive composition for:

$$\epsilon' = \sqrt{2k \log(1/\delta')}\epsilon + k\epsilon(e^\epsilon - 1)$$

Since $\epsilon_0 = c_1 \frac{\tau^2 \sqrt{\log(1/\delta_0)}}{\sqrt{b}\sigma_g^2} \leq \frac{c_1}{\sqrt{c_0}}$, so according to Lemma C.9, $M \circ \text{Sample}_m$ satisfies ($\epsilon_1, p\delta_0$)-DP with

$$\epsilon_1 = \log(1 + p(e^{\epsilon_0} - 1)) \leq c_2 p \epsilon_0 = c_1 c_2 \frac{p\tau^2 \sqrt{\log(1/\delta_0)}}{\sqrt{b}\sigma_g^2}$$

According to Lemma C.10, we can see that $T$-fold composition of $M \circ \texttt{Sample}_m$ satisfies $(\epsilon_2, pT\delta_0 + \delta')$-DP with

$$
\begin{aligned}
\epsilon_2 &= \sqrt{2T\log(1/\delta')}\epsilon_1 + T\epsilon_1\left(e^{\epsilon_1} - 1\right) \\
&\leq \sqrt{2T\log(1/\delta')}\epsilon_1 + c_3 T\epsilon_1^2 \\
&\leq \sqrt{2T\log(1/\delta')} \cdot c_1 c_2 \frac{p\tau^2\sqrt{\log(1/\delta_0)}}{\sqrt{b}\sigma_g^2} + c_3 T \cdot \left(c_1 c_2 \frac{p\tau^2\sqrt{\log(1/\delta_0)}}{\sqrt{b}\sigma_g^2}\right)^2 \\
&= \frac{c_1 c_2\sqrt{2T\log(1/\delta')\log(1/\delta_0)}p\tau^2}{\sqrt{b}\sigma_g^2} + c_1^2 c_2^2 c_3 \frac{Tp^2\tau^4\log(1/\delta_0)}{b\sigma_g^4} \\
&\leq \frac{c_1 c_2 p\sqrt{2T\log(1/\delta')}}{\sqrt{c_0}} + \frac{c_1^2 c_2^2 c_3 p^2 T}{c_0}
\end{aligned}
$$

By setting $\epsilon_2 = \epsilon$, and choosing $\delta_0 = \frac{\delta}{2pT}$, $\delta' = \frac{\delta}{2}$,

$$
\sigma_g^2 = \frac{C'\sqrt{T}p\tau^2\log(2pT/\delta)}{\sqrt{b}\epsilon},
$$

then we can obtain $(\epsilon, \delta)$-DP. $\qquad\square$

# D  Optimization Analysis with GD as `GLOBAL_OPT`

First we write down Algorithm 3, which will be called as `GLOBAL_OPT` each round to do a one-step GD update of the global parameter $\theta_{t-1}$ with desketched aggregated updates $\text{desk}\left(\bar{\bar{\Delta}}_{t-1}\right)$.

---
**Algorithm 3** `GLOBAL_OPT` (GD)

---
**Inputs:** model $\theta_{t-1}$, desketched update $\text{desk}\left(\bar{\bar{\Delta}}_{t-1}\right)$.
**Output:** model $\theta_T$.
**One-step GD:** $\theta_t \leftarrow \theta_{t-1} - \eta_{\text{global}} \cdot \text{desk}(\bar{\bar{\Delta}}_{t-1})$.

---

Then we will introduce the optimization result of Algorithm 2 with Algorithm 3. We follow a similar analysis to [59], but we exploit the second order structure of the deep learning losses, which helps avoid picking up dimension dependence due to the sketching operation. We will state the formal result.

**Theorem D.1.** Suppose $\{\theta_t\}_{t=0}^{T}$ is generated by Algorithm 2 with Algorithm 3 as `GLOBAL_OPT`. Denote $\mathcal{L}^*$ the minimum of the average empirical loss. Under Assumption 1-4, with learning rate $\eta = \eta_{\text{global}}\eta_{\text{local}}$, we have that with probability at least $1 - 10\delta$,

$$
\frac{1}{T}\sum_{t=1}^{T}\|\nabla\mathcal{L}(\theta_t)\|_2^2 \leq \frac{\mathcal{L}(\theta_0)-\mathcal{L}^*}{\eta KT} + \frac{2\log(2T/\delta)G^2}{\sqrt{NT}} + \frac{\eta_{\text{local}}LKG^2}{2} + \frac{\sqrt{2}G\log(2T/\delta)\sigma_s}{\sqrt{NTK}}
$$

$$
+ \max\left\{0, \frac{G(KG-\tau)}{K}\right\} + \frac{\sqrt{2}\log^2(NTd/\delta)G\tau_{K,G}}{\sqrt{b\overline{T}}K} + \frac{\log^2(2T/\delta)G^2}{\eta TK}
$$

$$
+ \frac{2\eta\log^2(2T/\delta)\sigma_g^2}{NK} + \frac{2\eta\alpha_1^2\mathcal{I}L\tau_{K,G}^2}{K} + \frac{2\eta\sigma_g^2\mathcal{I}L\log^2(2dT/\delta)}{NK}
$$

in which

$$
\alpha_1 = 1 + \frac{\log^{1.5}(NTd^2/\delta)}{\sqrt{b}}, \tau_{K,G} = \min\{\tau, KG\}
$$

**Remark D.1.** To analyze the bound in terms of $\tau$, we consider all other parameters as fixed and distinguish between two principal regimes:

1. When $\tau \leq KG$: In this regime, clipping may be activated, and $\tau$ acts as an upper bound on the norm of each clipped update $\Delta_{c,t}$ according to our algorithm. The total bound in this case is given by $\frac{\mathcal{L}(\theta_0)-\mathcal{L}^*}{\eta KT} + \frac{2\log(2T/\delta)G^2}{\sqrt{NT}} + \frac{\eta_{\text{local}}LKG^2}{2} + \frac{\sqrt{2}G\log(2T/\delta)\sigma_s}{\sqrt{NTK}} + \max\left\{0, \frac{G(KG-\tau)}{K}\right\} + \frac{\sqrt{2}\log^2(NTd/\delta)G\tau}{\sqrt{b\overline{T}}K} + \frac{\log^2(2T/\delta)G^2}{\eta TK} + \frac{2\eta\log^2(2T/\delta)\sigma_g^2}{NK} + \frac{2\eta\alpha_1^2\mathcal{I}L\tau^2}{K} + \frac{2\eta\sigma_g^2\mathcal{I}L\log^2(2dT/\delta)}{NK}$. We can observe that $\frac{\sqrt{2}\log^2(NTd/\delta)G\tau}{\sqrt{b\overline{T}}K} + \frac{2\eta\alpha_1^2\mathcal{I}L\tau^2}{K}$ in the optimization bound of the sketching-based algorithm is monotonically increasing in $\tau$, while $\max\left\{0, \frac{G(KG-\tau)}{K}\right\}$ caused by clipping is monotonically decreasing in $\tau$. This trade-off highlights the need for a careful choice of $\tau$, as overly aggressive clipping introduces clipping bias, while too loose a threshold amplifies optimization error of sketching algorithms.

2. When $\tau \geq KG$: In this regime, the clipping operation becomes inactive under our algorithm, as all updates fall within the clipping threshold $\tau$. $KG$ effectively bounds the norm of each update $\Delta_{c,t}$, and the clipping no longer influences the computation. The total bound becomes $\frac{\mathcal{L}(\theta_0)-\mathcal{L}^*}{\eta KT} + \frac{2\log(2T/\delta)G^2}{\sqrt{NT}} + \frac{\eta_{\text{local}}LKG^2}{2} + \frac{\sqrt{2}G\log(2T/\delta)\sigma_s}{\sqrt{NTK}} + \frac{\sqrt{2}\log^2(NTd/\delta)G^2}{\sqrt{b\overline{T}}} + \frac{\log^2(2T/\delta)G^2}{\eta TK} + \frac{2\eta\log^2(2T/\delta)\sigma_g^2}{NK} + 2\eta\alpha_1^2\mathcal{I}LKG^2 + \frac{2\eta\sigma_g^2\mathcal{I}L\log^2(2dT/\delta)}{NK}$. Consequently, the entire bound becomes independent of $\tau$ and remains constant as $\tau$ increases further.

*Proof.* According to the algorithm, we can write the update in the sync step as:

$$
\theta_{t+1} - \theta_t = -\eta_{\text{global}}\,\text{desk}\left(\bar{\bar{\Delta}}_t\right)
$$

$$= -\eta_{\text{global}} \text{ desk} \left( \frac{1}{N} \sum_{c \in \mathcal{C}_t} \tilde{\Delta}_{c,t} \right)$$

$$= -\eta_{\text{global}} \text{ desk} \left( \eta_{\text{local}} \frac{1}{N} \sum_{c \in \mathcal{C}_t} \left( \text{sk} \left( \text{clip} \left( \frac{\Delta_{c,t}}{\eta_{\text{local}}}, \tau \right) \right) + \mathbf{z}_{c,t} \right) \right)$$

$$= -\eta R_t^\top \left( \frac{1}{N} \sum_{i \in \mathcal{C}_t} \left[ R_t \left( \text{clip} \left( \sum_{k=1}^{K} g_{c,t,k}, \tau \right) \right) + \mathbf{z}_{c,t} \right] \right)$$

$$= -\frac{\eta}{N} \sum_{c \in \mathcal{C}_t} R_t^\top R_t \text{clip} \left( \sum_{k=1}^{K} g_{c,t,k}, \tau \right) - \frac{\eta}{N} \sum_{c \in \mathcal{C}_t} R_t^\top \mathbf{z}_{c,t}$$

in which $\eta = \eta_{\text{global}}\eta_{\text{local}}$. By Taylor expansion, we have

$$\mathcal{L}(\theta_{t+1}) = \mathcal{L}(\theta_t) + \nabla \mathcal{L}(\theta_t)^\top (\theta_{t+1} - \theta_t) + \frac{1}{2}(\theta_{t+1} - \theta_t)^\top \hat{H}_{\mathcal{L},t}(\theta_{t+1} - \theta_t)$$

By taking summation from 0 to $T - 1$, we can get that

$$\mathcal{L}(\theta_T) - \mathcal{L}(\theta_0) = \sum_{t=1}^{T} (\mathcal{L}(\theta_{t+1}) - \mathcal{L}(\theta_t)) = \underbrace{\sum_{t=1}^{T} \nabla \mathcal{L}(\theta_t)^\top (\theta_{t+1} - \theta_t)}_{T_1} + \underbrace{\frac{1}{2} \sum_{t=1}^{T} (\theta_{t+1} - \theta_t)^\top \hat{H}_{\mathcal{L},t}(\theta_{t+1} - \theta_t)}_{T_2}$$

(7)

### D.1  Bounding $T_1$

For each term in $T_1$, we have

$$\nabla \mathcal{L}(\theta_t)^\top (\theta_{t+1} - \theta_t)$$

$$= - \nabla \mathcal{L}(\theta_t)^\top \left( \frac{\eta}{N} \sum_{c \in \mathcal{C}_t} R_t^\top R_t \text{clip} \left( \sum_{k=1}^{K} g_{c,t,k}, \tau \right) + \frac{\eta}{N} \sum_{c \in \mathcal{C}_t} R_t^\top \mathbf{z}_{c,t} \right)$$

$$= - \frac{\eta}{N} \nabla \mathcal{L}(\theta_t)^\top \sum_{c \in \mathcal{C}_t} R_t^\top R_t \text{clip} \left( \sum_{k=1}^{K} g_{c,t,k}, \tau \right) - \frac{\eta}{N} \nabla \mathcal{L}(\theta_t)^\top \sum_{c \in \mathcal{C}_t} R_t^\top \mathbf{z}_{c,t}$$

By taking summation from 0 to $T - 1$, we can get that

$$\sum_{t=1}^{T} \nabla \mathcal{L}(\theta_t)^\top (\theta_{t+1} - \theta_t) = \underbrace{-\eta \sum_{t=1}^{T} \frac{1}{N} \nabla \mathcal{L}(\theta_t)^\top \sum_{c \in \mathcal{C}_t} R_t^\top R_t \text{clip} \left( \sum_{k=1}^{K} g_{c,t,k}, \tau \right)}_{S_1} \underbrace{-\eta \sum_{t=1}^{T} \frac{1}{N} \nabla \mathcal{L}(\theta_t)^\top \sum_{c \in \mathcal{C}_t} R_t^\top \mathbf{z}_{c,t}}_{S_2}$$

(8)

### D.1.1  Bounding $S_1$

For each term in $S_1$, we have that

$$\frac{1}{N} \nabla \mathcal{L}(\theta_t)^\top \sum_{c \in \mathcal{C}_t} R_t^\top R_t \text{clip} \left( \sum_{k=1}^{K} g_{c,t,k}, \tau \right)$$

$$= K \left\langle \nabla \mathcal{L}(\theta_t), \frac{1}{C} \sum_{c=1}^{C} \nabla \mathcal{L}_c(\theta_t) \right\rangle + K \left\langle \nabla \mathcal{L}(\theta_t), \frac{1}{N} \sum_{c \in \mathcal{C}_t} \nabla \mathcal{L}_c(\theta_t) - \frac{1}{C} \sum_{c=1}^{C} \nabla \mathcal{L}_c(\theta_t) \right\rangle$$

$$+ \left\langle \nabla \mathcal{L}(\theta_t), \frac{1}{N} \sum_{c \in \mathcal{C}_t} \sum_{k=1}^{K} (\nabla \mathcal{L}_c(\theta_{c,t,k}) - \nabla \mathcal{L}_c(\theta_t)) \right\rangle + \left\langle \nabla \mathcal{L}(\theta_t), \frac{1}{N} \sum_{c \in \mathcal{C}_t} \sum_{k=1}^{K} (g_{c,t,k} - \nabla \mathcal{L}_c(\theta_{c,t,k})) \right\rangle$$

$$+ \left\langle \nabla \mathcal{L}(\theta_t), \frac{1}{N} \sum_{c \in \mathcal{C}_t} \left( \text{clip} \left( \sum_{k=1}^{K} g_{c,t,k}, \tau \right) - \sum_{k=1}^{K} g_{c,t,k} \right) \right\rangle$$

$$+ \left\langle \nabla \mathcal{L}(\theta_t), \frac{1}{N} \sum_{c \in \mathcal{C}_t} \left( R_t^\top R_t \text{clip}\left( \sum_{k=1}^{K} g_{c,t,k}, \tau \right) - \text{clip}\left( \sum_{k=1}^{K} g_{c,t,k}, \tau \right) \right) \right\rangle$$

By taking summation from $0$ to $T-1$, we can get that

$$\sum_{t=1}^{T} \frac{1}{N} \nabla \mathcal{L}(\theta_t)^\top \sum_{c \in \mathcal{C}_t} R_t^\top R_t \text{clip}\left( \sum_{k=1}^{K} g_{c,t,k}, \tau \right)$$

$$= K \underbrace{\sum_{t=1}^{T} \left\langle \nabla \mathcal{L}(\theta_t), \frac{1}{C} \sum_{c=1}^{C} \nabla \mathcal{L}_c(\theta_t) \right\rangle}_{Y_1} + K \underbrace{\sum_{t=1}^{T} \left\langle \nabla \mathcal{L}(\theta_t), \frac{1}{N} \sum_{c \in \mathcal{C}_t} \nabla \mathcal{L}_c(\theta_t) - \frac{1}{C} \sum_{c=1}^{C} \nabla \mathcal{L}_c(\theta_t) \right\rangle}_{Y_2}$$

$$+ \underbrace{\sum_{t=1}^{T} \left\langle \nabla \mathcal{L}(\theta_t), \frac{1}{N} \sum_{c \in \mathcal{C}_t} \sum_{k=1}^{K} (\nabla \mathcal{L}_c(\theta_{c,t,k}) - \nabla \mathcal{L}_c(\theta_t)) \right\rangle}_{Y_3} + \underbrace{\sum_{t=1}^{T} \left\langle \nabla \mathcal{L}(\theta_t), \frac{1}{N} \sum_{c \in \mathcal{C}_t} \sum_{k=1}^{K} (g_{c,t,k} - \nabla \mathcal{L}_c(\theta_{c,t,k})) \right\rangle}_{Y_4}$$

$$+ \underbrace{\sum_{t=1}^{T} \left\langle \nabla \mathcal{L}(\theta_t), \frac{1}{N} \sum_{c \in \mathcal{C}_t} \left( \text{clip}\left( \sum_{k=1}^{K} g_{c,t,k}, \tau \right) - \sum_{k=1}^{K} g_{c,t,k} \right) \right\rangle}_{Y_5}$$

$$+ \underbrace{\sum_{t=1}^{T} \left\langle \nabla \mathcal{L}(\theta_t), \frac{1}{N} \sum_{c \in \mathcal{C}_t} \left( R_t^\top R_t \text{clip}\left( \sum_{k=1}^{K} g_{c,t,k}, \tau \right) - \text{clip}\left( \sum_{k=1}^{K} g_{c,t,k}, \tau \right) \right) \right\rangle}_{Y_6} \tag{9}$$

**D.1.1.1 Bounding $Y_1$**

According to the definition,

$$\left\langle \nabla \mathcal{L}(\theta_t), \frac{1}{C} \sum_{c=1}^{C} \nabla \mathcal{L}_c(\theta_t) \right\rangle = \|\nabla \mathcal{L}(\theta_t)\|_2^2$$

so

$$\sum_{t=1}^{T} \left\langle \nabla \mathcal{L}(\theta_t), \frac{1}{C} \sum_{c=1}^{C} \nabla \mathcal{L}_c(\theta_t) \right\rangle = \sum_{t=1}^{T} \|\nabla \mathcal{L}(\theta_t)\|_2^2 \tag{10}$$

**D.1.1.2 Bounding $Y_2$**

We first bound each term with a fixed $t \in [T]$ in $Y_2$. According to the assumption, each $c \in \mathcal{C}_t$ is uniformly randomly selected from $[C]$, so by Hoeffding's inequality, we have

$$\mathbb{P}\left( \left| \left\langle \nabla \mathcal{L}(\theta_t), \frac{1}{N} \sum_{c \in \mathcal{C}_t} \nabla \mathcal{L}_c(\theta_t) - \frac{1}{C} \sum_{c=1}^{C} \nabla \mathcal{L}_c(\theta_t) \right\rangle \right| \geq a \right)$$

$$\leq 2 \exp\left( -\frac{2Na^2}{(2G^2)^2} \right) = 2 \exp\left( -\frac{Na^2}{2G^4} \right)$$

By selecting $a = \frac{\sqrt{2\log(2T/\delta)}G^2}{\sqrt{N}}$, we have that with probability at least $1 - \frac{\delta}{T}$,

$$\left| \left\langle \nabla \mathcal{L}(\theta_t), \frac{1}{N} \sum_{c \in \mathcal{C}_t} \nabla \mathcal{L}_c(\theta_t) - \frac{1}{C} \sum_{c=1}^{C} \nabla \mathcal{L}_c(\theta_t) \right\rangle \right| \leq \frac{\sqrt{2\log(2T/\delta)}G^2}{\sqrt{N}}$$

Then denote $Z_t = \sum_{\tau'=0}^{t} \left\langle \nabla \mathcal{L}(\theta_{\tau'}), \frac{1}{N} \sum_{c \in \mathcal{C}_{\tau'}} \nabla \mathcal{L}_c(\theta_{\tau'}) - \frac{1}{C} \sum_{c=1}^{C} \nabla \mathcal{L}_c(\theta_{\tau'}) \right\rangle$, we can see that $Z_t$ is a martingale with respect to the selection each round, and from the above analysis, we have that

with probability at least $1 - \delta$, for all $t \in [T]$,

$$|Z_t - Z_{t-1}| = \left| \left\langle \nabla \mathcal{L}(\theta_t), \frac{1}{N} \sum_{c \in \mathcal{C}_t} \nabla \mathcal{L}_c(\theta_t) - \frac{1}{C} \sum_{c=1}^{C} \nabla \mathcal{L}_c(\theta_t) \right\rangle \right| \le \frac{\sqrt{2 \log(2T/\delta)} G^2}{\sqrt{N}}$$

Then by Azuma's inequality, we have

$$\mathbb{P}\left(Z_{T-1} \le -a\right) \le \exp\left( -\frac{a^2}{2 \cdot T \cdot \left( \frac{\sqrt{2 \log(2T/\delta)} G^2}{\sqrt{N}} \right)^2} \right) = \exp\left( -\frac{N a^2}{4T \log(2T/\delta) G^4} \right)$$

By selecting $a = \frac{2\sqrt{T} \log(2T/\delta) G^2}{\sqrt{N}}$, we can get that with probability at least $1 - 2\delta$,

$$Z_{T-1} = \sum_{t=1}^{T} \left\langle \nabla \mathcal{L}(\theta_t), \frac{1}{N} \sum_{c \in \mathcal{C}_t} \nabla \mathcal{L}_c(\theta_t) - \frac{1}{C} \sum_{c=1}^{C} \nabla \mathcal{L}_c(\theta_t) \right\rangle \ge -\frac{2\sqrt{T} \log(2T/\delta) G^2}{\sqrt{N}} \qquad (11)$$

**D.1.1.3 Bounding $Y_3$**

For each term, we have

$$\left\langle \nabla \mathcal{L}(\theta_t), \frac{1}{N} \sum_{c \in \mathcal{C}_t} \sum_{k=1}^{K} \left( \nabla \mathcal{L}_c(\theta_{c,t,k}) - \nabla \mathcal{L}_c(\theta_t) \right) \right\rangle$$

$$= \frac{1}{N} \sum_{c \in \mathcal{C}_t} \sum_{k=1}^{K} \left\langle \nabla \mathcal{L}(\theta_t), \hat{H}_{\mathcal{L}}^{c,t,k} \left( \theta_{c,t,k} - \theta_t \right) \right\rangle$$

$$= \frac{\eta_{\text{local}}}{N} \sum_{c \in \mathcal{C}_t} \sum_{k=1}^{K} \left\langle \nabla \mathcal{L}(\theta_t), \hat{H}_{\mathcal{L}}^{c,t,k} \sum_{\kappa=1}^{k} g_{c,t,\kappa} \right\rangle$$

$$\ge -\frac{\eta_{\text{local}}}{N} \sum_{c \in \mathcal{C}_t} \sum_{k=1}^{K} \|\nabla \mathcal{L}(\theta_t)\|_2 \cdot L \sum_{\kappa=1}^{k} \|g_{c,t,\kappa}\|_2$$

$$= -\frac{\eta_{\text{local}}}{N} \cdot N \cdot G^2 \cdot L \sum_{k=1}^{K} k$$

$$\ge -\frac{\eta_{\text{local}} L K^2 G^2}{2}$$

so

$$\sum_{t=1}^{T} \left\langle \nabla \mathcal{L}(\theta_t), \frac{1}{N} \sum_{c \in \mathcal{C}_t} \sum_{k=1}^{K} \left( \nabla \mathcal{L}_c(\theta_{c,t,k}) - \nabla \mathcal{L}_c(\theta_t) \right) \right\rangle \ge -\frac{\eta_{\text{local}} T L K^2 G^2}{2} \qquad (12)$$

**D.1.1.4 Bounding $Y_4$**

We first bound each term with a fixed $t \in [T]$ in $Y_4$.

$$\left| \left\langle \nabla \mathcal{L}(\theta_t), \frac{1}{N} \sum_{c \in \mathcal{C}_t} \sum_{k=1}^{K} \left( g_{c,t,k} - \nabla \mathcal{L}_c(\theta_{c,t,k}) \right) \right\rangle \right|$$

$$\le \frac{1}{N} \sum_{c \in \mathcal{C}_t} \sum_{k=1}^{K} \|\nabla \mathcal{L}(\theta_t)\|_2 \cdot \|g_{c,t,k} - \nabla \mathcal{L}_c(\theta_{c,t,k})\|_2$$

$$= \frac{G}{N} \sum_{c \in \mathcal{C}_t} \sum_{k=1}^{K} \|g_{c,t,k} - \nabla \mathcal{L}_c(\theta_{c,t,k})\|_2$$

According to the assumption, the stochastic noise $\|g_{c,t,k} - \nabla\mathcal{L}_c(\theta_{c,t,k})\|_2$ is a $\sigma_s$-sub-Gaussian random variable, so by Hoeffding's inequality,

$$\mathbb{P}\left(\sum_{c\in\mathcal{C}_t}\sum_{k=1}^{K}\|g_{c,t,k} - \nabla\mathcal{L}_c(\theta_{c,t,k})\|_2 \geq a\right) \leq 2\exp\left(-\frac{a^2}{NK\sigma_s^2}\right)$$

By selecting $a = \sqrt{NK\log(2T/\delta)}\sigma_s$, we have that with probability at least $1 - \frac{\delta}{T}$,

$$\sum_{c\in\mathcal{C}_t}\sum_{k=1}^{K}\|g_{c,t,k} - \nabla\mathcal{L}_c(\theta_{c,t,k})\|_2 \leq \sqrt{NK\log(2T/\delta)}\sigma_s$$

so

$$\left|\left\langle\nabla\mathcal{L}(\theta_t), \frac{1}{N}\sum_{c\in\mathcal{C}_t}\sum_{k=1}^{K}(g_{c,t,k} - \nabla\mathcal{L}_c(\theta_{c,t,k}))\right\rangle\right|$$

$$\leq \frac{G}{N}\sum_{c\in\mathcal{C}_t}\sum_{k=1}^{K}\|g_{c,t,k} - \nabla\mathcal{L}_c(\theta_{c,t,k})\|_2$$

$$\leq \frac{G}{N}\cdot\sqrt{NK\log(2T/\delta)}\sigma_s = \frac{G\sqrt{K\log(2T/\delta)}\sigma_s}{\sqrt{N}}$$

Then denote $W_t = \sum_{\tau'=0}^{t}\left\langle\nabla\mathcal{L}(\theta_\tau'), \frac{1}{N}\sum_{c\in\mathcal{C}_{\tau'}}\sum_{k=1}^{K}(g_{c,\tau',k} - \nabla\mathcal{L}_c(\theta_{c,\tau',k}))\right\rangle$, we can see that $W_t$ is a martingale with respect to the stochastic noise, and from the above analysis, we have that with probability at least $1 - \delta$, for all $t \in [T]$,

$$|W_t - W_{t-1}| = \left|\left\langle\nabla\mathcal{L}(\theta_t), \frac{1}{N}\sum_{c\in\mathcal{C}_t}\sum_{k=1}^{K}(g_{c,t,k} - \nabla\mathcal{L}_c(\theta_{c,t,k}))\right\rangle\right| \leq \frac{G\sqrt{K\log(2T/\delta)}\sigma_s}{\sqrt{N}}$$

Then by Azuma's inequality, we have

$$\mathbb{P}\left(W_{T-1} \leq -a\right) \leq \exp\left(-\frac{a^2}{2\cdot T\cdot\left(\frac{G\sqrt{K\log(2T/\delta)}\sigma_s}{\sqrt{N}}\right)^2}\right) = \exp\left(-\frac{Na^2}{2TG^2K\log(2T/\delta)\sigma_s^2}\right)$$

By selecting $a = \frac{G\sqrt{2TK}\log(2T/\delta)\sigma_s}{\sqrt{N}}$, we can get that with probability at least $1 - 2\delta$,

$$W_{T-1} = \sum_{t=1}^{T}\left\langle\nabla\mathcal{L}(\theta_t), \frac{1}{N}\sum_{c\in\mathcal{C}_t}\sum_{k=1}^{K}(g_{c,t,k} - \nabla\mathcal{L}_c(\theta_{c,t,k}))\right\rangle \geq -\frac{G\sqrt{2TK}\log(2T/\delta)\sigma_s}{\sqrt{N}} \quad (13)$$

**D.1.1.5 Bounding $Y_5$**

For each term, for $\tau \leq KG$, we have

$$\left\langle\nabla\mathcal{L}(\theta_t), \frac{1}{N}\sum_{c\in\mathcal{C}_t}\left(\text{clip}\left(\sum_{k=1}^{K}g_{c,t,k}, \tau\right) - \sum_{k=1}^{K}g_{c,t,k}\right)\right\rangle$$

$$\geq -\frac{1}{N}\sum_{c\in\mathcal{C}_t}\|\nabla\mathcal{L}(\theta_t)\|_2\left\|\sum_{c\in\mathcal{C}_t}\left(\text{clip}\left(\sum_{k=1}^{K}g_{c,t,k}, \tau\right) - \sum_{k=1}^{K}g_{c,t,k}\right)\right\|_2$$

$$\geq -G(KG - \tau)$$

for $\tau \geq KG$, we have

$$\left\langle\nabla\mathcal{L}(\theta_t), \frac{1}{N}\sum_{c\in\mathcal{C}_t}\left(\text{clip}\left(\sum_{k=1}^{K}g_{c,t,k}, \tau\right) - \sum_{k=1}^{K}g_{c,t,k}\right)\right\rangle = 0$$

so

$$\left\langle \nabla\mathcal{L}(\theta_t), \frac{1}{N}\sum_{c\in\mathcal{C}_t}\left(\text{clip}\left(\sum_{k=1}^{K}g_{c,t,k},\tau\right) - \sum_{k=1}^{K}g_{c,t,k}\right)\right\rangle \geq -\max\{0, G(KG-\tau)\}$$

By taking summation from $0$ to $T-1$, we can get that

$$\sum_{t=1}^{T}\left\langle \nabla\mathcal{L}(\theta_t), \frac{1}{N}\sum_{c\in\mathcal{C}_t}\left(\text{clip}\left(\sum_{k=1}^{K}g_{c,t,k},\tau\right) - \sum_{k=1}^{K}g_{c,t,k}\right)\right\rangle \geq -\max\{0, TG(KG-\tau)\} \quad (14)$$

**D.1.1.6 Bounding $Y_6$**

We first bound each term with a fixed $t\in[T]$ in $Y_6$. According to Lemma B.1, we have that with probability at least $1 - \frac{\delta}{T}$,

$$\left|\left\langle \nabla\mathcal{L}(\theta_t), \frac{1}{N}\sum_{c\in\mathcal{C}_t}\left(R_t^\top R_t\text{clip}\left(\sum_{k=1}^{K}g_{c,t,k},\tau\right) - \text{clip}\left(\sum_{k=1}^{K}g_{c,t,k},\tau\right)\right)\right\rangle\right|$$

$$\leq \frac{1}{N}\sum_{c\in\mathcal{C}_t}\frac{\log^{1.5}(NTd/\delta)}{\sqrt{b}}\|\nabla\mathcal{L}(\theta_t)\|_2\left\|\text{clip}\left(\sum_{k=1}^{K}g_{c,t,k},\tau\right)\right\|_2$$

$$\leq \frac{\log^{1.5}(NTd/\delta)G\tau_{K,G}}{\sqrt{b}}$$

with $\tau_{K,G} = \min\{\tau, KG\}$. Then denote

$$U_t = \sum_{\tau'=0}^{t}\left\langle \nabla\mathcal{L}(\theta_{\tau'}), \frac{1}{N}\sum_{c\in\mathcal{C}_{\tau'}}\left(R_{\tau'}^\top R_{\tau'}\text{clip}\left(\sum_{k=1}^{K}g_{c,\tau',k},\tau\right) - \text{clip}\left(\sum_{k=1}^{K}g_{c,\tau',k},\tau\right)\right)\right\rangle,$$

we can see that $U_t$ is a martingale with respect to the sketching matrices, and from the above analysis, we have that with probability at least $1-\delta$, for all $t\in[T]$,

$$|U_t - U_{t-1}| = \left|\left\langle \nabla\mathcal{L}(\theta_t), \frac{1}{N}\sum_{c\in\mathcal{C}_t}\left(R_t^\top R_t\text{clip}\left(\sum_{k=1}^{K}g_{c,t,k},\tau\right) - \text{clip}\left(\sum_{k=1}^{K}g_{c,t,k},\tau\right)\right)\right\rangle\right| \leq \frac{\log^{1.5}(NTd/\delta)G\tau_{K,G}}{\sqrt{b}}$$

Then by Azuma's inequality, we have

$$\mathbb{P}(U_{T-1}\leq -a) \leq \exp\left(-\frac{a^2}{2\cdot T\cdot\left(\frac{\log^{1.5}(NTd/\delta)G\tau_{K,G}}{\sqrt{b}}\right)^2}\right) = \exp\left(-\frac{ba^2}{2T\log^3(NTd/\delta)G^2\tau_{K,G}^2}\right)$$

By selecting $a = \frac{\log^2(NTd/\delta)\sqrt{2T}G\tau_{K,G}}{\sqrt{b}}$, we can get that with probability at least $1-2\delta$,

$$W_{T-1} = \sum_{t=1}^{T}\left\langle \nabla\mathcal{L}(\theta_t), \frac{1}{N}\sum_{c\in\mathcal{C}_t}\left(R_t^\top R_t\text{clip}\left(\sum_{k=1}^{K}g_{c,t,k},\tau\right) - \text{clip}\left(\sum_{k=1}^{K}g_{c,t,k},\tau\right)\right)\right\rangle \geq -\frac{\log^2(NTd/\delta)\sqrt{2T}G\tau_{K,G}}{\sqrt{b}}$$

$$(15)$$

Substituting 10, 11, 12, 13, 14, 15 into 9, we have that with probability at least $1-6\delta$,

$$\sum_{t=1}^{T}\frac{1}{N}\nabla\mathcal{L}(\theta_t)^\top\sum_{c\in\mathcal{C}_t}R_t^\top R_t\text{clip}\left(\sum_{k=1}^{K}g_{c,t,k},\tau\right)$$

$$= K\underbrace{\sum_{t=1}^{T}\left\langle \nabla\mathcal{L}(\theta_t), \frac{1}{C}\sum_{c=1}^{C}\nabla\mathcal{L}_c(\theta_t)\right\rangle}_{Y_1} + K\underbrace{\sum_{t=1}^{T}\left\langle \nabla\mathcal{L}(\theta_t), \frac{1}{N}\sum_{c\in\mathcal{C}_t}\nabla\mathcal{L}_c(\theta_t) - \frac{1}{C}\sum_{c=1}^{C}\nabla\mathcal{L}_c(\theta_t)\right\rangle}_{Y_2}$$

$$
+ \sum_{t=1}^{T} \underbrace{\left\langle \nabla\mathcal{L}(\theta_t), \frac{1}{N}\sum_{c\in\mathcal{C}_t}\sum_{k=1}^{K} \left(\nabla\mathcal{L}_c(\theta_{c,t,k}) - \nabla\mathcal{L}_c(\theta_t)\right) \right\rangle}_{Y_3} + \sum_{t=1}^{T} \underbrace{\left\langle \nabla\mathcal{L}(\theta_t), \frac{1}{N}\sum_{c\in\mathcal{C}_t}\sum_{k=1}^{K} \left(g_{c,t,k} - \nabla\mathcal{L}_c(\theta_{c,t,k})\right) \right\rangle}_{Y_4}
$$

$$
+ \sum_{t=1}^{T} \underbrace{\left\langle \nabla\mathcal{L}(\theta_t), \frac{1}{N}\sum_{c\in\mathcal{C}_t} \left( \mathrm{clip}\left(\sum_{k=1}^{K} g_{c,t,k}, \tau\right) - \sum_{k=1}^{K} g_{c,t,k}\right) \right\rangle}_{Y_5}
$$

$$
+ \sum_{t=1}^{T} \underbrace{\left\langle \nabla\mathcal{L}(\theta_t), \frac{1}{N}\sum_{c\in\mathcal{C}_t} \left( R_t^\top R_t\,\mathrm{clip}\left(\sum_{k=1}^{K} g_{c,t,k}, \tau\right) - \mathrm{clip}\left(\sum_{k=1}^{K} g_{c,t,k}, \tau\right)\right) \right\rangle}_{Y_6} \tag{16}
$$

$$
\geq K \sum_{t=1}^{T} \|\nabla\mathcal{L}(\theta_t)\|_2^2 - \frac{2K\sqrt{T}\log(2T/\delta)G^2}{\sqrt{N}} - \frac{\eta_{\mathrm{local}}TLK^2G^2}{2} - \frac{G\sqrt{2TK}\log(2T/\delta)\sigma_s}{\sqrt{N}}
$$

$$
- \max\left\{0, TG(KG-\tau)\right\} - \frac{\log^2(NTd/\delta)\sqrt{2T}G\tau_{K,G}}{\sqrt{b}} \tag{17}
$$

### D.1.2   Bounding $S_2$

We first bound each term with a fixed $t \in [T]$ in $S_2$. Noticing that $\frac{1}{N}\sum_{c\in\mathcal{C}_t}\mathbf{z}_{c,t} \sim \mathcal{N}\left(0, \frac{\sigma_g^2}{N}\mathbb{I}\right)$, and $R_t\nabla\mathcal{L}(\theta_t)$ is a $\frac{\|\nabla\mathcal{L}(\theta_t)\|_2}{\sqrt{b}}$-sub-Gaussian random vector, so according to Bernstein inequality,

$$
\mathbb{P}\left( \left|\left\langle R_t\nabla\mathcal{L}(\theta_t), \frac{1}{N}\sum_{c\in\mathcal{C}_t}\mathbf{z}_{c,t} \right\rangle\right| \geq a \right) \leq 2\exp\left( -\min\left( \frac{a^2}{b\cdot\frac{\sigma_g^2}{N}\cdot\frac{\|\nabla\mathcal{L}(\theta_t)\|_2^2}{b}}, \frac{a}{\frac{\sigma_g}{\sqrt{N}}\cdot\frac{\|\nabla\mathcal{L}(\theta_t)\|_2}{\sqrt{b}}} \right) \right)
$$

$$
= 2\exp\left( -\min\left( \frac{Na^2}{\sigma_g^2\|\nabla\mathcal{L}(\theta_t)\|_2^2}, \frac{a\sqrt{bN}}{\sigma_g\|\nabla\mathcal{L}(\theta_t)\|_2} \right) \right)
$$

so taking $a = \frac{\sigma_g\|\nabla\mathcal{L}(\theta_t)\|_2\log(2T/\delta)}{\sqrt{N}}$, we have that with probability at least $1 - \frac{\delta}{T}$,

$$
\left|\left\langle R_t\nabla\mathcal{L}(\theta_t), \frac{1}{N}\sum_{c\in\mathcal{C}_t}\mathbf{z}_{c,t} \right\rangle\right| \leq \frac{\sigma_g\|\nabla\mathcal{L}(\theta_t)\|_2\log(2T/\delta)}{\sqrt{N}} \leq \frac{\sigma_g G\log(2T/\delta)}{\sqrt{N}}
$$

Then denote $X_t = \sum_{\tau'=0}^{t}\left\langle R_{\tau'}\nabla\mathcal{L}(\theta_{\tau'}), \frac{1}{N}\sum_{c\in\mathcal{C}_{\tau'}}\mathbf{z}_{c,\tau'} \right\rangle$, we can see that $X_t$ is a martingale with respect to the Gaussian noise, and from the above analysis, we have that with probability at least $1 - \delta$, for all $t \in [T]$,

$$
|X_t - X_{t-1}| = \left|\left\langle R_t\nabla\mathcal{L}(\theta_t), \frac{1}{N}\sum_{c\in\mathcal{C}_t}\mathbf{z}_{c,t} \right\rangle\right| \leq \frac{\sigma_g G\log(2T/\delta)}{\sqrt{N}}
$$

Then by Azuma's inequality, we have

$$
\mathbb{P}\left(X_{T-1} \leq -a\right) \leq \exp\left( -\frac{a^2}{2\cdot T\cdot\left(\frac{\sigma_g G\log(2T/\delta)}{\sqrt{N}}\right)^2} \right) = \exp\left( -\frac{Na^2}{2T\sigma_g^2 G^2\log^2(2T/\delta)} \right)
$$

By selecting $a = \frac{\log^2(2T/\delta)\sqrt{2T}\sigma_g G}{\sqrt{N}}$, we can get that with probability at least $1 - 2\delta$,

$$
X_{T-1} = \sum_{t=1}^{T}\left\langle R_t\nabla\mathcal{L}(\theta_t), \frac{1}{N}\sum_{c\in\mathcal{C}_t}\mathbf{z}_{c,t} \right\rangle \geq -\frac{\log^2(2T/\delta)\sqrt{2T}\sigma_g G}{\sqrt{N}} \tag{18}
$$

Substituting 17 and 18 into 8, we have that with probability at least $1 - 8\delta$,

$$\sum_{t=1}^{T} \nabla\mathcal{L}(\theta_t)^\top (\theta_{t+1} - \theta_t)$$

$$= -\eta \underbrace{\sum_{t=1}^{T} \frac{1}{N} \nabla\mathcal{L}(\theta_t)^\top \sum_{c\in\mathcal{C}_t} R_t^\top R_t \mathrm{clip}\left(\sum_{k=1}^{K} g_{c,t,k}, \tau\right)}_{S_1} - \eta \underbrace{\sum_{t=1}^{T} \frac{1}{N} \nabla\mathcal{L}(\theta_t)^\top \sum_{c\in\mathcal{C}_t} R_t^\top \mathbf{z}_{c,t}}_{S_2}$$

$$\leq -\eta K \sum_{t=1}^{T} \|\nabla\mathcal{L}(\theta_t)\|_2^2 + \frac{2\eta K\sqrt{T}\log(2T/\delta)G^2}{\sqrt{N}} + \frac{\eta\eta_{\mathrm{local}}TLK^2G^2}{2} + \frac{\eta G\sqrt{2TK}\log(2T/\delta)\sigma_s}{\sqrt{N}}$$

$$+ \eta\max\{0, TG(KG - \tau)\} + \frac{\eta\log^2(NTd/\delta)\sqrt{2T}G\tau_{K,G}}{\sqrt{b}} + \frac{\eta\log^2(2T/\delta)\sqrt{2T}\sigma_g G}{\sqrt{N}}$$

$$\leq -\eta K \sum_{t=1}^{T} \|\nabla\mathcal{L}(\theta_t)\|_2^2 + \frac{2\eta K\sqrt{T}\log(2T/\delta)G^2}{\sqrt{N}} + \frac{\eta\eta_{\mathrm{local}}TLK^2G^2}{2} + \frac{\eta G\sqrt{2TK}\log(2T/\delta)\sigma_s}{\sqrt{N}}$$

$$+ \eta\max\{0, TG(KG - \tau)\} + \frac{\eta\log^2(NTd/\delta)\sqrt{2T}G\tau_{K,G}}{\sqrt{b}} + \log^2(2T/\delta)G^2 + \frac{2\eta^2 T\log^2(2T/\delta)\sigma_g^2}{N} \tag{19}$$

## D.2 Bounding $T_2$

We first bound each term in $T_2$ with a fixed $t \in [T]$.

$$(\theta_{t+1} - \theta_t)^\top \hat{H}_{\mathcal{L},t}(\theta_{t+1} - \theta_t) = \eta_{\mathrm{global}}^2 \left(\mathrm{desk}\left(\bar{\tilde{\Delta}}_t\right)\right)^\top \left(\sum_{i=1}^{d} \lambda_i v_i v_i^\top\right) \left(\mathrm{desk}\left(\bar{\tilde{\Delta}}_t\right)\right)$$

$$= \eta_{\mathrm{global}}^2 \sum_{i=1}^{d} \lambda_i \left|\left(\mathrm{desk}\left(\bar{\tilde{\Delta}}_t\right)\right)^\top v_i\right|^2 \tag{20}$$

For each $i \in [d]$, we have

$$\left|\left(\mathrm{desk}(\bar{\tilde{\Delta}}_t)\right)^\top v_i\right| = \left|\left\langle \mathrm{desk}(\bar{\tilde{\Delta}}_t), v_i\right\rangle\right|$$

$$= \frac{1}{N}\left|\left\langle R_t^\top \sum_{c\in\mathcal{C}_t} \tilde{\Delta}_{c,t}, v_i\right\rangle\right|$$

$$= \frac{1}{N}\left|\left\langle R_t^\top \sum_{c\in\mathcal{C}_\tau} \eta_{\mathrm{local}}\left(\mathrm{sk}\left(\mathrm{clip}\left(\frac{\Delta_{c,t}}{\eta_{\mathrm{local}}}, \tau\right)\right) + \mathbf{z}_{c,t}\right), v_i\right\rangle\right|$$

$$\leq \frac{\eta_{\mathrm{local}}}{N}\left|\left\langle \sum_{c\in\mathcal{C}_t} R_t^\top R_t\mathrm{clip}\left(\sum_{k=1}^{K} g_{c,t,k}, \tau\right), v_i\right\rangle\right| + \frac{\eta_{\mathrm{local}}}{N}\left|\left\langle \sum_{c\in\mathcal{C}_\tau} R_t^\top \mathbf{z}_{c,t}, v_i\right\rangle\right| \tag{21}$$

For the first term, according to Lemma B.1, with probability at least $1 - \frac{\delta}{dT}$,

$$\left|\left\langle \sum_{c\in\mathcal{C}_t} R_t^\top R_t\mathrm{clip}\left(\sum_{k=1}^{K} g_{c,t,k}, \tau\right), v_i\right\rangle\right|$$

$$\leq \left(1 + \frac{\log^{1.5}(NTd^2/\delta)}{\sqrt{b}}\right)\|v_i\|_2 \sum_{c\in\mathcal{C}_\tau}\left\|\mathrm{clip}\left(\sum_{k=1}^{K} g_{c,t,k}, \tau\right)\right\|$$

$$\leq \left(1 + \frac{\log^{1.5}(NTd^2/\delta)}{\sqrt{b}}\right)N\tau_{K,G} \tag{22}$$

For the second term, $\left\langle \frac{1}{N}\sum_{c\in\mathcal{C}_t} R_t^\top \mathbf{z}_{c,t}, v_i \right\rangle = \left\langle R_t v_i, \frac{1}{N}\sum_{c\in\mathcal{C}_t}\mathbf{z}_{c,\tau} \right\rangle$. Noticing that $\frac{1}{N}\sum_{c\in\mathcal{C}_t}\mathbf{z}_{c,t} \sim \mathcal{N}\left(0, \frac{\sigma_g^2}{N}\mathbb{I}\right)$, and $R_t v_i$ is a $\frac{1}{\sqrt{b}}$-sub-Gaussian random vector, so according to Bernstein inequality,

$$\mathbb{P}\left(\left|\left\langle R_t v_i, \frac{1}{N}\sum_{c\in\mathcal{C}_t}\mathbf{z}_{c,t}\right\rangle\right| \geq a\right) \leq 2\exp\left(-c\min\left(\frac{a^2}{\frac{\sigma_g^2}{N}}, \frac{t}{\frac{\sigma_g}{\sqrt{bN}}}\right)\right) = 2\exp\left(-c\min\left(\frac{Na^2}{\sigma_g^2}, \frac{a\sqrt{bN}}{\sigma_g}\right)\right)$$

so taking $a = \frac{\sigma_g \log(2dT/\delta)}{\sqrt{N}}$, we have that with probability at least $1 - \frac{\delta}{dT}$,

$$\left|\left\langle \frac{1}{N}\sum_{c\in\mathcal{C}_t} R_t^\top \mathbf{z}_{c,t}, v_i\right\rangle\right| \leq \frac{\sigma_g \log(2dT/\delta)}{\sqrt{N}} \tag{23}$$

Substituting 22 and 23 into 21, we have that with probability at least $1 - \frac{2\delta}{dT}$,

$$\left|\left(\mathrm{desk}(\bar{\bar{\Delta}}_t)\right)^\top v_i\right| \leq \frac{\eta_{\mathrm{local}}}{N}\left|\left\langle \sum_{c\in\mathcal{C}_t} R_t^\top R_t \mathrm{clip}\left(\sum_{k=1}^K g_{c,t,k}, \tau\right), v_i\right\rangle\right| + \frac{\eta_{\mathrm{local}}}{N}\left|\left\langle \sum_{c\in\mathcal{C}_\tau} R_t^\top \mathbf{z}_{c,t}, v_i\right\rangle\right|$$

$$\leq \eta_{\mathrm{local}}\left(1 + \frac{\log^{1.5}(NTd^2/\delta)}{\sqrt{b}}\right)\tau_{K,G} + \frac{\eta_{\mathrm{local}}\sigma_g \log(2dT/\delta)}{\sqrt{N}}$$

which implies that

$$\left|\left(\mathrm{desk}(\bar{\bar{\Delta}}_t)\right)^\top v_i\right|^2 \leq 2\left(\eta_{\mathrm{local}}\left(1 + \frac{\log^{1.5}(NTd^2/\delta)}{\sqrt{b}}\right)\tau_{K,G}\right)^2 + 2\left(\frac{\eta_{\mathrm{local}}\sigma_g \log(2dT/\delta)}{\sqrt{N}}\right)^2$$

$$= 2\eta_{\mathrm{local}}^2\left(1 + \frac{\log^{1.5}(NTd^2/\delta)}{\sqrt{b}}\right)^2 \tau_{K,G}^2 + \frac{2\eta_{\mathrm{local}}^2\sigma_g^2 \log^2(2dT/\delta)}{N} \tag{24}$$

Substituting 24 into 20, we have that with probability at least $1 - \frac{2\delta}{T}$,

$$(\theta_{t+1} - \theta_t)^\top \hat{H}_{\mathcal{L},t}(\theta_{t+1} - \theta_t) = \eta_{\mathrm{global}}^2 \sum_{i=1}^d \lambda_i \left|\left(\mathrm{desk}\left(\bar{\bar{\Delta}}_t\right)\right)^\top v_i\right|^2$$

$$\leq \eta_{\mathrm{global}}^2\left(2\eta_{\mathrm{local}}^2\left(1 + \frac{\log^{1.5}(NTd^2/\delta)}{\sqrt{b}}\right)^2 \tau_{K,G}^2 + \frac{2\eta_{\mathrm{local}}^2\sigma_g^2 \log^2(2dT/\delta)}{N}\right)\sum_{i=1}^d |\lambda_i|$$

$$= 2\eta^2\left(1 + \frac{\log^{1.5}(NTd^2/\delta)}{\sqrt{b}}\right)^2 \mathcal{I}L\tau_{K,G}^2 + \frac{2\eta^2\sigma_g^2\mathcal{I}L \log^2(2dT/\delta)}{N}$$

By taking summation from 0 to $T - 1$, we have that with probability at least $1 - 2\delta$,

$$\sum_{t=1}^T (\theta_{t+1} - \theta_t)^\top \hat{H}_{\mathcal{L},t}(\theta_{t+1} - \theta_t) \leq 2\eta^2\left(1 + \frac{\log^{1.5}(NTd^2/\delta)}{\sqrt{b}}\right)^2 \mathcal{I}LT\tau_{K,G}^2 + \frac{2\eta^2\sigma_g^2\mathcal{I}LT \log^2(2dT/\delta)}{N} \tag{25}$$

Substituting 19 and 25 into 7, we have that with probability at least $1 - 10\delta$,

$$\mathcal{L}(\theta_T) - \mathcal{L}(\theta_0) = \sum_{t=1}^T \nabla\mathcal{L}(\theta_t)^\top (\theta_{t+1} - \theta_t) + \frac{1}{2}\sum_{t=1}^T (\theta_{t+1} - \theta_t)^\top \hat{H}_{\mathcal{L},t}(\theta_{t+1} - \theta_t)$$

$$\leq -\eta K\sum_{t=1}^T \|\nabla\mathcal{L}(\theta_t)\|_2^2 + \frac{2\eta K\sqrt{T}\log(2T/\delta)G^2}{\sqrt{N}} + \frac{\eta\eta_{\mathrm{local}}TLK^2G^2}{2} + \frac{\eta G\sqrt{2TK}\log(2T/\delta)\sigma_s}{\sqrt{N}}$$

$$+ \eta\max\{0, TG(KG - \tau)\} + \frac{\eta\log^2(NTd/\delta)\sqrt{2T}G\tau_{K,G}}{\sqrt{b}} + \frac{\eta\log^2(NTd/\delta)\sqrt{2T}G\tau}{\sqrt{b}} + \log^2(2T/\delta)G^2$$

$$+ \frac{2\eta^2 T\log^2(2T/\delta)\sigma_g^2}{N} + 2\eta^2\left(1 + \frac{\log^{1.5}(NTd^2/\delta)}{\sqrt{b}}\right)^2 \mathcal{I}LT\tau_{K,G}^2 + \frac{2\eta^2\sigma_g^2\mathcal{I}LT \log^2(2dT/\delta)}{N}$$

Since

$$\mathcal{L}(\theta_0) - \mathcal{L}(\theta_T) \leq \mathcal{L}(\theta_0) - \mathcal{L}^*$$

we can get that with probability at least $1 - 10\delta$,

$$\frac{1}{T}\sum_{t=1}^{T} \|\nabla\mathcal{L}(\theta_t)\|_2^2 \leq \frac{\mathcal{L}(\theta_0) - \mathcal{L}^*}{\eta KT} + \frac{2\log(2T/\delta)G^2}{\sqrt{NT}} + \frac{\eta_{\text{local}}LKG^2}{2} + \frac{\sqrt{2}G\log(2T/\delta)\sigma_s}{\sqrt{NTK}}$$

$$+ \max\left\{0, \frac{G(KG-\tau)}{K}\right\} + \frac{\sqrt{2}\log^2(NTd/\delta)G\tau_{K,G}}{\sqrt{bT}K} + \frac{\log^2(2T/\delta)G^2}{\eta TK}$$

$$+ \frac{2\eta\log^2(2T/\delta)\sigma_g^2}{NK} + \frac{2\eta\alpha_1^2\mathcal{I}L\tau_{K,G}^2}{K} + \frac{2\eta\sigma_g^2\mathcal{I}L\log^2(2dT/\delta)}{NK}$$

in which

$$\alpha_1 = 1 + \frac{\log^{1.5}(NTd^2/\delta)}{\sqrt{b}}, \tau_{K,G} = \min\{\tau, KG\},$$

then we finish the proof. $\qquad\square$

# E  Optimization Analysis with AMSGrad as `GLOBAL_OPT`

Similar to Appendix D, we will state Algorithm 4, which will be used as `GLOBAL_OPT` to do a one-step adaptive update of $\theta_{t-1}$ with $\text{desk}\left(\bar{\bar{\Delta}}_{t-1}\right)$.

---

**Algorithm 4** `GLOBAL_OPT` (AMSGrad)

---

**Inputs:** model $\theta_{t-1}$, desketched update $\text{desk}\left(\bar{\bar{\Delta}}_{t-1}\right)$.

**Output:** model $\theta_T$.

**Update first moment estimate:** $m_t = \beta_1 m_{t-1} + (1 - \beta_1) \text{desk}(\bar{\bar{\Delta}}_{t-1})$

**Update second moment estimate:** $\hat{v}_t = \beta_2 v_{t-1} + (1 - \beta_2) \left(\text{desk}(\bar{\bar{\Delta}}_{t-1})\right)^2$

**Update maximum of past second moment estimates:** $v_t = \max\left(\hat{v}_t, v_{t-1}\right)$

**Update parameters:** $\theta_t = \theta_{t-1} - \eta_{\text{global}} \frac{m_{t-1}}{\sqrt{v_{t-1}} + \epsilon} := \eta_{\text{global}} V_{t-1}^{-\frac{1}{2}} m_{t-1}$

---

Then we demonstrate the optimization result of Algorithm 2 with Algorithm 4, which is the formal version of Theorem 3.1.

**Theorem E.1.** Suppose $\{\theta_t\}_{t=0}^T$ is generated by Algorithm 2 with Algorithm 4 as `GLOBAL_OPT`. Denote $\mathcal{L}^*$ the minimum of the average empirical loss. Under Assumption 1-4, with learning rate $\eta = \eta_{\text{global}} \eta_{\text{local}}$, we have that with probability at least $1 - 19\delta$,

$$
\frac{1}{T} \sum_{t=1}^T \|\nabla \mathcal{L}(\theta_t)\|_2^2
$$

$$
\leq \frac{\alpha_2 \left(\mathcal{L}(\theta_0) - \mathcal{L}^*\right)}{\eta KT} + \frac{2\alpha_2 \log(2T/\delta) G^2}{\sqrt{NT}\epsilon} + \frac{\eta_{\text{local}} \alpha_2 LKG^2}{2\epsilon} + \frac{\sqrt{2}\alpha_2 G \log(2T/\delta)\sigma_s}{\sqrt{NTK}\epsilon}
$$

$$
+ \max\left\{0, \frac{\alpha_2 G(KG - \tau)}{K\epsilon}\right\} + \frac{\sqrt{2}\alpha_2 \log^2(NTd/\delta) G \tau_{K,G}}{\sqrt{bT}K\epsilon} + \frac{\eta_{\text{local}} \alpha_1^2 \alpha_2 (2 + \beta_1)\sqrt{1 - \beta_2} G \tau_{K,G}^2 \log(2dT/\delta)}{K\epsilon^2(1 - \beta_1)}
$$

$$
+ \frac{\eta_{\text{local}} \alpha_2 (1 + 2\beta_1)\sqrt{1 - \beta_2}\sigma_g^2 G \log^2(2dT/\delta)}{NK\epsilon^2(1 - \beta_1)} + \frac{\alpha_2 \log^2(2T/\delta) G^2}{\eta TK\epsilon} + \frac{2\eta\alpha_2 \log^2(2T/\delta)\sigma_g^2}{NK\epsilon}
$$

$$
+ \frac{2\eta\alpha_1^2 \alpha_2 (1 + 2\beta_1)(1 + \beta_1)\mathcal{I}L\tau_{K,G}^2}{K\epsilon^2(1 - \beta_1)^2} + \frac{2\eta\alpha_2(1 + 2\beta_1)(1 + \beta_1)\sigma_g^2 \mathcal{I}L \log^2(2dT/\delta)}{NK\epsilon^2(1 - \beta_1)^2}
$$

in which

$$
\alpha_1 = 1 + \frac{\log^{1.5}(NTd^2/\delta)}{\sqrt{b}}, \quad \alpha_2 = \eta_{\text{local}}\left(1 + \frac{\log^{1.5}(NTd^2/\delta)}{\sqrt{b}}\right)\tau_{K,G} + \frac{\eta_{\text{local}}\sigma_g \log(2dT/\delta)}{\sqrt{N}} + \epsilon
$$

with $\tau_{k,G} = \min\{\tau, KG\}$.

**Remark E.1.** To analyze the bound in terms of $\tau$, we consider all other parameters as fixed and distinguish between two principal regimes:

1. When $\tau \leq KG$: In this regime, clipping may be activated, and $\tau$ acts as an upper bound on the norm of each clipped update $\Delta_{c,t}$ according to our algorithm. The total bound in this case is given by $\frac{\alpha_2(\mathcal{L}(\theta_0) - \mathcal{L}^*)}{\eta KT} + \frac{2\alpha_2 \log(2T/\delta) G^2}{\sqrt{NT}\epsilon} + \frac{\eta_{\text{local}} \alpha_2 LKG^2}{2\epsilon} + \frac{\sqrt{2}\alpha_2 G \log(2T/\delta)\sigma_s}{\sqrt{NTK}\epsilon} + \max\left\{0, \frac{\alpha_2 G(KG - \tau)}{K\epsilon}\right\} + \frac{\sqrt{2}\alpha_2 \log^2(NTd/\delta) G\tau}{\sqrt{bT}K\epsilon} + \frac{\eta_{\text{local}}\alpha_1^2\alpha_2(2+\beta_1)\sqrt{1 - \beta_2}G\tau^2 \log(2dT/\delta)}{K\epsilon^2(1 - \beta_1)} + \frac{\eta_{\text{local}}\alpha_2(1 + 2\beta_1)\sqrt{1 - \beta_2}\sigma_g^2 G \log^2(2dT/\delta)}{NK\epsilon^2(1 - \beta_1)} + \frac{\alpha_2 \log^2(2T/\delta) G^2}{\eta TK\epsilon} + \frac{2\eta\alpha_2 \log^2(2T/\delta)\sigma_g^2}{NK\epsilon} + \frac{2\eta\alpha_1^2\alpha_2(1+2\beta_1)(1+\beta_1)\mathcal{I}L\tau^2}{K\epsilon^2(1-\beta_1)^2} + \frac{2\eta\alpha_2(1+2\beta_1)(1+\beta_1)\sigma_g^2\mathcal{I}L\log^2(2dT/\delta)}{NK\epsilon^2(1-\beta_1)^2}$. We can observe that $\frac{\sqrt{2}\alpha_2 \log^2(NTd/\delta)G\tau}{\sqrt{bT}K\epsilon} + \frac{\eta_{\text{local}}\alpha_1^2\alpha_2(2+\beta_1)\sqrt{1-\beta_2}G\tau^2\log(2dT/\delta)}{K\epsilon^2(1-\beta_1)} + \frac{2\eta\alpha_1^2\alpha_2(1+2\beta_1)(1+\beta_1)\mathcal{I}L\tau^2}{K\epsilon^2(1-\beta_1)^2}$ in the optimization bound of the sketching-based algorithm is monotonically increasing in $\tau$, while $\max\left\{0, \frac{\alpha_2 G(KG - \tau)}{K\epsilon}\right\}$ caused by clipping is monotonically decreasing in $\tau$. This trade-off highlights the need for a careful choice of $\tau$, as overly aggressive

clipping introduces clipping bias, while too loose a threshold amplifies optimization error of sketching algorithms.

2. When $\tau \geq KG$: In this regime, the clipping operation becomes inactive under our algorithm, as all updates fall within the clipping threshold $\tau$. $KG$ effectively bounds the norm of each update $\Delta_{c,t}$, and the clipping no longer influences the computation. The total bound becomes $\frac{\alpha_2(\mathcal{L}(\theta_0)-\mathcal{L}^*)}{\eta KT} + \frac{2\alpha_2 \log(2T/\delta)G^2}{\sqrt{NT}\epsilon} + \frac{\eta_{\text{local}}\alpha_2 LKG^2}{2\epsilon} + \frac{\sqrt{2}\alpha_2 G \log(2T/\delta)\sigma_s}{\sqrt{NTK}\epsilon} + \frac{\sqrt{2}\alpha_2 \log^2(NTd/\delta)G^2}{\sqrt{bT}\epsilon} + \frac{\eta_{\text{local}}\alpha_1^2\alpha_2(2+\beta_1)\sqrt{1-\beta_2}KG^3 \log(2dT/\delta)}{\epsilon^2(1-\beta_1)} + \frac{\eta_{\text{local}}\alpha_2(1+2\beta_1)\sqrt{1-\beta_2}\sigma_g^2 G \log^2(2dT/\delta)}{NK\epsilon^2(1-\beta_1)} + \frac{\alpha_2 \log^2(2T/\delta)G^2}{\eta TK\epsilon} + \frac{2\eta\alpha_2 \log^2(2T/\delta)\sigma_g^2}{NK\epsilon} + \frac{2\eta\alpha_1^2\alpha_2(1+2\beta_1)(1+\beta_1)\mathcal{I}LKG^2}{\epsilon^2(1-\beta_1)^2} + \frac{2\eta\alpha_2(1+2\beta_1)(1+\beta_1)\sigma_g^2\mathcal{I}L \log^2(2dT/\delta)}{NK\epsilon^2(1-\beta_1)^2}$. Consequently, the entire bound becomes independent of $\tau$ and remains constant as $\tau$ increases further.

*Proof.* Let

$$\gamma_t = \theta_t + \frac{\beta_1}{1-\beta_1}(\theta_t - \theta_{t-1}) = \frac{1}{1-\beta_1}\theta_t - \frac{\beta_1}{1-\beta_1}\theta_{t-1}$$

and set $\theta_{-1} = \theta_0$ so that $\gamma_0 = \theta_0$. Then, the update on $\gamma_t$ can be expressed as

$$\begin{aligned}
\gamma_{t+1} - \gamma_t &= \frac{1}{1-\beta_1}(\theta_{t+1} - \theta_t) - \frac{\beta_1}{1-\beta_1}(\theta_t - \theta_{t-1}) \\
&= -\frac{1}{1-\beta_1}\eta_{\text{global}}V_t^{-1/2} \cdot m_t + \frac{\beta_1}{1-\beta_1}\eta_{\text{global}}V_{t-1}^{-1/2} \cdot m_{t-1} \\
&= -\frac{1}{1-\beta_1}\eta_{\text{global}}V_t^{-1/2} \cdot (\beta_1 m_{t-1} + (1-\beta_1)\operatorname{desk}(\bar{\tilde{\Delta}}_t)) + \frac{\beta_1}{1-\beta_1}\eta_{\text{global}}V_{t-1}^{-1/2} \cdot m_{t-1} \\
&= \frac{\beta_1\eta_{\text{global}}}{1-\beta_1}\left(V_{t-1}^{-1/2} - V_t^{-1/2}\right)m_{t-1} - \eta_{\text{global}}V_t^{-1/2}\operatorname{desk}(\bar{\tilde{\Delta}}_t) \\
&= \frac{\beta_1\eta_{\text{global}}}{1-\beta_1}\left(V_{t-1}^{-1/2} - V_t^{-1/2}\right)m_{t-1} - \frac{\eta_{\text{global}}}{N}V_t^{-1/2}R_t^\top \sum_{c\in\mathcal{C}_t}\tilde{\Delta}_{c,t} \\
&= \frac{\beta_1\eta_{\text{global}}}{1-\beta_1}\left(V_{t-1}^{-1/2} - V_t^{-1/2}\right)m_{t-1} - \frac{\eta_{\text{global}}}{N}V_t^{-1/2}R_t^\top \sum_{c\in\mathcal{C}_t}\eta_{\text{local}}\left(\operatorname{sk}\left(\operatorname{clip}\left(\frac{\Delta_{c,t}}{\eta_{\text{local}}},\tau\right)\right) + \mathbf{z}_{c,t}\right) \\
&= \frac{\beta_1\eta_{\text{global}}}{1-\beta_1}\left(V_{t-1}^{-1/2} - V_t^{-1/2}\right)m_{t-1} - \frac{\eta}{N}V_t^{-1/2}\sum_{c\in\mathcal{C}_t}R_t^\top R_t\operatorname{clip}\left(\sum_{k=1}^K g_{c,t,k},\tau\right) - \frac{\eta}{N}V_t^{-1/2}\sum_{c\in\mathcal{C}_t}R_t^\top \mathbf{z}_{c,t}
\end{aligned}$$

in which $\eta = \eta_{\text{global}}\eta_{\text{local}}$. By Taylor expansion, we have

$$\begin{aligned}
\mathcal{L}(\gamma_{t+1}) &= \mathcal{L}(\gamma_t) + \nabla\mathcal{L}(\gamma_t)^\top(\gamma_{t+1} - \gamma_t) + \frac{1}{2}(\gamma_{t+1} - \gamma_t)^\top\hat{H}_\mathcal{L}(\gamma_{t+1} - \gamma_t) \\
&= \mathcal{L}(\gamma_t) + \nabla\mathcal{L}(\theta_t)^\top(\gamma_{t+1} - \gamma_t) + (\nabla\mathcal{L}(\gamma_t) - \nabla\mathcal{L}(\theta_t))^\top(\gamma_{t+1} - \gamma_t) + \frac{1}{2}(\gamma_{t+1} - \gamma_t)^\top\hat{H}_{\mathcal{L},t}(\gamma_{t+1} - \gamma_t).
\end{aligned}$$

By taking summation from 0 to $T-1$, we can get that

$$\mathcal{L}(\gamma_T) - \mathcal{L}(\theta_0) = \mathcal{L}(\gamma_T) - \mathcal{L}(\gamma_0) = \sum_{t=1}^T (\mathcal{L}(\gamma_{t+1}) - \mathcal{L}(\gamma_t))$$

$$= \underbrace{\sum_{t=1}^T \nabla\mathcal{L}(\theta_t)^\top(\gamma_{t+1} - \gamma_t)}_{T_1} + \underbrace{\sum_{t=1}^T (\nabla\mathcal{L}(\gamma_t) - \nabla\mathcal{L}(\theta_t))^\top(\gamma_{t+1} - \gamma_t)}_{T_2} + \underbrace{\frac{1}{2}\sum_{t=1}^T (\gamma_{t+1} - \gamma_t)^\top\hat{H}_{\mathcal{L},t}(\gamma_{t+1} - \gamma_t)}_{T_3}$$

$$\tag{26}$$

### E.1 Bounding $T_1$

For each term in $T_1$, we have

$$\nabla\mathcal{L}(\theta_t)^\top(\gamma_{t+1} - \gamma_t)$$

$$=\nabla\mathcal{L}(\theta_t)^\top\left(\frac{\beta_1\eta_{\text{global}}}{1-\beta_1}\left(V_{t-1}^{-1/2}-V_t^{-1/2}\right)m_{t-1}-\frac{\eta}{N}V_t^{-1/2}\sum_{c\in\mathcal{C}_t}R_t^\top R_t\text{clip}\left(\sum_{k=1}^K g_{c,t,k},\tau\right)-\frac{\eta}{L}V_t^{-1/2}\sum_{c\in\mathcal{C}_t}R_t^\top\mathbf{z}_{c,t}\right)$$

$$=\nabla\mathcal{L}(\theta_t)^\top\frac{\beta_1\eta_{\text{global}}}{1-\beta_1}\left(V_{t-1}^{-1/2}-V_t^{-1/2}\right)m_{t-1}-\frac{\eta}{N}\nabla\mathcal{L}(\theta_t)^\top\left(V_t^{-1/2}-V_{t-1}^{-1/2}\right)\sum_{c\in\mathcal{C}_t}R_t^\top R_t\text{clip}\left(\sum_{k=1}^K g_{c,t,k},\tau\right)$$

$$-\frac{\eta}{N}\nabla\mathcal{L}(\theta_t)^\top V_{t-1}^{-1/2}\sum_{c\in\mathcal{C}_t}R_t^\top R_t\text{clip}\left(\sum_{k=1}^K g_{c,t,k},\tau\right)-\frac{\eta}{N}\nabla\mathcal{L}(\theta_t)^\top V_t^{-1/2}\sum_{c\in\mathcal{C}_t}R_t^\top\mathbf{z}_{c,t}$$

By taking summation from $0$ to $T-1$, we can get that

$$\sum_{t=1}^T\nabla\mathcal{L}(\theta_t)^\top(\gamma_{t+1}-\gamma_t)$$

$$=\underbrace{\sum_{t=1}^T\nabla\mathcal{L}(\theta_t)^\top\frac{\beta_1\eta_{\text{global}}}{1-\beta_1}\left(V_{t-1}^{-1/2}-V_t^{-1/2}\right)m_{t-1}}_{S_1}$$

$$\underbrace{-\eta\sum_{t=1}^T\frac{1}{N}\nabla\mathcal{L}(\theta_t)^\top\left(V_t^{-1/2}-V_{t-1}^{-1/2}\right)\sum_{c\in\mathcal{C}_t}R_t^\top R_t\text{clip}\left(\sum_{k=1}^K g_{c,t,k},\tau\right)}_{S_2}$$

$$\underbrace{-\eta\sum_{t=1}^T\frac{1}{N}\nabla\mathcal{L}(\theta_t)^\top V_{t-1}^{-1/2}\sum_{c\in\mathcal{C}_t}R_t^\top R_t\text{clip}\left(\sum_{k=1}^K g_{c,t,k},\tau\right)}_{S_3}\underbrace{-\eta\sum_{t=1}^T\frac{1}{N}\nabla\mathcal{L}(\theta_t)^\top V_t^{-1/2}\sum_{c\in\mathcal{C}_t}R_t^\top\mathbf{z}_{c,t}}_{S_4}$$

$$(27)$$

### E.1.1 Bounding $S_2$

We first bound each term with a fixed $t\in[T]$ in $S_2$. According to Lemma B.1, we have that with probability at least $1-\frac{\delta}{T}$,

$$\left|\frac{1}{N}\nabla\mathcal{L}(\theta_t)^\top\left(V_t^{-1/2}-V_{t-1}^{-1/2}\right)\sum_{c\in\mathcal{C}_t}R_t^\top R_t\text{clip}\left(\sum_{k=1}^K g_{c,t,k},\tau\right)\right|$$

$$=\left|\frac{1}{N}\left\langle\nabla\mathcal{L}(\theta_t),\left(V_t^{-1/2}-V_{t-1}^{-1/2}\right)\sum_{c\in\mathcal{C}_t}R_t^\top R_t\text{clip}\left(\sum_{k=1}^K g_{c,t,k},\tau\right)\right\rangle\right|$$

$$\leq\frac{1}{N}\left(1+\frac{\log^{1.5}(NTd^2/\delta)}{\sqrt{b}}\right)\|\nabla\mathcal{L}(\theta_t)\|_2\left\|V_t^{-1/2}-V_{t-1}^{-1/2}\right\|_2\sum_{c\in\mathcal{C}_t}\left\|\text{clip}\left(\sum_{k=1}^K g_{c,t,k},\tau\right)\right\|_2$$

$$\leq G\tau_{K,G}\left(1+\frac{\log^{1.5}(NTd^2/\delta)}{\sqrt{b}}\right)\left\|V_t^{-1/2}-V_{t-1}^{-1/2}\right\|_2 \qquad (28)$$

In addition, if $v_{t,i}=v_{t-1,i}$, then $\left[V_t^{-1/2}-V_{t-1}^{-1/2}\right]_i=0$. If $v_{t,i}>v_{t-1,i}$, then $\left[V_t^{-1/2}-V_{t-1}^{-1/2}\right]_i<0$, and

$$\left|\left[V_t^{-1/2}-V_{t-1}^{-1/2}\right]_i\right|=\left|\frac{1}{\sqrt{v_{t,i}}+\epsilon}-\frac{1}{\sqrt{v_{t-1,i}}+\epsilon}\right|$$

$$=\left|\frac{\sqrt{v_{t-1,i}}-\sqrt{v_{t,i}}}{\left(\sqrt{v_{t,i}}+\epsilon\right)\left(\sqrt{v_{t-1,i}}+\epsilon\right)}\right|$$

$$=\left|\frac{v_{t-1,i}-v_{t,i}}{\left(\sqrt{v_{t,i}}+\epsilon\right)\left(\sqrt{v_{t-1,i}}+\epsilon\right)\left(\sqrt{v_{t-1,i}}+\sqrt{v_{t,i}}\right)}\right|$$

$$= \frac{(1-\beta_2)\left(\left[\left(\mathrm{desk}(\bar{\bar{\Delta}}_t)\right)^2\right]_i - v_{t-1,i}\right)}{\left(\sqrt{v_{t,i}}+\epsilon\right)\left(\sqrt{v_{t-1,i}}+\epsilon\right)\left(\sqrt{v_{t-1,i}}+\sqrt{v_{t,i}}\right)}$$

$$\leq \frac{(1-\beta_2)\left[\left(\mathrm{desk}(\bar{\bar{\Delta}}_t)\right)^2\right]_i}{\epsilon^2\sqrt{(1-\beta_2)\left[\left(\mathrm{desk}(\bar{\bar{\Delta}}_t)\right)^2\right]_i}}$$

$$= \frac{\sqrt{1-\beta_2}}{\epsilon^2}\sqrt{\left[\left(\mathrm{desk}(\bar{\bar{\Delta}}_t)\right)^2\right]_i}$$

$$= \frac{\sqrt{1-\beta_2}}{\epsilon^2}\left|\left[\frac{\eta_{\mathrm{local}}}{N}\sum_{c\in\mathcal{C}_t}\left(R_t^\top R_t \mathrm{clip}\left(\frac{\Delta_{c,t}}{\eta_{\mathrm{local}}},\tau\right) + R_t^\top \mathbf{z}_{c,t}\right)\right]_i\right|$$

$$\leq \frac{\sqrt{1-\beta_2}\eta_{\mathrm{local}}}{N\epsilon^2}\left|\left[\sum_{c\in\mathcal{C}_t}\left[R_t^\top R_t \mathrm{clip}\left(\frac{\Delta_{c,t}}{\eta_{\mathrm{local}}},\tau\right)\right]\right]_i\right| + \frac{\sqrt{1-\beta_2}\eta_{\mathrm{local}}}{N\epsilon^2}\left|\left[\sum_{c\in\mathcal{C}_t}R_t^\top \mathbf{z}_{c,t}\right]_i\right|$$
(29)

Therefore, we have

$$\left\|V_t^{-1/2} - V_{t-1}^{-1/2}\right\|_2 = \max_i \left|\left[V_t^{-1/2} - V_{t-1}^{-1/2}\right]_i\right|$$

$$\leq \max_i \left\{\frac{\sqrt{1-\beta_2}\eta_{\mathrm{local}}}{N\epsilon^2}\left|\left[\sum_{c\in\mathcal{C}_t}\left[R_t^\top R_t \mathrm{clip}\left(\frac{\Delta_{c,t}}{\eta_{\mathrm{local}}},\tau\right)\right]\right]_i\right| + \frac{\sqrt{1-\beta_2}\eta_{\mathrm{local}}}{N\epsilon^2}\left|\left[\sum_{c\in\mathcal{C}_t}R_t^\top \mathbf{z}_{c,t}\right]_i\right|\right\}$$

$$\leq \max_i \left\{\frac{\sqrt{1-\beta_2}\eta_{\mathrm{local}}}{N\epsilon^2}\left|\left[\sum_{c\in\mathcal{C}_t}\left[R_t^\top R_t \mathrm{clip}\left(\frac{\Delta_{c,t}}{\eta_{\mathrm{local}}},\tau\right)\right]\right]_i\right|\right\} + \max_i \left\{\frac{\sqrt{1-\beta_2}\eta_{\mathrm{local}}}{N\epsilon^2}\left|\left[\sum_{c\in\mathcal{C}_t}R_t^\top \mathbf{z}_{c,t}\right]_i\right|\right\}$$
(30)

For the first term, according to Lemma B.1, for each $i \in [d]$, with probability at least $1-\frac{\delta}{Td}$,

$$\left|\left[R_t^\top R_t \mathrm{clip}\left(\frac{\Delta_{c,t}}{\eta_{\mathrm{local}}},\tau\right)\right]_i\right| = \left|e_i R_t^\top R_t \mathrm{clip}\left(\frac{\Delta_{c,t}}{\eta_{\mathrm{local}}},\tau\right)\right| = \left|\left\langle R_t e_i, R_t \mathrm{clip}\left(\frac{\Delta_{c,t}}{\eta_{\mathrm{local}}},\tau\right)\right\rangle\right| \leq \left(1 + \frac{\log^{1.5}(NTd^2/\delta)}{\sqrt{b}}\right)\tau_{K,G}$$
(31)

then we can get that with probability at least $1-\frac{\delta}{T}$,

$$\max_i \frac{\sqrt{1-\beta_2}\eta_{\mathrm{local}}}{N\epsilon^2}\left|\left[\sum_{c\in\mathcal{C}_t}\left[R_t^\top R_t \mathrm{clip}\left(\frac{\Delta_{c,t}}{\eta_{\mathrm{local}}},\tau\right)\right]\right]_i\right|$$

$$\leq \max_i \frac{\sqrt{1-\beta_2}\eta_{\mathrm{local}}}{N\epsilon^2}\sum_{c\in\mathcal{C}_t}\left|\left[R_t^\top R_t \mathrm{clip}\left(\frac{\Delta_{c,t}}{\eta_{\mathrm{local}}},\tau\right)\right]_i\right|$$

$$\leq \frac{\sqrt{1-\beta_2}\eta_{\mathrm{local}}}{N\epsilon^2}\cdot N\cdot\left(1 + \frac{\log^{1.5}(NTd^2/\delta)}{\sqrt{b}}\right)\tau_{K,G}$$

$$= \frac{\sqrt{1-\beta_2}\eta_{\mathrm{local}}\tau_{K,G}}{\epsilon^2}\left(1 + \frac{\log^{1.5}(NTd^2/\delta)}{\sqrt{b}}\right)$$
(32)

For the second term, $\frac{1}{N}\left[\sum_{c\in\mathcal{C}_t}R_t^\top \mathbf{z}_{c,t}\right]_i = \left\langle R_t e_i, \frac{1}{N}\sum_{c\in\mathcal{C}_t}\mathbf{z}_{c,t}\right\rangle$. Noticing that $\frac{1}{N}\sum_{c\in\mathcal{C}_t}\mathbf{z}_{c,t} \sim \mathcal{N}\left(0, \frac{\sigma_g^2}{N}\mathbb{I}\right)$, and $R_t e_i$ is a $\frac{1}{\sqrt{b}}$-sub-Gaussian random vector, so according to Bernstein inequality,

$$\mathbb{P}\left(\left|\left\langle R_t e_i, \frac{1}{N}\sum_{c\in\mathcal{C}_t}\mathbf{z}_{c,t}\right\rangle\right| \geq a\right) \leq 2\exp\left(-c\min\left(\frac{a^2}{\frac{\sigma_g^2}{N}}, \frac{a}{\frac{\sigma_g}{\sqrt{bN}}}\right)\right) = 2\exp\left(-c\min\left(\frac{Na^2}{\sigma_g^2}, \frac{a\sqrt{bN}}{\sigma_g}\right)\right)$$

so taking $a = \frac{\sigma_g \log(2dT/\delta)}{\sqrt{N}}$, we have that for each $i \in [d]$, with probability at least $1 - \frac{\delta}{Td}$,

$$\frac{1}{N} \left| \left[ \sum_{c \in \mathcal{C}_t} R_t^\top \mathbf{z}_{c,t} \right]_i \right| \leq \frac{\sigma_g \log(2dT/\delta)}{\sqrt{N}} \tag{33}$$

then we can get that with probability at least $1 - \frac{\delta}{T}$,

$$\max_i \frac{\sqrt{1-\beta_2}\eta_{\text{local}}}{N\epsilon^2} \left| \left[ \sum_{c \in \mathcal{C}_t} R_t^\top \mathbf{z}_{c,t} \right]_i \right| \leq \frac{\sqrt{1-\beta_2}\eta_{\text{local}}}{\epsilon^2} \cdot \frac{\sigma_g \log(2dT/\delta)}{\sqrt{N}} = \frac{\sqrt{1-\beta_2}\eta_{\text{local}}\sigma_g \log(2dT/\delta)}{\sqrt{N}\epsilon^2} \tag{34}$$

Substituting 32 and 34 into 30, we have that with probability at least $1 - \frac{2\delta}{T}$,

$$\left\| V_t^{-1/2} - V_{t-1}^{-1/2} \right\|_2$$

$$\leq \max_i \left\{ \frac{\sqrt{1-\beta_2}\eta_{\text{local}}}{N\epsilon^2} \left| \sum_{c \in \mathcal{C}_t} \left[ R^\top R \text{clip}(\frac{\Delta_{c,t}}{\eta_{\text{local}}}, \tau)) \right]_i \right| \right\} + \max_i \left\{ \frac{\sqrt{1-\beta_2}\eta_{\text{local}}}{N\epsilon^2} \left| \left[ \sum_{c \in \mathcal{C}_t} R^\top \mathbf{z}_{c,t} \right]_i \right| \right\}$$

$$\leq \frac{\sqrt{1-\beta_2}\eta_{\text{local}}\tau_{K,G}}{\epsilon^2} \left( 1 + \frac{\log^{1.5}(NTd^2/\delta)}{\sqrt{b}} \right) + \frac{\sqrt{1-\beta_2}\eta_{\text{local}}\sigma_g \log(2dT/\delta)}{\sqrt{N}\epsilon^2} \tag{35}$$

Substituting 35 into 28, we have that with probability at least $1 - \frac{3\delta}{T}$,

$$\left| \frac{1}{N}\nabla\mathcal{L}(\theta_t)^\top \left( V_t^{-1/2} - V_{t-1}^{-1/2} \right) \sum_{c \in \mathcal{C}_t} R_t^\top R_t \text{clip} \left( \sum_{k=1}^K g_{c,t,k}, \tau \right) \right|$$

$$\leq G\tau_{K,G} \left( 1 + \frac{\log^{1.5}(NTd^2/\delta)}{\sqrt{b}} \right) \left\| V_t^{-1/2} - V_{t-1}^{-1/2} \right\|_2$$

$$\leq G\tau_{K,G} \left( 1 + \frac{\log^{1.5}(NTd^2/\delta)}{\sqrt{b}} \right) \cdot \left( \frac{\sqrt{1-\beta_2}\eta_{\text{local}}\tau_{K,G}}{\epsilon^2} \left( 1 + \frac{\log^{1.5}(NTd^2/\delta)}{\sqrt{b}} \right) + \frac{\sqrt{1-\beta_2}\eta_{\text{local}}\sigma_g \log(2dT/\delta)}{\sqrt{N}\epsilon^2} \right)$$

$$= \frac{\eta_{\text{local}}\sqrt{1-\beta_2}G\tau_{K,G}^2}{\epsilon^2} \left( 1 + \frac{\log^{1.5}(NTd^2/\delta)}{\sqrt{b}} \right)^2 + \frac{\eta_{\text{local}}\sqrt{1-\beta_2}G\tau_{K,G}\sigma_g \log(2dT/\delta)}{\sqrt{N}\epsilon^2} \left( 1 + \frac{\log^{1.5}(NTd^2/\delta)}{\sqrt{b}} \right) \tag{36}$$

By taking summation from 0 to $T-1$, we can get that with probability at least $1 - 3\delta$,

$$\left| \sum_{t=1}^T \frac{1}{N}\nabla\mathcal{L}(\theta_t)^\top \left( V_t^{-1/2} - V_{t-1}^{-1/2} \right) \sum_{c \in \mathcal{C}_t} R_t^\top R_t \text{clip} \left( \sum_{k=1}^K g_{c,t,k}, \tau \right) \right|$$

$$\leq \frac{\eta_{\text{local}}\sqrt{1-\beta_2}TG\tau_{K,G}^2}{\epsilon^2} \left( 1 + \frac{\log^{1.5}(NTd^2/\delta)}{\sqrt{b}} \right)^2 + \frac{\eta_{\text{local}}\sqrt{1-\beta_2}TG\tau_{K,G}\sigma_g \log(2dT/\delta)}{\sqrt{N}\epsilon^2} \left( 1 + \frac{\log^{1.5}(NTd^2/\delta)}{\sqrt{b}} \right) \tag{37}$$

### E.1.2  Bounding $S_1$

We first bound each term with a fixed $t \in [T]$ in $S_1$. According to the definition of $m_{t-1}$, we have that

$$\nabla\mathcal{L}(\theta_t)^\top \frac{\beta_1 \eta_{\text{global}}}{1-\beta_1} \left( V_{t-1}^{-1/2} - V_t^{-1/2} \right) m_{t-1}$$

$$= \nabla\mathcal{L}(\theta_t)^\top \frac{\beta_1 \eta_{\text{global}}}{1-\beta_1} \left( V_{t-1}^{-1/2} - V_t^{-1/2} \right) \cdot \sum_{\tau'=1}^{t-1} (1-\beta_1)\beta_1^{t-1-\tau'} \text{desk}(\bar{\bar{\Delta}}_{\tau'})$$

$$= \nabla\mathcal{L}(\theta_t)^\top \beta_1 \eta_{\text{global}} \left( V_{t-1}^{-1/2} - V_t^{-1/2} \right) \cdot \sum_{\tau'=1}^{t-1} \beta_1^{t-1-\tau'} \frac{1}{N} \text{desk}(\bar{\bar{\Delta}}_{\tau'})$$

$$= \sum_{\tau'=0}^{t-1} \frac{\beta_1^{t-\tau'} \eta_{\text{global}}}{N} \nabla\mathcal{L}(\theta_t)^\top \left( V_{t-1}^{-1/2} - V_t^{-1/2} \right) R_{\tau'}^\top \sum_{c \in \mathcal{C}_{\tau'}} \tilde{\Delta}_{c,\tau'}$$

$$= \sum_{\tau'=0}^{t-1} \frac{\beta_1^{t-\tau'} \eta_{\text{global}}}{N} \nabla\mathcal{L}(\theta_t)^\top \left( V_{t-1}^{-1/2} - V_t^{-1/2} \right) R_\tau^\top \sum_{c \in \mathcal{C}_{\tau'}} \eta_{\text{local}} \left( \text{sk} \left( \text{clip} \left( \frac{\Delta_{c,\tau'}}{\eta_{\text{local}}}, \tau \right) \right) + \mathbf{z}_{c,\tau'} \right)$$

$$= \underbrace{\sum_{\tau'=0}^{t-1} \frac{\beta_1^{t-\tau'} \eta}{N} \nabla\mathcal{L}(\theta_t)^\top \left( V_{t-1}^{-1/2} - V_t^{-1/2} \right) \sum_{c \in \mathcal{C}_{\tau'}} R_{\tau'}^\top R_{\tau'} \text{clip} \left( \frac{\Delta_{c,\tau'}}{\eta_{\text{local}}}, \tau \right)}_{W_1}$$

$$+ \underbrace{\sum_{\tau'=1}^{t-1} \frac{\beta_1^{t-\tau'} \eta}{N} \nabla\mathcal{L}(\theta_t)^\top \left( V_{t-1}^{-1/2} - V_t^{-1/2} \right) \sum_{c \in \mathcal{C}_{\tau'}} R_{\tau'}^\top \mathbf{z}_{c,\tau'}}_{W_2} \tag{38}$$

**E.1.2.1 Bounding $W_1$**

We first bound each term with a fixed $t \in [T]$ in $W_1$. By applying the same analysis as above, we can get the same bound as 36, with probability at least $1 - \frac{3\delta}{T}$,

$$\frac{1}{N} \nabla\mathcal{L}(\theta_t)^\top \left( V_t^{-1/2} - V_{t-1}^{-1/2} \right) \sum_{c \in \mathcal{C}_{\tau'}} R_{\tau'}^\top R_{\tau'} \text{clip} \left( \sum_{k=1}^{K} g_{c,\tau',k}, \tau \right)$$

$$\leq \frac{\eta_{\text{local}} \sqrt{1-\beta_2} G \tau_{K,G}^2}{\epsilon^2} \left( 1 + \frac{\log^{1.5}(NTd^2/\delta)}{\sqrt{b}} \right)^2 + \frac{\eta_{\text{local}} \sqrt{1-\beta_2} G \tau_{K,G} \sigma_g \log(2dT/\delta)}{\sqrt{N} \epsilon^2} \left( 1 + \frac{\log^{1.5}(NTd^2/\delta)}{\sqrt{b}} \right)$$

By taking summation from 0 to $t-1$, we have that with probability at least $1 - 3\delta$,

$$\sum_{\tau'=1}^{t-1} \frac{\beta_1^{t-\tau'} \eta}{N} \nabla\mathcal{L}(\theta_t)^\top \left( V_{t-1}^{-1/2} - V_t^{-1/2} \right) \sum_{c \in \mathcal{C}_{\tau'}} R_{\tau'}^\top R_{\tau'} \text{clip} \left( \frac{\Delta_{c,\tau'}}{\eta_{\text{local}}}, \tau \right)$$

$$\leq \left( \frac{\eta\eta_{\text{local}} \sqrt{1-\beta_2} G \tau_{K,G}^2}{\epsilon^2} \left( 1 + \frac{\log^{1.5}(NTd^2/\delta)}{\sqrt{b}} \right)^2 + \frac{\eta\eta_{\text{local}} \sqrt{1-\beta_2} G \tau_{K,G} \sigma_g \log(2dT/\delta)}{\sqrt{N} \epsilon^2} \left( 1 + \frac{\log^{1.5}(NTd^2/\delta)}{\sqrt{b}} \right) \right) \sum_{\tau'=0}^{t-1} \beta_1^{t-\tau'}$$

$$\leq \frac{\eta\eta_{\text{local}} \beta_1 \sqrt{1-\beta_2} G \tau_{K,G}^2}{\epsilon^2 (1-\beta_1)} \left( 1 + \frac{\log^{1.5}(NTd^2/\delta)}{\sqrt{b}} \right)^2 + \frac{\eta\eta_{\text{local}} \beta_1 \sqrt{1-\beta_2} G \tau_{K,G} \sigma_g \log(2dT/\delta)}{\sqrt{N} \epsilon^2 (1-\beta_1)} \left( 1 + \frac{\log^{1.5}(NTd^2/\delta)}{\sqrt{b}} \right)$$
$$\tag{39}$$

**E.1.2.2 Bounding $W_2$**

We first bound each term with a fixed $t \in [T]$ in $W_2$.

$$\frac{1}{N} \nabla\mathcal{L}(\theta_t)^\top \left( V_{t-1}^{-1/2} - V_t^{-1/2} \right) \sum_{c \in \mathcal{C}_{\tau'}} R_{\tau'}^\top \mathbf{z}_{c,\tau'} = \left\langle R_{\tau'} \left( V_{t-1}^{-1/2} - V_t^{-1/2} \right)^\top \nabla\mathcal{L}(\theta_t), \frac{1}{N} \sum_{c \in \mathcal{C}_{\tau'}} R_{\tau'}^\top \mathbf{z}_{c,\tau'} \right\rangle.$$

Noticing that

$$\frac{1}{N} \left( V_{t-1}^{-1/2} - V_t^{-1/2} \right) \sum_{c \in \mathcal{C}_{\tau'}} \mathbf{z}_{c,\tau'} \sim \mathcal{N} \left( 0, \frac{\sigma_g^2}{N} \mathbb{I} \right),$$

and $R_{\tau'} \left( V_{t-1}^{-1/2} - V_t^{-1/2} \right)^\top \nabla\mathcal{L}(\theta_t)$ is a $\frac{\left\| V_{t-1}^{-1/2} - V_t^{-1/2} \right\|_2 \|\nabla\mathcal{L}(\theta)_t\|_2}{\sqrt{b}}$-sub-Gaussian random vector, so according to Bernstein inequality,

$$\mathbb{P} \left( \left\langle R_{\tau'} \left( V_{t-1}^{-1/2} - V_t^{-1/2} \right)^\top \nabla\mathcal{L}(\theta_t), \frac{1}{N} \sum_{c \in \mathcal{C}_{\tau'}} R_{\tau'}^\top \mathbf{z}_{c,\tau'} \right\rangle \geq a \right)$$

$$\leq 2\exp\left(-c\min\left(\frac{a^2}{b\cdot\frac{\sigma_g^2}{N}\cdot\frac{\left\|V_{t-1}^{-1/2}-V_t^{-1/2}\right\|_2^2\|\nabla\mathcal{L}(\theta)_t\|_2^2}{b}},\frac{a}{\frac{\sigma_g}{\sqrt{N}}\cdot\frac{\left\|V_{t-1}^{-1/2}-V_t^{-1/2}\right\|_2\|\nabla\mathcal{L}(\theta)_t\|_2}{\sqrt{b}}}\right)\right)$$

$$=2\exp\left(-c\min\left(\frac{Na^2}{\sigma_g^2\left\|V_{t-1}^{-1/2}-V_t^{-1/2}\right\|_2^2\|\nabla\mathcal{L}(\theta)_t\|_2^2},\frac{a\sqrt{bN}}{\sigma_g\left\|V_{t-1}^{-1/2}-V_t^{-1/2}\right\|_2\|\nabla\mathcal{L}(\theta)_t\|_2}\right)\right)$$

so taking $a=\frac{\sigma_g\left\|V_{t-1}^{-1/2}-V_t^{-1/2}\right\|_2\|\nabla\mathcal{L}(\theta)_t\|_2\log(2T/\delta)}{\sqrt{N}}$, and combining with 35, we have that with probability at least $1-\frac{3\delta}{T}$,

$$\left\langle R_{\tau'}\left(V_{t-1}^{-1/2}-V_t^{-1/2}\right)^\top\nabla\mathcal{L}(\theta_t),\frac{1}{N}\sum_{c\in\mathcal{C}_{\tau'}}R_{\tau'}^\top\mathbf{z}_{c,\tau'}\right\rangle$$

$$\leq\frac{\sigma_g\left\|V_{t-1}^{-1/2}-V_t^{-1/2}\right\|_2\|\nabla\mathcal{L}(\theta_t)\|_2\log(2T/\delta)}{\sqrt{N}}$$

$$\leq\frac{\sigma_gG\log(2T/\delta)}{\sqrt{N}}\cdot\left(\frac{\sqrt{1-\beta_2}\eta_{\text{local}}\tau_{K,G}}{\epsilon^2}\left(1+\frac{\log^{1.5}(NTd^2/\delta)}{\sqrt{b}}\right)+\frac{\sqrt{1-\beta_2}\eta_{\text{local}}\sigma_g\log(2dT/\delta)}{\sqrt{N}\epsilon^2}\right)$$

$$=\frac{\eta_{\text{local}}\sqrt{1-\beta_2}\sigma_gG\tau_{K,G}\log(2T/\delta)}{\sqrt{N}\epsilon^2}\left(1+\frac{\log^{1.5}(NTd^2/\delta)}{\sqrt{b}}\right)+\frac{\eta_{\text{local}}\sqrt{1-\beta_2}\sigma_g^2G\log(2T/\delta)\log(2dT/\delta)}{N\epsilon^2}$$

By taking summation from 0 to $t-1$, we have that with probability at least $1-3\delta$,

$$\sum_{\tau'=1}^{t-1}\frac{\beta_1^{t-\tau'}\eta}{N}\nabla\mathcal{L}(\theta_t)^\top\left(V_{t-1}^{-1/2}-V_t^{-1/2}\right)\sum_{c\in\mathcal{C}_{\tau'}}R_{\tau'}^\top\mathbf{z}_{c,\tau'}$$

$$\leq\left(\frac{\eta\eta_{\text{local}}\sqrt{1-\beta_2}\sigma_gG\tau_{K,G}\log(2T/\delta)}{\sqrt{N}\epsilon^2}\left(1+\frac{\log^{1.5}(NTd^2/\delta)}{\sqrt{b}}\right)+\frac{\eta\eta_{\text{local}}\sqrt{1-\beta_2}\sigma_g^2G\log(2T/\delta)\log(2dT/\delta)}{N\epsilon^2}\right)\sum_{\tau'=0}^{t-1}\beta_1^{t-\tau'}$$

$$\leq\frac{\eta\eta_{\text{local}}\beta_1\sqrt{1-\beta_2}\sigma_gG\tau_{K,G}\log(2T/\delta)}{\sqrt{N}\epsilon^2(1-\beta_1)}\left(1+\frac{\log^{1.5}(NTd^2/\delta)}{\sqrt{b}}\right)+\frac{\eta\eta_{\text{local}}\beta_1\sqrt{1-\beta_2}\sigma_g^2G\log(2T/\delta)\log(2dT/\delta)}{N\epsilon^2(1-\beta_1)}$$

$$\tag{40}$$

Substituting 39 and 40 into 38, we can get that with probability at least $1-4\delta$,

$$\nabla\mathcal{L}(\theta_t)^\top\frac{\beta_1\eta_{\text{global}}}{1-\beta_1}\left(V_{t-1}^{-1/2}-V_t^{-1/2}\right)m_{t-1}$$

$$=\sum_{\tau'=0}^{t-1}\frac{\beta_1^{t-\tau'}\eta}{N}\nabla\mathcal{L}(\theta_t)^\top\left(V_{t-1}^{-1/2}-V_t^{-1/2}\right)\sum_{c\in\mathcal{C}_{\tau'}}R_{\tau'}^\top R_{\tau'}\text{clip}\left(\frac{\Delta_{c,\tau'}}{\eta_{\text{local}}},\tau\right)$$

$$+\sum_{\tau'=1}^{t-1}\frac{\beta_1^{t-\tau'}\eta}{N}\nabla\mathcal{L}(\theta_t)^\top\left(V_{t-1}^{-1/2}-V_t^{-1/2}\right)\sum_{c\in\mathcal{C}_{\tau'}}R_{\tau'}^\top\mathbf{z}_{c,\tau'}$$

$$\leq\frac{\eta\eta_{\text{local}}\beta_1\sqrt{1-\beta_2}G\tau_{K,G}^2}{\epsilon^2(1-\beta_1)}\left(1+\frac{\log^{1.5}(NTd^2/\delta)}{\sqrt{b}}\right)^2+\frac{\eta\eta_{\text{local}}\beta_1\sqrt{1-\beta_2}G\tau_{K,G}\sigma_g\log(2dT/\delta)}{\sqrt{N}\epsilon^2(1-\beta_1)}\left(1+\frac{\log^{1.5}(NTd^2/\delta)}{\sqrt{b}}\right)$$

$$+\frac{\eta\eta_{\text{local}}\beta_1\sqrt{1-\beta_2}\sigma_gG\tau_{K,G}\log(2T/\delta)}{\sqrt{N}\epsilon^2(1-\beta_1)}\left(1+\frac{\log^{1.5}(NTd^2/\delta)}{\sqrt{b}}\right)+\frac{\eta\eta_{\text{local}}\beta_1\sqrt{1-\beta_2}\sigma_g^2G\log(2T/\delta)\log(2dT/\delta)}{N\epsilon^2(1-\beta_1)}$$

$$\leq\frac{\eta\eta_{\text{local}}\beta_1\sqrt{1-\beta_2}G\tau_{K,G}^2}{\epsilon^2(1-\beta_1)}\left(1+\frac{\log^{1.5}(NTd^2/\delta)}{\sqrt{b}}\right)^2+\frac{2\eta\eta_{\text{local}}\beta_1\sqrt{1-\beta_2}G\tau_{K,G}\sigma_g\log(2dT/\delta)}{\sqrt{N}\epsilon^2(1-\beta_1)}\left(1+\frac{\log^{1.5}(NTd^2/\delta)}{\sqrt{b}}\right)$$

$$+\frac{\eta\eta_{\text{local}}\beta_1\sqrt{1-\beta_2}\sigma_g^2G\log^2(2dT/\delta)}{N\epsilon^2(1-\beta_1)}$$

By taking summation from 0 to $T - 1$, we can get that with probability at least $1 - 4\delta$,

$$\sum_{t=1}^{T} \nabla\mathcal{L}(\theta_t)^{\top} \frac{\beta_1 \eta_{\text{global}}}{1 - \beta_1} \left(V_{t-1}^{-1/2} - V_t^{-1/2}\right) m_{t-1}$$

$$\leq \frac{\eta\eta_{\text{local}}\beta_1\sqrt{1-\beta_2}TG\tau_{K,G}^2}{\epsilon^2(1-\beta_1)} \left(1 + \frac{\log^{1.5}(NTd^2/\delta)}{\sqrt{b}}\right)^2 + \frac{2\eta\eta_{\text{local}}\beta_1\sqrt{1-\beta_2}TG\tau_{K,G}\sigma_g\log(2dT/\delta)}{\sqrt{N}\epsilon^2(1-\beta_1)} \left(1 + \frac{\log^{1.5}(NTd^2/\delta)}{\sqrt{b}}\right)$$

$$+ \frac{\eta\eta_{\text{local}}\beta_1\sqrt{1-\beta_2}T\sigma_g^2G\log^2(2dT/\delta)}{N\epsilon^2(1-\beta_1)} \tag{41}$$

### E.1.3 Bounding $S_4$

We first bound each term with a fixed $t \in [T]$ in $S_4$. $\frac{1}{N}\nabla\mathcal{L}(\theta_t)^{\top}V_t^{-1/2}\sum_{c\in\mathcal{C}_t} R_t^{\top}\mathbf{z}_{c,t} = \left\langle R_t V_t^{-1/2}\nabla\mathcal{L}(\theta_t), \frac{1}{N}\sum_{c\in\mathcal{C}_t}\mathbf{z}_{c,t}\right\rangle$. Noticing that $\frac{1}{N}\sum_{c\in\mathcal{C}_t}\mathbf{z}_{c,t} \sim \mathcal{N}\left(0, \frac{\sigma_g^2}{N}\mathbb{I}\right)$, and $R_t V_t^{-1/2}\nabla\mathcal{L}(\theta_t)$ is a $\frac{\|\nabla\mathcal{L}(\theta_t)\|_2\|V_t^{-1/2}\|}{\sqrt{b}}$-sub-Gaussian random vector, so according to Bernstein inequality,

$$\mathbb{P}\left(\left|\left\langle R_t V_t^{-1/2}\nabla\mathcal{L}(\theta_t), \frac{1}{N}\sum_{c\in\mathcal{C}_t}\mathbf{z}_{c,t}\right\rangle\right| \geq a\right)$$

$$\leq 2\exp\left(-c\min\left(\frac{a^2}{b\cdot\frac{\sigma_g^2}{N}\cdot\frac{\|\nabla\mathcal{L}(\theta_t)\|_2^2\|V_t^{-1/2}\|_2^2}{b}}, \frac{a}{\frac{\sigma_g}{\sqrt{N}}\cdot\frac{\|\nabla\mathcal{L}(\theta_t)\|_2\|V_t^{-1/2}\|_2}{\sqrt{b}}}\right)\right)$$

$$= 2\exp\left(-c\min\left(\frac{Na^2}{\sigma_g^2\|\nabla\mathcal{L}(\theta_t)\|_2^2\|V_t^{-1/2}\|_2^2}, \frac{a\sqrt{bN}}{\sigma_g\|\nabla\mathcal{L}(\theta_t)\|_2\|V_t^{-1/2}\|_2}\right)\right)$$

so taking $a = \frac{\sigma_g\|\nabla\mathcal{L}(\theta_t)\|_2\|V_t^{-1/2}\|_2\log(2T/\delta)}{\sqrt{N}}$, and noticing that $\|V_t^{-1/2}\|_2 \leq \frac{1}{\epsilon}$, we have that with probability at least $1 - \frac{\delta}{T}$,

$$\left|\left\langle R_t V_t^{-1/2}\nabla\mathcal{L}(\theta_t), \frac{1}{N}\sum_{c\in\mathcal{C}_t}\mathbf{z}_{c,t}\right\rangle\right| \leq \frac{\sigma_g\|\nabla\mathcal{L}(\theta_t)\|_2\log(2T/\delta)}{\sqrt{N}\epsilon} \leq \frac{\sigma_g G\log(2T/\delta)}{\sqrt{N}\epsilon}$$

Then denote $X_t = \sum_{\tau'=0}^{t}\left\langle \nabla\mathcal{L}(\theta_{\tau'}), \frac{1}{N}V_{\tau'}^{-1/2}\sum_{c\in\mathcal{C}_{\tau'}} R_{\tau'}^{\top}\mathbf{z}_{c,\tau'}\right\rangle$, we can see that $X_t$ is a martingale with respect to the Gaussian noise, and from the above analysis, we have that with probability at least $1 - \delta$, for all $t \in [T]$,

$$|X_t - X_{t-1}| = \left|\left\langle\nabla\mathcal{L}(\theta_t), \frac{1}{N}V_t^{-1/2}\sum_{c\in\mathcal{C}_t} R_t^{\top}\mathbf{z}_{c,t}\right\rangle\right| \leq \frac{\sigma_g G\log(2T/\delta)}{\sqrt{N}\epsilon}$$

Then by Azuma's inequality, we have

$$\mathbb{P}\left(X_{T-1} \leq -a\right) \leq \exp\left(-\frac{a^2}{2\cdot T\cdot\left(\frac{\sigma_g G\log(2T/\delta)}{\sqrt{N}\epsilon}\right)^2}\right) = \exp\left(-\frac{N\epsilon^2 a^2}{2T\sigma_g^2 G^2\log^2(2T/\delta)}\right)$$

By selecting $a = \frac{\log^2(2T/\delta)\sqrt{2T}\sigma_g G}{\sqrt{N}\epsilon}$, we can get that with probability at least $1 - 2\delta$,

$$X_{T-1} = \sum_{t=1}^{T}\left\langle\nabla\mathcal{L}(\theta_t), \frac{1}{N}V_t^{-1/2}\sum_{c\in\mathcal{C}_t} R_t^{\top}\mathbf{z}_{c,t}\right\rangle \geq -\frac{\log^2(2T/\delta)\sqrt{2T}\sigma_g G}{\sqrt{N}\epsilon} \tag{42}$$

### E.1.4 Bounding $S_3$

For each term in $S_1$, we have that

$$\frac{1}{N}\nabla\mathcal{L}(\theta_t)^\top V_{t-1}^{-1/2}\sum_{c\in\mathcal{C}_t}R_t^\top R_t\text{clip}\left(\sum_{k=1}^{K}g_{c,t,k},\tau\right)$$

$$=K\left\langle\nabla\mathcal{L}(\theta_t),V_{t-1}^{-1/2}\frac{1}{C}\sum_{c=1}^{C}\nabla\mathcal{L}_c(\theta_t)\right\rangle+K\left\langle\nabla\mathcal{L}(\theta_t),V_{t-1}^{-1/2}\left(\frac{1}{N}\sum_{c\in\mathcal{C}_t}\nabla\mathcal{L}_c(\theta_t)-\frac{1}{C}\sum_{i=1}^{C}\nabla\mathcal{L}_c(\theta_t)\right)\right\rangle$$

$$+\left\langle\nabla\mathcal{L}(\theta_t),\frac{1}{N}V_{t-1}^{-1/2}\sum_{c\in\mathcal{C}_t}\sum_{k=1}^{K}(\nabla\mathcal{L}_c(\theta_{c,t,k})-\nabla\mathcal{L}_c(\theta_t))\right\rangle+\left\langle\nabla\mathcal{L}(\theta_t),\frac{1}{N}V_{t-1}^{-1/2}\sum_{c\in\mathcal{C}_t}\sum_{k=1}^{K}(g_{c,t,k}-\nabla\mathcal{L}_c(\theta_{c,t,k}))\right\rangle$$

$$+\left\langle\nabla\mathcal{L}(\theta_t),\frac{1}{N}V_{t-1}^{-1/2}\sum_{c\in\mathcal{C}_t}\left(\text{clip}\left(\sum_{k=1}^{K}g_{c,t,k},\tau\right)-\sum_{k=1}^{K}g_{c,t,k}\right)\right\rangle$$

$$+\left\langle\nabla\mathcal{L}(\theta_t),\frac{1}{N}V_{t-1}^{-1/2}\sum_{c\in\mathcal{C}_t}\left(R_t^\top R_t\text{clip}\left(\sum_{k=1}^{K}g_{c,t,k},\tau\right)-\text{clip}\left(\sum_{k=1}^{K}g_{c,t,k},\tau\right)\right)\right\rangle$$

By taking summation from $0$ to $T-1$, we can get that

$$\sum_{t=1}^{T}\frac{1}{N}\nabla\mathcal{L}(\theta_t)^\top V_{t-1}^{-1/2}\sum_{c\in\mathcal{C}_t}R_t^\top R_t\text{clip}\left(\sum_{k=1}^{K}g_{c,t,k},\tau\right)$$

$$=\underbrace{K\sum_{t=1}^{T}\left\langle\nabla\mathcal{L}(\theta_t),V_{t-1}^{-1/2}\frac{1}{C}\sum_{c=1}^{C}\nabla\mathcal{L}_c(\theta_t)\right\rangle}_{Y_1}+\underbrace{K\sum_{t=1}^{T}\left\langle\nabla\mathcal{L}(\theta_t),V_{t-1}^{-1/2}\left(\frac{1}{N}\sum_{c\in\mathcal{C}_t}\nabla\mathcal{L}_c(\theta_t)-\frac{1}{C}\sum_{i=1}^{C}\nabla\mathcal{L}_c(\theta_t)\right)\right\rangle}_{Y_2}$$

$$+\underbrace{\sum_{t=1}^{T}\left\langle\nabla\mathcal{L}(\theta_t),\frac{1}{N}V_{t-1}^{-1/2}\sum_{c\in\mathcal{C}_t}\sum_{k=1}^{K}(\nabla\mathcal{L}_c(\theta_{c,t,k})-\nabla\mathcal{L}_c(\theta_t))\right\rangle}_{Y_3}$$

$$+\underbrace{\sum_{t=1}^{T}\left\langle\nabla\mathcal{L}(\theta_t),\frac{1}{N}V_{t-1}^{-1/2}\sum_{c\in\mathcal{C}_t}\sum_{k=1}^{K}(g_{c,t,k}-\nabla\mathcal{L}_c(\theta_{c,t,k}))\right\rangle}_{Y_4}$$

$$+\underbrace{\sum_{t=1}^{T}\left\langle\nabla\mathcal{L}(\theta_t),\frac{1}{N}V_{t-1}^{-1/2}\sum_{c\in\mathcal{C}_t}\left(\text{clip}\left(\sum_{k=1}^{K}g_{c,t,k},\tau\right)-\sum_{k=1}^{K}g_{c,t,k}\right)\right\rangle}_{Y_5}$$

$$+\underbrace{\sum_{t=1}^{T}\left\langle\nabla\mathcal{L}(\theta_t),\frac{1}{N}V_{t-1}^{-1/2}\sum_{c\in\mathcal{C}_t}\left(R_t^\top R_t\text{clip}\left(\sum_{k=1}^{K}g_{c,t,k},\tau\right)-\text{clip}\left(\sum_{k=1}^{K}g_{c,t,k},\tau\right)\right)\right\rangle}_{Y_6}\tag{43}$$

### E.1.4.1 Bounding $Y_1$

According to the definition of $V_{t-1}$, we have that

$$\left[V_{t-1}^{-1/2}\right]_i=\left[\left(\sqrt{v_{t-1}}+\epsilon\right)^{-1}\right]_i=\left(\sqrt{v_{t-1,i}}+\epsilon\right)^{-1}$$

Substituting 31 and 33 into 29, we can see that with probability at least $1-\frac{2\delta}{d}$, for any $t\in[T]$,

$$\sqrt{\left[\left(\text{desk}(\bar{\bar{\Delta}}_\tau)\right)^2\right]_i}=\left|\left[\frac{\eta_{\text{local}}}{N}\sum_{c\in\mathcal{C}_t}\left(R^\top R\text{clip}(\frac{\Delta_{c,t}}{\eta_{\text{local}}},\tau))+R^\top\mathbf{z}_{c,t}\right)\right]_i\right|$$

$$\leq \frac{\eta_{\text{local}}}{N} \left| \sum_{c \in \mathcal{C}_t} \left[ R_t^\top R_t \text{clip}\left(\frac{\Delta_{c,t}}{\eta_{\text{local}}}, \tau\right)\right]_i \right| + \frac{\eta_{\text{local}}}{N} \left| \left[\sum_{c \in \mathcal{C}_t} R_t^\top \mathbf{z}_{c,t}\right]_i \right|$$

$$\leq \eta_{\text{local}} \left(1 + \frac{\log^{1.5}(NTd^2/\delta)}{\sqrt{b}}\right) \tau_{K,G} + \frac{\eta_{\text{local}}\sigma_g \log(2dT/\delta)}{\sqrt{N}}$$

Then according to the definition of second order moment, we have that with probability at least $1 - \frac{2\delta}{d}$,

$$\sqrt{v_{t-1,i}} \leq \max_{\tau \leq t-1} \sqrt{\left[\left(\text{desk}(\bar{\bar{\Delta}}_\tau)\right)^2\right]_i} \leq \eta_{\text{local}} \left(1 + \frac{\log^{1.5}(NTd^2/\delta)}{\sqrt{b}}\right) \tau_{K,G} + \frac{\eta_{\text{local}}\sigma_g \log(2dT/\delta)}{\sqrt{N}}$$

so we can get that with probability at least $1 - 2\delta$, for all $i \in [d]$,

$$\left[V_{t-1}^{-1/2}\right]_i = \left(\sqrt{v_{t-1,i}} + \epsilon\right)^{-1} \geq \left(\eta_{\text{local}} \left(1 + \frac{\log^{1.5}(NTd^2/\delta)}{\sqrt{b}}\right) \tau_{K,G} + \frac{\eta_{\text{local}}\sigma_g \log(2dT/\delta)}{\sqrt{N}} + \epsilon\right)^{-1}$$

which implies that with probability

$$\left\langle \nabla\mathcal{L}(\theta_t), V_{t-1}^{-1/2} \frac{1}{C} \sum_{c=1}^C \nabla\mathcal{L}_c(\theta_t) \right\rangle$$

$$\geq \left(\eta_{\text{local}} \left(1 + \frac{\log^{1.5}(NTd^2/\delta)}{\sqrt{b}}\right) \tau_{K,G} + \frac{\eta_{\text{local}}\sigma_g \log(2dT/\delta)}{\sqrt{N}} + \epsilon\right)^{-1} \nabla\mathcal{L}(\theta_t)^\top \frac{1}{C} \sum_{c=1}^C \nabla\mathcal{L}_c(\theta_t)$$

$$= \left(\eta_{\text{local}} \left(1 + \frac{\log^{1.5}(NTd^2/\delta)}{\sqrt{b}}\right) \tau_{K,G} + \frac{\eta_{\text{local}}\sigma_g \log(2dT/\delta)}{\sqrt{N}} + \epsilon\right)^{-1} \|\nabla\mathcal{L}(\theta_t)\|_2^2$$

so with probability at least $1 - 2\delta$,

$$\sum_{t=1}^T \left\langle \nabla\mathcal{L}(\theta_t), V_{t-1}^{-1/2} \frac{1}{C} \sum_{c=1}^C \nabla\mathcal{L}_c(\theta_t) \right\rangle \geq \left(\eta_{\text{local}} \left(1 + \frac{\log^{1.5}(NTd^2/\delta)}{\sqrt{b}}\right) \tau_{K,G} + \frac{\eta_{\text{local}}\sigma_g \log(2dT/\delta)}{\sqrt{N}} + \epsilon\right)^{-1} \sum_{t=1}^T \|\nabla\mathcal{L}(\theta_t)\|_2^2$$

(44)

### E.1.4.2 Bounding $Y_2$

We first bound each term with a fixed $t \in [T]$ in $Y_2$. According to the assumption, each $c \in \mathcal{C}_t$ is uniformly randomly selected from $[C]$, so by Hoeffding's inequality, we have

$$\mathbb{P}\left(\left|\left\langle \nabla\mathcal{L}(\theta_t), V_{t-1}^{-1/2}\left(\frac{1}{N}\sum_{c \in \mathcal{C}_t} \nabla\mathcal{L}_c(\theta_t) - \frac{1}{C}\sum_{i=1}^C \nabla\mathcal{L}_c(\theta_t)\right)\right\rangle\right| \geq a\right)$$

$$\leq 2\exp\left(-\frac{2Nt^2}{\left(\frac{2G^2}{\epsilon}\right)^2}\right) = 2\exp\left(-\frac{N\epsilon^2 t^2}{2G^4}\right)$$

By selecting $a = \frac{\sqrt{2\log(2T/\delta)}G^2}{\sqrt{N}\epsilon}$, we have that with probability at least $1 - \frac{\delta}{T}$,

$$\left|\left\langle \nabla\mathcal{L}(\theta_t), V_{t-1}^{-1/2}\left(\frac{1}{N}\sum_{c \in \mathcal{C}_t} \nabla\mathcal{L}_c(\theta_t) - \frac{1}{C}\sum_{i=1}^C \nabla\mathcal{L}_c(\theta_t)\right)\right\rangle\right| \leq \frac{\sqrt{2\log(2T/\delta)}G^2}{\sqrt{N}\epsilon}$$

Then denote $Z_t = \sum_{\tau'=0}^t \left\langle \nabla\mathcal{L}(\theta_{\tau'}), V_{t-1}^{-1/2}\left(\frac{1}{N}\sum_{c \in \mathcal{C}_{\tau'}} \nabla\mathcal{L}_c(\theta_{\tau'}) - \frac{1}{C}\sum_{i=1}^C \nabla\mathcal{L}_c(\theta_{\tau'})\right)\right\rangle$, we can see that $Z_t$ is a martingale with respect to the selection each round, and from the above analysis, we have that with probability at least $1 - \delta$, for all $t \in [T]$,

$$|Z_t - Z_{t-1}| = \left|\left\langle \nabla\mathcal{L}(\theta_t), V_{t-1}^{-1/2}\left(\frac{1}{N}\sum_{c \in \mathcal{C}_t} \nabla\mathcal{L}_c(\theta_t) - \frac{1}{C}\sum_{i=1}^C \nabla\mathcal{L}_c(\theta_t)\right)\right\rangle\right| \leq \frac{\sqrt{2\log(2T/\delta)}G^2}{\sqrt{N}\epsilon}$$

Then by Azuma's inequality, we have

$$\mathbb{P}\left(Z_{T-1} \leq -a\right) \leq \exp\left(-\frac{a^2}{2 \cdot T \cdot \left(\frac{\sqrt{2\log(2T/\delta)G^2}}{\sqrt{N}\epsilon}\right)^2}\right) = \exp\left(-\frac{N\epsilon^2 a^2}{4TG^4\log(2T/\delta)}\right)$$

By selecting $t = \frac{2\log(2T/\delta)\sqrt{T}G^2}{\sqrt{N}\epsilon}$, we can get that with probability at least $1 - 2\delta$,

$$Z_{T-1} = \sum_{t=1}^{T}\left\langle \nabla\mathcal{L}(\theta_t), \frac{1}{N}\sum_{c\in\mathcal{C}_t}\nabla\mathcal{L}_c(\theta_t) - \frac{1}{C}\sum_{i=1}^{C}\nabla\mathcal{L}_c(\theta_t)\right\rangle \geq -\frac{2\log(2T/\delta)\sqrt{T}G^2}{\sqrt{N}\epsilon} \tag{45}$$

### E.1.4.3 Bounding $Y_3$

For each term, we have

$$\left\langle \nabla\mathcal{L}(\theta_t), \frac{1}{N}V_{t-1}^{-1/2}\sum_{c\in\mathcal{C}_t}\sum_{k=1}^{K}\left(\nabla\mathcal{L}_c(\theta_{c,t,k}) - \nabla\mathcal{L}_c(\theta_t)\right)\right\rangle$$

$$=\frac{1}{N}\sum_{c\in\mathcal{C}_t}\sum_{k=1}^{K}\left\langle \nabla\mathcal{L}(\theta_t), V_{t-1}^{-1/2}\hat{H}_{\mathcal{L}}^{c,t,k}\left(\theta_{c,t,k} - \theta_t\right)\right\rangle$$

$$=\frac{\eta_{\text{local}}}{N}\sum_{c\in\mathcal{C}_t}\sum_{k=1}^{K}\left\langle \nabla\mathcal{L}(\theta_t), V_{t-1}^{-1/2}\hat{H}_{\mathcal{L}}^{c,t,k}\sum_{\kappa=1}^{k}g_{c,t,\kappa}\right\rangle$$

$$\geq -\frac{\eta_{\text{local}}}{N}\sum_{c\in\mathcal{C}_t}\sum_{k=1}^{K}\|\nabla\mathcal{L}(\theta_t)\|_2 \cdot \frac{L}{\epsilon}\sum_{\kappa=1}^{k}\|g_{c,t,\kappa}\|_2$$

$$= -\frac{\eta_{\text{local}}}{N}\cdot N\cdot G^2\cdot\frac{L}{\epsilon}\sum_{k=1}^{K}k$$

$$\geq -\frac{\eta_{\text{local}}LK^2G^2}{2\epsilon}$$

so

$$\sum_{t=1}^{T}\left\langle \nabla\mathcal{L}(\theta_t), \frac{1}{N}V_{t-1}^{-1/2}\sum_{c\in\mathcal{C}_t}\sum_{k=1}^{K}\left(\nabla\mathcal{L}_c(\theta_{c,t,k}) - \nabla\mathcal{L}_c(\theta_t)\right)\right\rangle \geq -\frac{\eta_{\text{local}}TLK^2G^2}{2\epsilon} \tag{46}$$

### E.1.4.4 Bounding $Y_4$

We first bound each term with a fixed $t \in [T]$ in $Y_4$.

$$\left|\left\langle \nabla\mathcal{L}(\theta_t), \frac{1}{N}V_{t-1}^{-1/2}\sum_{c\in\mathcal{C}_t}\sum_{k=1}^{K}\left(g_{c,t,k} - \nabla\mathcal{L}_c(\theta_{c,t,k})\right)\right\rangle\right|$$

$$\leq \frac{1}{N}\sum_{c\in\mathcal{C}_t}\sum_{k=1}^{K}\|\nabla\mathcal{L}(\theta_t)\|_2 \cdot \frac{1}{\epsilon}\|g_{c,t,k} - \nabla\mathcal{L}_c(\theta_{c,t,k})\|_2$$

$$=\frac{G}{N\epsilon}\sum_{c\in\mathcal{C}_t}\sum_{k=1}^{K}\|g_{c,t,k} - \nabla\mathcal{L}_c(\theta_{c,t,k})\|_2$$

According to the assumption, the stochastic noise $\|g_{c,t,k} - \nabla\mathcal{L}_c(\theta_{c,t,k})\|_2$ is a $\sigma_s$-sub-Gaussian random variable, so by Hoeffding's inequality,

$$\mathbb{P}\left(\sum_{c\in\mathcal{C}_t}\sum_{k=1}^{K}\|g_{c,t,k} - \nabla\mathcal{L}_c(\theta_{c,t,k})\|_2 \geq a\right) \leq 2\exp\left(-\frac{a^2}{NK\sigma_s^2}\right)$$

By selecting $a = \sqrt{NK\log(2T/\delta)}\sigma_s$, we have that with probability at least $1 - \frac{\delta}{T}$,

$$\sum_{c\in\mathcal{C}_t}\sum_{k=1}^{K}\|g_{c,t,k} - \nabla\mathcal{L}_c(\theta_{c,t,k})\|_2 \le \sqrt{NK\log(2T/\delta)}\sigma_s$$

so

$$\left|\left\langle\nabla\mathcal{L}(\theta_t), \frac{1}{N}V_{t-1}^{-1/2}\sum_{c\in\mathcal{C}_t}\sum_{k=1}^{K}(g_{c,t,k} - \nabla\mathcal{L}_c(\theta_{c,t,k}))\right\rangle\right|$$

$$\le \frac{G}{N\epsilon}\sum_{c\in\mathcal{C}_t}\sum_{k=1}^{K}\|g_{c,t,k} - \nabla\mathcal{L}_c(\theta_{c,t,k})\|_2$$

$$\le \frac{G}{N\epsilon}\cdot\sqrt{NK\log(2T/\delta)}\sigma_s = \frac{G\sqrt{K\log(2T/\delta)}\sigma_s}{\sqrt{N}\epsilon}$$

Then denote $W_t = \sum_{\tau'=0}^{t}\left\langle\nabla\mathcal{L}(\theta'_\tau), \frac{1}{N}V_{\tau'-1}^{-1/2}\sum_{c\in\mathcal{C}_{\tau'}}\sum_{k=1}^{K}(g_{c,\tau',k} - \nabla\mathcal{L}_c(\theta_{c,\tau',k}))\right\rangle$, we can see that $W_t$ is a martingale with respect to the stochastic noise, and from the above analysis, we have that with probability at least $1 - \delta$, for all $t \in [T]$,

$$|W_t - W_{t-1}| = \left|\left\langle\nabla\mathcal{L}(\theta_t), \frac{1}{N}V_{t-1}^{-1/2}\sum_{c\in\mathcal{C}_t}\sum_{k=1}^{K}(g_{c,t,k} - \nabla\mathcal{L}_c(\theta_{c,t,k}))\right\rangle\right| \le \frac{G\sqrt{K\log(2T/\delta)}\sigma_s}{\sqrt{N}\epsilon}$$

Then by Azuma's inequality, we have

$$\mathbb{P}\left(W_{T-1} \le -a\right) \le \exp\left(-\frac{a^2}{2\cdot T\cdot\left(\frac{G\sqrt{K\log(2T/\delta)}\sigma_s}{\sqrt{N}\epsilon}\right)^2}\right) = \exp\left(-\frac{N\epsilon^2 a^2}{2TG^2 K\log(2T/\delta)\sigma_s^2}\right)$$

By selecting $a = \frac{G\sqrt{2TK}\log(2T/\delta)\sigma_s}{\sqrt{N}\epsilon}$, we can get that with probability at least $1 - 2\delta$,

$$W_{T-1} = \sum_{t=1}^{T}\left\langle\nabla\mathcal{L}(\theta_t), \frac{1}{N}V_{t-1}^{-1/2}\sum_{c\in\mathcal{C}_t}\sum_{k=1}^{K}(g_{c,t,k} - \nabla\mathcal{L}_c(\theta_{c,t,k}))\right\rangle \ge -\frac{G\sqrt{2TK}\log(2T/\delta)\sigma_s}{\sqrt{N}\epsilon}$$

$$(47)$$

### E.1.4.5 Bounding $Y_5$

For each term, for $\tau \le KG$, we have

$$\left\langle\nabla\mathcal{L}(\theta_t), \frac{1}{N}V_{t-1}^{-1/2}\sum_{c\in\mathcal{C}_t}\left(\text{clip}\left(\sum_{k=1}^{K}g_{c,t,k}, \tau\right) - \sum_{k=1}^{K}g_{c,t,k}\right)\right\rangle$$

$$\ge -\frac{1}{N}\sum_{c\in\mathcal{C}_t}\|\nabla\mathcal{L}(\theta_t)\|_2\left\|V_{t-1}^{-1/2}\sum_{c\in\mathcal{C}_t}\left(\text{clip}\left(\sum_{k=1}^{K}g_{c,t,k}, \tau\right) - \sum_{k=1}^{K}g_{c,t,k}\right)\right\|_2$$

$$\ge -\frac{G(KG - \tau)}{\epsilon}$$

for $\tau \ge KG$, we have

$$\left\langle\nabla\mathcal{L}(\theta_t), \frac{1}{N}V_{t-1}^{-1/2}\sum_{c\in\mathcal{C}_t}\left(\text{clip}\left(\sum_{k=1}^{K}g_{c,t,k}, \tau\right) - \sum_{k=1}^{K}g_{c,t,k}\right)\right\rangle = 0$$

so

$$\left\langle\nabla\mathcal{L}(\theta_t), \frac{1}{N}V_{t-1}^{-1/2}\sum_{c\in\mathcal{C}_t}\left(\text{clip}\left(\sum_{k=1}^{K}g_{c,t,k}, \tau\right) - \sum_{k=1}^{K}g_{c,t,k}\right)\right\rangle \ge -\max\left\{0, \frac{G(KG - \tau)}{\epsilon}\right\}$$

By taking summation from $0$ to $T-1$, we can get that

$$\sum_{t=1}^{T}\left\langle\nabla\mathcal{L}(\theta_t),\frac{1}{N}V_{t-1}^{-1/2}\sum_{c\in\mathcal{C}_t}\left(\mathrm{clip}\left(\sum_{k=1}^{K}g_{c,t,k},\tau\right)-\sum_{k=1}^{K}g_{c,t,k}\right)\right\rangle\geq-\max\left\{0,\frac{TG(KG-\tau)}{\epsilon}\right\}$$

(48)

### E.1.4.6 Bounding $Y_6$

We first bound each term with a fixed $t\in[T]$ in $Y_6$. According to Lemma B.1, we have that with probability at least $1-\frac{\delta}{T}$,

$$\left|\left\langle\nabla\mathcal{L}(\theta_t),\frac{1}{N}V_{t-1}^{-1/2}\sum_{c\in\mathcal{C}_t}\left(R_t^\top R_t\mathrm{clip}\left(\sum_{k=1}^{K}g_{c,t,k},\tau\right)-\mathrm{clip}\left(\sum_{k=1}^{K}g_{c,t,k},\tau\right)\right)\right\rangle\right|$$

$$\leq\frac{1}{N}\sum_{c\in\mathcal{C}_t}\frac{\log^{1.5}(NTd/\delta)}{\sqrt{b}}\|\nabla\mathcal{L}(\theta_t)\|_2\left\|V_{t-1}^{-1/2}\mathrm{clip}\left(\sum_{k=1}^{K}g_{c,t,k},\tau\right)\right\|_2$$

$$\leq\frac{\log^{1.5}(NTd/\delta)G\tau_{K,G}}{\sqrt{b}\epsilon}$$

Then denote $U_t=\sum_{\tau'=0}^{t}\left\langle\nabla\mathcal{L}(\theta_{\tau'}),\frac{1}{N}V_{t-1}^{-1/2}\sum_{c\in\mathcal{C}_{\tau'}}\left(R_{\tau'}^\top R_{\tau'}\mathrm{clip}\left(\sum_{k=1}^{K}g_{c,\tau',k},\tau\right)-\mathrm{clip}\left(\sum_{k=1}^{K}g_{c,\tau',k},\tau\right)\right)\right\rangle$, we can see that $U_t$ is a martingale with respect to the sketching matrices, and from the above analysis, we have that with probability at least $1-\delta$, for all $t\in[T]$,

$$|U_t-U_{t-1}|=\left|\left\langle\nabla\mathcal{L}(\theta_t),\frac{1}{N}V_{t-1}^{-1/2}\sum_{c\in\mathcal{C}_t}\left(R_t^\top R_t\mathrm{clip}\left(\sum_{k=1}^{K}g_{c,t,k},\tau\right)-\mathrm{clip}\left(\sum_{k=1}^{K}g_{c,t,k},\tau\right)\right)\right\rangle\right|$$

$$\leq\frac{\log^{1.5}(NTd/\delta)G\tau_{K,G}}{\sqrt{b}\epsilon}$$

Then by Azuma's inequality, we have

$$\mathbb{P}\left(U_{T-1}\leq-a\right)\leq\exp\left(-\frac{a^2}{2\cdot T\cdot\left(\frac{\log^{1.5}(NTd/\delta)G\tau_{K,G}}{\sqrt{b}\epsilon}\right)^2}\right)=\exp\left(-\frac{b\epsilon^2a^2}{2T\log^2(NTd/\delta)G^2\tau_{K,G}^2}\right)$$

By selecting $a=\frac{\log^2(NTd/\delta)\sqrt{2T}G\tau}{\sqrt{b}\epsilon}$, we can get that with probability at least $1-2\delta$,

$$W_{T-1}=\sum_{t=1}^{T}\left\langle\nabla\mathcal{L}(\theta_t),\frac{1}{N}V_{t-1}^{-1/2}\sum_{c\in\mathcal{C}_t}\left(R_t^\top R_t\mathrm{clip}\left(\sum_{k=1}^{K}g_{c,t,k},\tau\right)-\mathrm{clip}\left(\sum_{k=1}^{K}g_{c,t,k},\tau\right)\right)\right\rangle$$

$$\geq-\frac{\log^2(NTd/\delta)\sqrt{2T}G\tau_{K,G}}{\sqrt{b}\epsilon}$$

(49)

Substituting 44, 45, 46, 47, 48, 49 into 43, we have that with probability at least $1-8\delta$,

$$\sum_{t=1}^{T}\frac{1}{N}\nabla\mathcal{L}(\theta_t)^\top V_{t-1}^{-1/2}\sum_{c\in\mathcal{C}_t}R_t^\top R_t\mathrm{clip}\left(\sum_{k=1}^{K}g_{c,t,k},\tau\right)$$

$$=K\sum_{t=1}^{T}\left\langle\nabla\mathcal{L}(\theta_t),V_{t-1}^{-1/2}\frac{1}{C}\sum_{c=1}^{C}\nabla\mathcal{L}_c(\theta_t)\right\rangle+K\sum_{t=1}^{T}\left\langle\nabla\mathcal{L}(\theta_t),V_{t-1}^{-1/2}\left(\frac{1}{N}\sum_{c\in\mathcal{C}_t}\nabla\mathcal{L}_c(\theta_t)-\frac{1}{C}\sum_{i=1}^{C}\nabla\mathcal{L}_c(\theta_t)\right)\right\rangle$$

$$+\sum_{t=1}^{T}\left\langle\nabla\mathcal{L}(\theta_t),\frac{1}{N}V_{t-1}^{-1/2}\sum_{c\in\mathcal{C}_t}\sum_{k=1}^{K}(\nabla\mathcal{L}_c(\theta_{c,t,k})-\nabla\mathcal{L}_c(\theta_t))\right\rangle$$

$$+\sum_{t=1}^{T}\left\langle\nabla\mathcal{L}(\theta_t),\frac{1}{N}V_{t-1}^{-1/2}\sum_{c\in\mathcal{C}_t}\sum_{k=1}^{K}(g_{c,t,k}-\nabla\mathcal{L}_c(\theta_{c,t,k}))\right\rangle$$

$$+\sum_{t=1}^{T}\left\langle \nabla\mathcal{L}(\theta_t), \frac{1}{N}V_{t-1}^{-1/2}\sum_{c\in\mathcal{C}_t}\left(\text{clip}\left(\sum_{k=1}^{K}g_{c,t,k},\tau\right)-\sum_{k=1}^{K}g_{c,t,k}\right)\right\rangle$$

$$+\sum_{t=1}^{T}\left\langle \nabla\mathcal{L}(\theta_t), \frac{1}{N}V_{t-1}^{-1/2}\sum_{c\in\mathcal{C}_t}\left(R_t^{\top}R_t\text{clip}\left(\sum_{k=1}^{K}g_{c,t,k},\tau\right)-\text{clip}\left(\sum_{k=1}^{K}g_{c,t,k},\tau\right)\right)\right\rangle$$

$$\geq\left(\eta_{\text{local}}\left(1+\frac{\log^{1.5}(NTd^2/\delta)}{\sqrt{b}}\right)\tau_{K,G}+\frac{\eta_{\text{local}}\sigma_g\log(2dT/\delta)}{\sqrt{N}}+\epsilon\right)^{-1}K\sum_{t=1}^{T}\|\nabla\mathcal{L}(\theta_t)\|_2^2-\frac{2K\sqrt{T}\log(2T/\delta)G^2}{\sqrt{N}\epsilon}$$

$$-\frac{\eta_{\text{local}}TLK^2G^2}{2\epsilon}-\frac{G\sqrt{2TK}\log(2T/\delta)\sigma_s}{\sqrt{N}\epsilon}-\max\left\{0,\frac{TG(KG-\tau)}{\epsilon}\right\}-\frac{\log^2(NTd/\delta)\sqrt{2T}G\tau_{K,G}}{\sqrt{b}\epsilon}$$

$$(50)$$

Substituting 37, 41, 42, 50 into 27, we have that with probability at least $1-17\delta$,

$$\sum_{t=1}^{T}\nabla\mathcal{L}(\theta_t)^{\top}(\gamma_{t+1}-\gamma_t)$$

$$=\sum_{t=1}^{T}\nabla\mathcal{L}(\theta_t)^{\top}\frac{\beta_1\eta_{\text{global}}}{1-\beta_1}\left(V_{t-1}^{-1/2}-V_t^{-1/2}\right)m_{t-1}$$

$$-\eta\sum_{t=1}^{T}\frac{1}{N}\nabla\mathcal{L}(\theta_t)^{\top}\left(V_t^{-1/2}-V_{t-1}^{-1/2}\right)\sum_{c\in\mathcal{C}_t}R_t^{\top}R_t\text{clip}\left(\sum_{k=1}^{K}g_{c,t,k},\tau\right)$$

$$-\eta\sum_{t=1}^{T}\frac{1}{N}\nabla\mathcal{L}(\theta_t)^{\top}V_{t-1}^{-1/2}\sum_{c\in\mathcal{C}_t}R_t^{\top}R_t\text{clip}\left(\sum_{k=1}^{K}g_{c,t,k},\tau\right)-\eta\sum_{t=1}^{T}\frac{1}{N}\nabla\mathcal{L}(\theta_t)^{\top}V_t^{-1/2}\sum_{c\in\mathcal{C}_t}R_t^{\top}\mathbf{z}_{c,t}$$

$$\leq-\left(\eta_{\text{local}}\left(1+\frac{\log^{1.5}(NTd^2/\delta)}{\sqrt{b}}\right)\tau_{K,G}+\frac{\eta_{\text{local}}\sigma_g\log(2dT/\delta)}{\sqrt{N}}+\epsilon\right)^{-1}\eta K\sum_{t=1}^{T}\|\nabla\mathcal{L}(\theta_t)\|_2^2$$

$$+\frac{2\eta K\sqrt{T}\log(2T/\delta)G^2}{\sqrt{N}\epsilon}+\frac{\eta\eta_{\text{local}}TLK^2G^2}{2\epsilon}+\frac{\eta G\sqrt{2TK}\log(2T/\delta)\sigma_s}{\sqrt{N}\epsilon}+\eta\max\left\{0,\frac{TG(KG-\tau)}{\epsilon}\right\}$$

$$+\frac{\eta\log^2(NTd^2/\delta)\sqrt{2T}G\tau_{K,G}}{\sqrt{b}\epsilon}+\frac{\eta\eta_{\text{local}}\sqrt{1-\beta_2}TG\tau_{K,G}^2}{\epsilon^2}\left(1+\frac{\log^{1.5}(NTd^2/\delta)}{\sqrt{b}}\right)^2$$

$$+\frac{\eta\eta_{\text{local}}\sqrt{1-\beta_2}TG\tau_{K,G}\sigma_g\log(2dT/\delta)}{\sqrt{N}\epsilon^2}\left(1+\frac{\log^{1.5}(NTd^2/\delta)}{\sqrt{b}}\right)$$

$$+\frac{\eta\eta_{\text{local}}\beta_1\sqrt{1-\beta_2}TG\tau_{K,G}^2}{\epsilon^2(1-\beta_1)}\left(1+\frac{\log^{1.5}(NTd^2/\delta)}{\sqrt{b}}\right)^2$$

$$+\frac{2\eta\eta_{\text{local}}\beta_1\sqrt{1-\beta_2}TG\tau_{K,G}\sigma_g\log(2dT/\delta)}{\sqrt{N}\epsilon^2(1-\beta_1)}\left(1+\frac{\log^{1.5}(CKTd/\delta)}{\sqrt{b}}\right)$$

$$+\frac{\eta\eta_{\text{local}}\beta_1\sqrt{1-\beta_2}T\sigma_g^2G\log^2(2dT/\delta)}{N\epsilon^2(1-\beta_1)}+\frac{\eta\log^2(2T/\delta)\sqrt{2T}\sigma_gG}{\sqrt{N}\epsilon}$$

$$=-\left(\eta_{\text{local}}\left(1+\frac{\log^{1.5}(NTd^2/\delta)}{\sqrt{b}}\right)\tau_{K,G}+\frac{\eta_{\text{local}}\sigma_g\log(2dT/\delta)}{\sqrt{N}}+\epsilon\right)^{-1}\eta K\sum_{t=1}^{T}\|\nabla\mathcal{L}(\theta_t)\|_2^2$$

$$+\frac{2\eta K\sqrt{T}\log(2T/\delta)G^2}{\sqrt{N}\epsilon}+\frac{\eta\eta_{\text{local}}TLK^2G^2}{2\epsilon}+\frac{\eta G\sqrt{2TK}\log(2T/\delta)\sigma_s}{\sqrt{N}\epsilon}+\eta\max\left\{0,\frac{TG(KG-\tau)}{\epsilon}\right\}$$

$$+\frac{\eta\log^2(NTd/\delta)\sqrt{2T}G\tau_{K,G}}{\sqrt{b}\epsilon}+\frac{\eta\eta_{\text{local}}\sqrt{1-\beta_2}TG\tau_{K,G}^2}{\epsilon^2(1-\beta_1)}\left(1+\frac{\log^{1.5}(NTd^2/\delta)}{\sqrt{b}}\right)^2$$

$$+\frac{\eta\eta_{\text{local}}(1+\beta_1)\sqrt{1-\beta_2}TG\tau_{K,G}\sigma_g\log(2dT/\delta)}{\sqrt{N}\epsilon^2(1-\beta_1)}\left(1+\frac{\log^{1.5}(NTd^2/\delta)}{\sqrt{b}}\right)$$

$$+\frac{\eta\eta_{\text{local}}\beta_1\sqrt{1-\beta_2}T\sigma_g^2 G\log^2(2dT/\delta)}{N\epsilon^2(1-\beta_1)}+\frac{\eta\log^2(2T/\delta)\sqrt{2T}\sigma_g G}{\sqrt{N}\epsilon}$$

$$\leq-\left(\eta_{\text{local}}\left(1+\frac{\log^{1.5}(NTd^2/\delta)}{\sqrt{b}}\right)\tau_{K,G}+\frac{\eta_{\text{local}}\sigma_g\log(2dT/\delta)}{\sqrt{N}}+\epsilon\right)^{-1}\eta K\sum_{t=1}^T\|\nabla\mathcal{L}(\theta_t)\|_2^2$$

$$+\frac{2\eta K\sqrt{T}\log(2T/\delta)G^2}{\sqrt{N}\epsilon}+\frac{\eta\eta_{\text{local}}TLK^2G^2}{2\epsilon}+\frac{\eta G\sqrt{2TK}\log(2T/\delta)\sigma_s}{\sqrt{N}\epsilon}+\eta\max\left\{0,\frac{TG(KG-\tau)}{\epsilon}\right\}$$

$$+\frac{\eta\log^2(NTd/\delta)\sqrt{2T}G\tau_{K,G}}{\sqrt{b}\epsilon}+\frac{\eta\eta_{\text{local}}(2+\beta_1)\sqrt{1-\beta_2}TG\tau_{K,G}^2}{\epsilon^2(1-\beta_1)}\left(1+\frac{\log^{1.5}(NTd^2/\delta)}{\sqrt{b}}\right)^2$$

$$+\frac{\eta\eta_{\text{local}}(1+2\beta_1)\sqrt{1-\beta_2}T\sigma_g^2 G\log^2(2dT/\delta)}{N\epsilon^2(1-\beta_1)}$$

$$+\frac{\log^2(2T/\delta)G^2}{\epsilon}+\frac{2\eta^2T\log^2(2T/\delta)\sigma_g^2}{N\epsilon}\tag{51}$$

### E.2   Bounding $T_2$

For each term in $T_2$, we have

$$(\nabla\mathcal{L}(\gamma_t)-\nabla\mathcal{L}(\theta_t))^\top(\gamma_{t+1}-\gamma_t)$$

$$=(\gamma_t-\theta_t)^\top\hat{H}'_{\mathcal{L},t}(\gamma_{t+1}-\gamma_t)$$

$$=\frac{\beta_1}{1-\beta_1}(\theta_t-\theta_{t-1})^\top\hat{H}'_{\mathcal{L}.t}\left(\frac{1}{1-\beta_1}(\theta_{t+1}-\theta_t)-\frac{\beta_1}{1-\beta_1}(\theta_t-\theta_{t-1})\right)$$

$$=\frac{\beta_1}{(1-\beta_1)^2}(\theta_t-\theta_{t-1})^\top\hat{H}'_{\mathcal{L},t}(\theta_{t+1}-\theta_t)-\frac{\beta_1^2}{(1-\beta_1)^2}(\theta_t-\theta_{t-1})^\top\hat{H}'_{\mathcal{L},t}(\theta_t-\theta_{t-1})$$

By taking summation from $0$ to $T-1$, we can get that

$$\sum_{t=1}^T(\nabla\mathcal{L}(\gamma_t)-\nabla\mathcal{L}(\theta_t))^\top(\gamma_{t+1}-\gamma_t)$$

$$=\frac{\beta_1}{(1-\beta_1)^2}\underbrace{\sum_{t=1}^T(\theta_t-\theta_{t-1})^\top\hat{H}'_{\mathcal{L},t}(\theta_{t+1}-\theta_t)}_{S_5}-\frac{\beta_1^2}{(1-\beta_1)^2}\underbrace{\sum_{t=1}^T(\theta_t-\theta_{t-1})^\top\hat{H}'_{\mathcal{L},t}(\theta_t-\theta_{t-1})}_{S_6}$$

$$\tag{52}$$

### E.2.1   Bounding $S_5$

We first bound each term in $S_5$ with a fixed $t\in[T]$.

$$(\theta_t-\theta_{t-1})^\top\hat{H}'_{\mathcal{L},t}(\theta_{t+1}-\theta_t)=\eta_{\text{global}}^2\left(V_{t-1}^{-1/2}m_{t-1}\right)^\top\hat{H}'_{\mathcal{L},t}\left(V_t^{-1/2}m_t\right)$$

$$=\eta_{\text{global}}^2\left(V_{t-1}^{-1/2}m_{t-1}\right)^\top\left(\sum_{i=1}^d\lambda_iv_iv_i^\top\right)\left(V_t^{-1/2}m_t\right)$$

$$=\eta_{\text{global}}^2\sum_{i=1}^d\lambda_i\left|\left(V_{t-1}^{-1/2}m_{t-1}\right)^\top v_i\right|\left|\left(V_t^{-1/2}m_t\right)^\top v_i\right|\tag{53}$$

For each $i\in[d]$, we have that

$$\left|\left(V_{t-1}^{-1/2}m_{t-1}\right)^\top v_i\right|=(1-\beta_1)\left|\left(\sum_{\tau=1}^{t-1}\beta_1^{t-1-\tau}V_{t-1}^{-1/2}\operatorname{desk}(\bar{\bar{\Delta}}_\tau)\right)^\top v_i\right|$$

$$\leq(1-\beta_1)\sum_{\tau'=0}^{t-1}\beta_1^{t-1-\tau'}\left|\left(V_{t-1}^{-1/2}\operatorname{desk}(\bar{\bar{\Delta}}_{\tau'})\right)^\top v_i\right|$$

$$\leq \max_{\tau' \in [t-1]} \left| \left( V_{t-1}^{-1/2} \operatorname{desk}(\bar{\tilde{\Delta}}_{\tau'}) \right)^\top v_i \right| \tag{54}$$

Similarly,

$$\left| \left( V_t^{-1/2} m_t \right)^\top v_i \right| \leq \max_{\tau' \in [t]} \left| \left( V_t^{-1/2} \operatorname{desk}(\bar{\tilde{\Delta}}_{\tau'}) \right)^\top v_i \right| \tag{55}$$

For each $\tau' \in [t-1]$,

$$\left| \left( V_{t-1}^{-1/2} \operatorname{desk}(\bar{\tilde{\Delta}}_{\tau'}) \right)^\top v_i \right|$$

$$= \left| \left\langle \operatorname{desk}(\bar{\tilde{\Delta}}_{\tau'}), V_{t-1}^{-1/2} v_i \right\rangle \right|$$

$$= \frac{1}{N} \left| \left\langle R_{\tau'}^\top \sum_{c \in \mathcal{C}_{\tau'}} \tilde{\Delta}_{c,\tau'}, V_{t-1}^{-1/2} v_i \right\rangle \right|$$

$$= \frac{1}{N} \left| \left\langle R_{\tau'}^\top \sum_{c \in \mathcal{C}_{\tau'}} \eta_{\text{local}} \left( \operatorname{sk} \left( \operatorname{clip} \left( \frac{\Delta_{c,\tau'}}{\eta_{\text{local}}}, \tau \right) \right) + \mathbf{z}_{c,\tau'} \right), V_{t-1}^{-1/2} v_i \right\rangle \right|$$

$$\leq \frac{\eta_{\text{local}}}{N} \left| \left\langle \sum_{c \in \mathcal{C}_\tau} R_{\tau'}^\top R_{\tau'} \operatorname{clip} \left( \sum_{k=1}^K g_{c,\tau',k}, \tau \right), V_{t-1}^{-1/2} v_i \right\rangle \right| + \frac{\eta_{\text{local}}}{N} \left| \left\langle \sum_{c \in \mathcal{C}_{\tau'}} R_{\tau'}^\top \mathbf{z}_{c,\tau'}, V_{t-1}^{-1/2} v_i \right\rangle \right| \tag{56}$$

For the first term, according to Lemma B.1, with probability at least $1 - \frac{\delta}{Td}$,

$$\left| \left\langle \sum_{c \in \mathcal{C}_{\tau'}} R_{\tau'}^\top R_{\tau'} \operatorname{clip} \left( \sum_{k=1}^K g_{c,\tau',k}, \tau \right), V_{t-1}^{-1/2} v_i \right\rangle \right|$$

$$\leq \left( 1 + \frac{\log^{1.5}(NTd^2/\delta)}{\sqrt{b}} \right) \left\| V_{t-1}^{-1/2} v_i \right\|_2 \sum_{c \in \mathcal{C}_{\tau'}} \left\| \operatorname{clip} \left( \sum_{k=1}^K g_{c,\tau',k}, \tau \right) \right\|$$

$$\leq \left( 1 + \frac{\log^{1.5}(NTd^2/\delta)}{\sqrt{b}} \right) \frac{N\tau_{K,G}}{\epsilon} \tag{57}$$

For the second term, $\left\langle \frac{1}{N} \sum_{c \in \mathcal{C}_{\tau'}} R_{\tau'}^\top \mathbf{z}_{c,\tau}, V_{t-1}^{-1/2} v_i \right\rangle = \left\langle R_{\tau'} V_{t-1}^{-1/2} v_i, \frac{1}{N} \sum_{c \in \mathcal{C}_{\tau'}} \mathbf{z}_{c,\tau'} \right\rangle$. Noticing that $\frac{1}{N} \sum_{c \in \mathcal{C}_{\tau'}} \mathbf{z}_{c,\tau'} \sim \mathcal{N}\left(0, \frac{\sigma_g^2}{N} \mathbb{I}\right)$, and $R_{\tau'} V_{t-1}^{-1/2} v_i$ is a $\frac{\left\| V_{t-1}^{-1/2} \right\|_2}{\sqrt{b}}$-sub-Gaussian random vector, so according to Bernstein inequality,

$$\mathbb{P}\left( \left\langle R_{\tau'} V_{t-1}^{-1/2} v_i, \frac{1}{N} \sum_{c \in \mathcal{C}_{\tau'}} \mathbf{z}_{c,\tau'} \right\rangle \geq a \right) \leq 2 \exp\left( -\min\left( \frac{a^2}{\frac{\sigma_g^2 \left\| V_{t-1}^{-1/2} \right\|_2^2}{N}}, \frac{a}{\frac{\sigma_g \left\| V_{t-1}^{-1/2} \right\|_2}{\sqrt{bN}}} \right) \right)$$

$$= 2 \exp\left( -c \min\left( \frac{Na^2}{\sigma_g^2 \left\| V_{t-1}^{-1/2} \right\|_2^2}, \frac{a\sqrt{bN}}{\sigma_g \left\| V_{t-1}^{-1/2} \right\|_2} \right) \right)$$

so taking $a = \frac{\sigma_g \left\| V_{t-1}^{-1/2} \right\|_2 \log(2T/\delta)}{\sqrt{N}}$, and noticing that $\left\| V_{t-1}^{-1/2} \right\|_2 \leq \frac{1}{\epsilon}$, we have that with probability at least $1 - \frac{\delta}{Td}$,

$$\left| \left\langle \frac{1}{N} \sum_{c \in \mathcal{C}_{\tau'}} R_{\tau'}^\top \mathbf{z}_{c,\tau'}, V_{\tau'-1}^{-1/2} v_i \right\rangle \right| \leq \frac{\sigma_g \left\| V_{t-1}^{-1/2} \right\|_2 \log(2dT/\delta)}{\sqrt{N}} \leq \frac{\sigma_g \log(2dT/\delta)}{\sqrt{N}\epsilon} \tag{58}$$

Substituting 57 and 58 into 56, we have that with probability at least $1 - \frac{2\delta}{dT}$,

$$\left| \left( V_{t-1}^{-1/2} \operatorname{desk}(\bar{\bar{\Delta}}_{\tau'}) \right)^\top v_i \right|$$

$$\leq \frac{\eta_{\text{local}}}{N} \left| \left\langle \sum_{c \in \mathcal{C}_\tau} R_{\tau'}^\top R_{\tau'} \operatorname{clip}\left( \sum_{k=1}^K g_{c,\tau',k}, \tau \right), V_{t-1}^{-1/2} v_i \right\rangle \right| + \frac{\eta_{\text{local}}}{N} \left| \left\langle \sum_{c \in \mathcal{C}_{\tau'}} R_{\tau'}^\top \mathbf{z}_{c,\tau'}, V_{t-1}^{-1/2} v_i \right\rangle \right|$$

$$\leq \frac{\eta_{\text{local}} \tau_{K,G}}{\epsilon} \left( 1 + \frac{\log^{1.5}(NTd^2/\delta)}{\sqrt{b}} \right) + \frac{\eta_{\text{local}} \sigma_g \log(2dT/\delta)}{\sqrt{N}\epsilon} \tag{59}$$

Similarly, we can get that for $\tau' \in [t]$, with probability at least $1 - \frac{2\delta}{dT}$,

$$\left| \left( V_t^{-1/2} \operatorname{desk}(\bar{\bar{\Delta}}_{\tau'}) \right)^\top v_i \right| \leq \frac{\eta_{\text{local}} \tau_{K,G}}{\epsilon} \left( 1 + \frac{\log^{1.5}(NTd^2/\delta)}{\sqrt{b}} \right) + \frac{\eta_{\text{local}} \sigma_g \log(2dT/\delta)}{\sqrt{N}\epsilon} \tag{60}$$

Substituting 59 into 54, 60 and 55, we have that with probability at least $1 - \frac{2\delta}{d}$,

$$\left| \left( V_{t-1}^{-1/2} m_{t-1} \right)^\top v_i \right| \leq \max_{\tau' \in [t-1]} \left| \left( V_{t-1}^{-1/2} \operatorname{desk}(\bar{\bar{\Delta}}_{\tau'}) \right)^\top v_i \right|$$

$$\leq \frac{\eta_{\text{local}} \tau_{K,G}}{\epsilon} \left( 1 + \frac{\log^{1.5}(NTd^2/\delta)}{\sqrt{b}} \right) + \frac{\eta_{\text{local}} \sigma_g \log(2dT/\delta)}{\sqrt{N}\epsilon} \tag{61}$$

$$\left| \left( V_t^{-1/2} m_t \right)^\top v_i \right| \leq \max_{\tau' \in [t]} \left| \left( V_t^{-1/2} \operatorname{desk}(\bar{\bar{\Delta}}_{\tau'}) \right)^\top v_i \right|$$

$$\leq \frac{\eta_{\text{local}} \tau_{K,G}}{\epsilon} \left( 1 + \frac{\log^{1.5}(NTd^2/\delta)}{\sqrt{b}} \right) + \frac{\eta_{\text{local}} \sigma_g \log(2dT/\delta)}{\sqrt{N}\epsilon} \tag{62}$$

Substituting 61 and 62 into 53, we have that with probability at least $1 - 2\delta$,

$$(\theta_t - \theta_{t-1})^\top \hat{H}'_{\mathcal{L},t} (\theta_{t+1} - \theta_t)$$

$$= \eta_{\text{global}}^2 \sum_{i=1}^d \lambda_i \left| \left( V_{t-1}^{-1/2} m_{t-1} \right)^\top v_i \right| \left| \left( V_t^{-1/2} m_t \right)^\top v_i \right|$$

$$\leq \eta_{\text{global}}^2 \sum_{i=1}^d |\lambda_i| \left( \frac{\eta_{\text{local}} \tau_{K,G}}{\epsilon} \left( 1 + \frac{\log^{1.5}(NTd^2/\delta)}{\sqrt{b}} \right) + \frac{\eta_{\text{local}} \sigma_g \log(2dT/\delta)}{\sqrt{N}\epsilon} \right)^2$$

$$\leq \eta_{\text{global}}^2 \left( 2 \left( \frac{\eta_{\text{local}} \tau_{K,G}}{\epsilon} \left( 1 + \frac{\log^{1.5}(NTd^2/\delta)}{\sqrt{b}} \right) \right)^2 + 2 \left( \frac{\eta_{\text{local}} \sigma_g \log(2dT/\delta)}{\sqrt{N}\epsilon} \right)^2 \right) \sum_{i=1}^d |\lambda_i|$$

$$= \frac{2\eta^2 \mathcal{I} L \tau_{K,G}^2}{\epsilon^2} \left( 1 + \frac{\log^{1.5}(NTd^2/\delta)}{\sqrt{b}} \right)^2 + \frac{2\eta^2 \sigma_g^2 \mathcal{I} L \log^2(2dT/\delta)}{N\epsilon^2}$$

By taking summation from 0 to $T - 1$, we have that with probability at least $1 - 2\delta$,

$$\sum_{t=1}^T (\theta_t - \theta_{t-1})^\top \hat{H}'_{\mathcal{L},t} (\theta_{t+1} - \theta_t) \leq \frac{2\eta^2 \mathcal{I} L T \tau_{K,G}^2}{\epsilon^2} \left( 1 + \frac{\log^{1.5}(NTd^2/\delta)}{\sqrt{b}} \right)^2 + \frac{2\eta^2 \sigma_g^2 \mathcal{I} L T \log^2(2dT/\delta)}{N\epsilon^2} \tag{63}$$

### E.2.2 Bounding $S_6$

We first bound each term in $S_6$ with a fixed $t \in [T]$.

$$(\theta_t - \theta_{t-1})^\top \hat{H}'_{\mathcal{L},t} (\theta_t - \theta_{t-1}) = \eta_{\text{global}}^2 \left( V_{t-1}^{-1/2} m_{t-1} \right) \hat{H}'_{\mathcal{L},t} \left( V_{t-1}^{-1/2} m_{t-1} \right)$$

$$= \eta_{\text{global}}^2 \left( V_{t-1}^{-1/2} m_{t-1} \right) \left( \sum_{i=1}^d \lambda_i v_i v_i^\top \right) \left( V_{t-1}^{-1/2} m_{t-1} \right)$$

$$= \eta_{\text{global}}^2 \sum_{i=1}^{d} \lambda_i \left| \left(V_{t-1}^{-1/2} m_{t-1}\right)^\top v_i \right|^2 \tag{64}$$

Substituting 61 into 64, we have that with probability at least $1 - 2\delta$,

$$
(\theta_t - \theta_{t-1})^\top \hat{H}'_{\mathcal{L},t} (\theta_t - \theta_{t-1}) = \eta_{\text{global}}^2 \sum_{i=1}^{d} \lambda_i \left| \left(V_{t-1}^{-1/2} m_{t-1}\right)^\top v_i \right|^2
$$

$$
\geq -\eta_{\text{global}}^2 \sum_{i=1}^{d} |\lambda_i| \left( \frac{\eta_{\text{local}} \tau_{K,G}}{\epsilon} \left(1 + \frac{\log^{1.5}(NTd^2/\delta)}{\sqrt{b}}\right) + \frac{\eta_{\text{local}} \sigma_g \log(2dT/\delta)}{\sqrt{N}\epsilon} \right)^2
$$

$$
\geq -\frac{2\eta^2 \mathcal{I} L \tau_{K,G}^2}{\epsilon^2} \left(1 + \frac{\log^{1.5}(NTd^2/\delta)}{\sqrt{b}}\right)^2 - \frac{2\eta^2 \sigma_g^2 \mathcal{I} L \log^2(2dT/\delta)}{N\epsilon^2}
$$

By taking summation from 0 to $T - 1$, we have that with probability at least $1 - 2\delta$,

$$
\sum_{t=1}^{T} (\theta_t - \theta_{t-1})^\top \hat{H}'_{\mathcal{L},t} (\theta_t - \theta_{t-1}) \geq -\frac{2\eta^2 \mathcal{I} L T \tau_{K,G}^2}{\epsilon^2} \left(1 + \frac{\log^{1.5}(NTd^2/\delta)}{\sqrt{b}}\right)^2 - \frac{2\eta^2 \sigma_g^2 \mathcal{I} L T \log^2(2dT/\delta)}{N\epsilon^2}
$$
$$\tag{65}$$

Substituting 63 and 65 into 52, we have that with probability at least $1 - 2\delta$,

$$
\sum_{t=1}^{T} (\nabla \mathcal{L}(\gamma_t) - \nabla \mathcal{L}(\theta_t))^\top (\gamma_{t+1} - \gamma_t)
$$

$$
= \frac{\beta_1}{(1-\beta_1)^2} \sum_{t=1}^{T} (\theta_t - \theta_{t-1})^\top \hat{H}'_{\mathcal{L},t} (\theta_{t+1} - \theta_t) - \frac{\beta_1^2}{(1-\beta_1)^2} \sum_{t=1}^{T} (\theta_t - \theta_{t-1})^\top \hat{H}'_{\mathcal{L},t} (\theta_t - \theta_{t-1})
$$

$$
\leq \frac{\beta_1(1+\beta_1)}{(1-\beta_1)^2} \left( \frac{2\eta^2 \mathcal{I} L T \tau_{K,G}^2}{\epsilon^2} \left(1 + \frac{\log^{1.5}(NTd^2/\delta)}{\sqrt{b}}\right)^2 + \frac{2\eta^2 \sigma_g^2 \mathcal{I} L T \log^2(2dT/\delta)}{N\epsilon^2} \right)
$$

$$
= \frac{2\eta^2 \beta_1(1+\beta_1) \mathcal{I} L T \tau_{K,G}^2}{\epsilon^2(1-\beta_1)^2} \left(1 + \frac{\log^{1.5}(NTd^2/\delta)}{\sqrt{b}}\right)^2 + \frac{2\eta^2 \beta_1(1+\beta_1) \sigma_g^2 \mathcal{I} L T \log^2(2dT/\delta)}{N\epsilon^2(1-\beta_1)^2}
$$
$$\tag{66}$$

### E.3  Bounding $T_3$

For each term in $T_3$, we have

$$
(\gamma_{t+1} - \gamma_t)^\top \hat{H}_{\mathcal{L},t} (\gamma_{t+1} - \gamma_t)
$$

$$
= \left( \frac{1}{1-\beta_1} (\theta_{t+1} - \theta_t) - \frac{\beta_1}{1-\beta_1} (\theta_t - \theta_{t-1}) \right) \hat{H}_{\mathcal{L},t} \left( \frac{1}{1-\beta_1} (\theta_{t+1} - \theta_t) - \frac{\beta_1}{1-\beta_1} (\theta_t - \theta_{t-1}) \right)
$$

$$
= \frac{1}{(1-\beta_1)^2} (\theta_{t+1} - \theta_t) \hat{H}_{\mathcal{L},t} (\theta_{t+1} - \theta_t) - \frac{2\beta_1}{(1-\beta_1)^2} (\theta_{t+1} - \theta_t) \hat{H}_{\mathcal{L},t} (\theta_t - \theta_{t-1})
$$

$$
+ \frac{\beta_1^2}{(1-\beta_1)^2} (\theta_{t+1} - \theta_t) \hat{H}_{\mathcal{L},t} (\theta_{t+1} - \theta_t)
$$

By taking summation from 0 to $T - 1$, we can get that

$$
\sum_{t=1}^{T} (\gamma_{t+1} - \gamma_t)^\top \hat{H}_{\mathcal{L},t} (\gamma_{t+1} - \gamma_t)
$$

$$
= \frac{1}{(1-\beta_1)^2} \underbrace{\sum_{t=1}^{T} (\theta_{t+1} - \theta_t) \hat{H}_{\mathcal{L},t} (\theta_{t+1} - \theta_t)}_{S_7} - \frac{2\beta_1}{(1-\beta_1)^2} \underbrace{\sum_{t=1}^{T} (\theta_{t+1} - \theta_t) \hat{H}_{\mathcal{L},t} (\theta_t - \theta_{t-1})}_{S_8}
$$

$$
+ \frac{\beta_1^2}{(1-\beta_1)^2} \underbrace{\sum_{t=1}^{T} (\theta_{t+1} - \theta_t) \hat{H}_{\mathcal{L},t} (\theta_{t+1} - \theta_t)}_{S_9} \tag{67}
$$

### E.3.1 Bounding $S_7$

We first bound each term in $S_7$ with a fixed $t \in [T]$.

$$
\begin{aligned}
(\theta_{t+1} - \theta_t) \, \hat{H}_{\mathcal{L},t} \, (\theta_{t+1} - \theta_t) &= \eta_{\text{global}}^2 \left( V_t^{-1/2} m_t \right) \hat{H}_{\mathcal{L},t} \left( V_t^{-1/2} m_t \right) \\
&= \eta_{\text{global}}^2 \left( V_t^{-1/2} m_t \right) \left( \sum_{i=1}^d \lambda_i v_i v_i^\top \right) \left( V_t^{-1/2} m_t \right) \\
&= \eta_{\text{global}}^2 \sum_{i=1}^d \lambda_i \left| \left( V_t^{-1/2} m_t \right)^\top v_i \right|^2
\end{aligned}
\tag{68}
$$

Substituting 62 into 68, we have that with probability at least $1 - 2\delta$,

$$
\begin{aligned}
&(\theta_{t+1} - \theta_t) \, \hat{H}_{\mathcal{L},t} \, (\theta_{t+1} - \theta_t) \\
=& \eta_{\text{global}}^2 \sum_{i=1}^d \lambda_i \left| \left( V_t^{-1/2} m_t \right)^\top v_i \right|^2 \\
\leq& \eta_{\text{global}}^2 \sum_{i=1}^d |\lambda_i| \left( \frac{\eta_{\text{local}} \tau_{K,G}}{\epsilon} \left( 1 + \frac{\log^{1.5}(NTd^2/\delta)}{\sqrt{b}} \right) + \frac{\eta_{\text{local}} \sigma_g \log(2dT/\delta)}{\sqrt{N}\epsilon} \right)^2 \\
=& \frac{2\eta^2 \mathcal{I} L \tau_{K,G}^2}{\epsilon^2} \left( 1 + \frac{\log^{1.5}(NTd^2/\delta)}{\sqrt{b}} \right)^2 + \frac{2\eta^2 \sigma_g^2 \mathcal{I} L \log^2(2dT/\delta)}{N\epsilon^2}
\end{aligned}
$$

By taking summation from 0 to $T - 1$, we have that with probability at least $1 - 2\delta$,

$$
\sum_{t=1}^T (\theta_{t+1} - \theta_t) \, \hat{H}_{\mathcal{L},t} \, (\theta_{t+1} - \theta_t) \leq \frac{2\eta^2 \mathcal{I} L T \tau^2}{\epsilon^2} \left( 1 + \frac{\log^{1.5}(NTd^2/\delta)}{\sqrt{b}} \right)^2 + \frac{2\eta^2 \sigma_g^2 \mathcal{I} L T \log^2(2dT/\delta)}{N\epsilon^2}
\tag{69}
$$

### E.3.2 Bounding $S_8$

We first bound each term in $S_8$ with a fixed $t \in [T]$.

$$
\begin{aligned}
(\theta_{t+1} - \theta_t) \, \hat{H}_{\mathcal{L}} \, (\theta_t - \theta_{t-1}) &= \eta_{\text{global}}^2 \left( V_t^{-1/2} m_t \right) \hat{H}_{\mathcal{L}} \left( V_{t-1}^{-1/2} m_{t-1} \right) \\
&= \eta_{\text{global}}^2 \left( V_t^{-1/2} m_t \right) \left( \sum_{i=1}^d \lambda_i v_i v_i^\top \right) \left( V_{t-1}^{-1/2} m_{t-1} \right) \\
&= \eta_{\text{global}}^2 \sum_{i=1}^d \lambda_i \left| \left( V_t^{-1/2} m_t \right)^\top v_i \right| \left| \left( V_{t-1}^{-1/2} m_{t-1} \right)^\top v_i \right|
\end{aligned}
\tag{70}
$$

Substituting 61 and 62 into 70, we have that with probability at least $1 - 2\delta$,

$$
\begin{aligned}
(\theta_{t+1} - \theta_t) \, \hat{H}_{\mathcal{L},t} \, (\theta_t - \theta_{t-1}) &= \eta_{\text{global}}^2 \sum_{i=1}^d \lambda_i \left| \left( V_t^{-1/2} m_t \right)^\top v_i \right| \left| \left( V_{t-1}^{-1/2} m_{t-1} \right)^\top v_i \right| \\
&\geq -\eta_{\text{global}}^2 \sum_{i=1}^d |\lambda_i| \left( \frac{\eta_{\text{local}} \tau_{K,G}}{\epsilon} \left( 1 + \frac{\log^{1.5}(NTd^2/\delta)}{\sqrt{b}} \right) + \frac{\eta_{\text{local}} \sigma_g \log(2dT/\delta)}{\sqrt{N}\epsilon} \right)^2 \\
&\geq -\frac{2\eta^2 \mathcal{I} L \tau_{K,G}^2}{\epsilon^2} \left( 1 + \frac{\log^{1.5}(NTd^2/\delta)}{\sqrt{b}} \right)^2 - \frac{2\eta^2 \sigma_g^2 \mathcal{I} L \log^2(2dT/\delta)}{N\epsilon^2}
\end{aligned}
$$

By taking summation from 0 to $T - 1$, we have that with probability at least $1 - 2\delta$,

$$
\sum_{t=1}^T (\theta_{t+1} - \theta_t) \, \hat{H}_{\mathcal{L},t} \, (\theta_t - \theta_{t-1}) \geq -\frac{2\eta^2 \mathcal{I} L T \tau_{K,G}^2}{\epsilon^2} \left( 1 + \frac{\log^{1.5}(NTd^2/\delta)}{\sqrt{b}} \right)^2 - \frac{2\eta^2 \sigma_g^2 \mathcal{I} L T \log^2(2dT/\delta)}{N\epsilon^2}
\tag{71}
$$

### E.3.3 Bounding $S_9$

We first bound each term in $S_9$ with a fixed $t \in [T]$.

$$
\begin{aligned}
(\theta_t - \theta_{t-1})\, \hat{H}_{\mathcal{L},t} \, (\theta_t - \theta_{t-1}) &= \eta_{\text{global}}^2 \left( V_{t-1}^{-1/2} m_{t-1} \right) \hat{H}_{\mathcal{L},t} \left( V_{t-1}^{-1/2} m_{t-1} \right) \\
&= \eta_{\text{global}}^2 \left( V_{t-1}^{-1/2} m_{t-1} \right) \left( \sum_{i=1}^d \lambda_i v_i v_i^\top \right) \left( V_{t-1}^{-1/2} m_{t-1} \right) \\
&= \eta_{\text{global}}^2 \sum_{i=1}^d \lambda_i \left| \left( V_{t-1}^{-1/2} m_{t-1} \right)^\top v_i \right|^2
\end{aligned}
\tag{72}
$$

Substituting 61 into 72, we have that with probability at least $1 - 2\delta$,

$$
\begin{aligned}
(\theta_t - \theta_{t-1})\, \hat{H}_{\mathcal{L},t} \, (\theta_{t+1} - \theta_t) &= \eta_{\text{global}}^2 \sum_{i=1}^d \lambda_i \left| \left( V_{t-1}^{-1/2} m_{t-1} \right)^\top v_i \right|^2 \\
&\leq \eta_{\text{global}}^2 \sum_{i=1}^d |\lambda_i| \left( \frac{\eta_{\text{local}} \tau_{K,G}}{\epsilon} \left( 1 + \frac{\log^{1.5}(NTd^2/\delta)}{\sqrt{b}} \right) + \frac{\eta_{\text{local}} \sigma_g \log(2dT/\delta)}{\sqrt{N}\epsilon} \right)^2 \\
&= \frac{2\eta^2 \mathcal{I} L \tau_{K,G}^2}{\epsilon^2} \left( 1 + \frac{\log^{1.5}(NTd^2/\delta)}{\sqrt{b}} \right)^2 + \frac{2\eta^2 \sigma_g^2 \mathcal{I} L \log^2(2dT/\delta)}{N\epsilon^2}
\end{aligned}
$$

By taking summation from 0 to $T-1$, we have that with probability at least $1 - 2\delta$,

$$
\sum_{t=1}^T (\theta_t - \theta_{t-1})\, \hat{H}_{\mathcal{L},t} \, (\theta_t - \theta_{t-1}) \leq \frac{2\eta^2 \mathcal{I} L T \tau_{K,G}^2}{\epsilon^2} \left( 1 + \frac{\log^{1.5}(NTd^2/\delta)}{\sqrt{b}} \right)^2 + \frac{2\eta^2 \sigma_g^2 \mathcal{I} L T \log^2(2dT/\delta)}{N\epsilon^2}
\tag{73}
$$

Substituting 69, 71, 73 into 67, we have that with probability at least $1 - 2\delta$,

$$
\begin{aligned}
&\sum_{t=1}^T (\gamma_{t+1} - \gamma_t)^\top \hat{H}_{\mathcal{L},t} \, (\gamma_{t+1} - \gamma_t) \\
={}& \frac{1}{(1-\beta_1)^2} \sum_{t=1}^T (\theta_{t+1} - \theta_t)\, \hat{H}_{\mathcal{L},t} \, (\theta_{t+1} - \theta_t) - \frac{2\beta_1}{(1-\beta_1)^2} \sum_{t=1}^T (\theta_{t+1} - \theta_t)\, \hat{H}_{\mathcal{L},t} \, (\theta_t - \theta_{t-1}) \\
&+ \frac{\beta_1^2}{(1-\beta_1)^2} \sum_{t=1}^T (\theta_{t+1} - \theta_t)\, \hat{H}_{\mathcal{L},t} \, (\theta_{t+1} - \theta_t) \\
\leq{}& \frac{(1+\beta_1)^2}{(1-\beta_1)^2} \left( \frac{2\eta^2 \mathcal{I} L T \tau_{K,G}^2}{\epsilon^2} \left( 1 + \frac{\log^{1.5}(NTd^2/\delta)}{\sqrt{b}} \right)^2 + \frac{2\eta^2 \sigma_g^2 \mathcal{I} L T \log^2(2dT/\delta)}{N\epsilon^2} \right) \\
={}& \frac{2\eta^2 (1+\beta_1)^2 \mathcal{I} L T \tau_{K,G}^2}{\epsilon^2 (1-\beta_1)^2} \left( 1 + \frac{\log^{1.5}(NTd^2/\delta)}{\sqrt{b}} \right)^2 + \frac{2\eta^2 (1+\beta_1)^2 \sigma_g^2 \mathcal{I} L T \log^2(2dT/\delta)}{N\epsilon^2 (1-\beta_1)^2}
\end{aligned}
\tag{74}
$$

Substituting 51, 66, 74 into 26, we have that with probability at least $1 - 19\delta$,

$$
\begin{aligned}
&\mathcal{L}(\gamma_T) - \mathcal{L}(\theta_0) \\
={}& \sum_{t=1}^T \nabla\mathcal{L}(\theta_t)^\top (\gamma_{t+1} - \gamma_t) + \sum_{t=1}^T (\nabla\mathcal{L}(\gamma_t) - \nabla\mathcal{L}(\theta_t))^\top (\gamma_{t+1} - \gamma_t) + \frac{1}{2} \sum_{t=1}^T (\gamma_{t+1} - \gamma_t)^\top \hat{H}_{\mathcal{L},t}(\gamma_{t+1} - \gamma_t) \\
\leq{}& -\left( \eta_{\text{local}} \left( 1 + \frac{\log^{1.5}(NTd^2/\delta)}{\sqrt{b}} \right) \tau_{K,G} + \frac{\eta_{\text{local}} \sigma_g \log(2dT/\delta)}{\sqrt{N}} + \epsilon \right)^{-1} \eta K \sum_{t=1}^T \|\nabla\mathcal{L}(\theta_t)\|_2^2 \\
&+ \frac{2\eta K \sqrt{T} \log(2T/\delta) G^2}{\sqrt{N}\epsilon} + \frac{\eta \eta_{\text{local}} T L K^2 G^2}{2\epsilon} + \frac{\eta G \sqrt{2TK} \log(2T/\delta) \sigma_s}{\sqrt{N}\epsilon} + \eta \max\left\{ 0, \frac{TG(KG - \tau)}{\epsilon} \right\}
\end{aligned}
$$

$$+ \frac{\eta \log^2(NTd/\delta)\sqrt{2T}G\tau_{K,G}}{\sqrt{b}\epsilon} + \frac{\eta\eta_{\text{local}}(2+\beta_1)\sqrt{1-\beta_2}TG\tau_{K,G}^2}{\epsilon^2(1-\beta_1)}\left(1 + \frac{\log^{1.5}(NTd^2/\delta)}{\sqrt{b}}\right)^2$$

$$+ \frac{\eta\eta_{\text{local}}(1+2\beta_1)\sqrt{1-\beta_2}T\sigma_g^2 G\log^2(2dT/\delta)}{N\epsilon^2(1-\beta_1)} + \frac{\log^2(2T/\delta)G^2}{\epsilon} + \frac{2\eta^2 T\log^2(2T/\delta)\sigma_g^2}{N\epsilon}$$

$$+ \frac{2\eta^2\beta_1(1+\beta_1)\mathcal{I}LT\tau_{K,G}^2}{\epsilon^2(1-\beta_1)^2}\left(1 + \frac{\log^{1.5}(NTd^2/\delta)}{\sqrt{b}}\right)^2 + \frac{2\eta^2\beta_1(1+\beta_1)\sigma_g^2\mathcal{I}LT\log^2(2dT/\delta)}{N\epsilon^2(1-\beta_1)^2}$$

$$+ \frac{2\eta^2(1+\beta_1)^2\mathcal{I}LT\tau_{K,G}^2}{\epsilon^2(1-\beta_1)^2}\left(1 + \frac{\log^{1.5}(NTd^2/\delta)}{\sqrt{b}}\right)^2 + \frac{2\eta^2(1+\beta_1)^2\sigma_g^2\mathcal{I}LT\log^2(2dT/\delta)}{N\epsilon^2(1-\beta_1)^2}$$

$$= -\left(\eta_{\text{local}}\left(1 + \frac{\log^{1.5}(NTd^2/\delta)}{\sqrt{b}}\right)\tau_{K,G} + \frac{\eta_{\text{local}}\sigma_g\log(2dT/\delta)}{\sqrt{N}} + \epsilon\right)^{-1}\eta K\sum_{t=1}^{T}\|\nabla\mathcal{L}(\theta_t)\|_2^2$$

$$+ \frac{2\eta K\sqrt{T}\log(2T/\delta)G^2}{\sqrt{N}\epsilon} + \frac{\eta\eta_{\text{local}}TLK^2G^2}{2\epsilon} + \frac{\eta G\sqrt{2TK}\log(2T/\delta)\sigma_s}{\sqrt{N}\epsilon} + \eta\max\left\{0, \frac{TG(KG-\tau)}{\epsilon}\right\}$$

$$+ \frac{\eta\log^2(NTd/\delta)\sqrt{2T}G\tau_{K,G}}{\sqrt{b}\epsilon} + \frac{\eta\eta_{\text{local}}(2+\beta_1)\sqrt{1-\beta_2}TG\tau_{K,G}^2}{\epsilon^2(1-\beta_1)}\left(1 + \frac{\log^{1.5}(NTd^2/\delta)}{\sqrt{b}}\right)^2$$

$$+ \frac{\eta\eta_{\text{local}}(1+2\beta_1)\sqrt{1-\beta_2}T\sigma_g^2 G\log^2(2dT/\delta)}{N\epsilon^2(1-\beta_1)} + \frac{\log^2(2T/\delta)G^2}{\epsilon} + \frac{2\eta^2 T\log^2(2T/\delta)\sigma_g^2}{N\epsilon}$$

$$+ \frac{2\eta^2(1+2\beta_1)(1+\beta_1)\mathcal{I}LT\tau_{K,G}^2}{\epsilon^2(1-\beta_1)^2}\left(1 + \frac{\log^{1.5}(NTd^2/\delta)}{\sqrt{b}}\right)^2 + \frac{2\eta^2(1+2\beta_1)(1+\beta_1)\sigma_g^2\mathcal{I}LT\log^2(2dT/\delta)}{N\epsilon^2(1-\beta_1)^2}$$

Since

$$\mathcal{L}(\theta_0) - \mathcal{L}(\gamma_T) \le \mathcal{L}(\theta_0) - \mathcal{L}^*$$

we can get that with probability at least $1 - 19\delta$,

$$\frac{1}{T}\sum_{t=1}^{T}\|\nabla\mathcal{L}(\theta_t)\|_2^2 \le \frac{\alpha_2\left(\mathcal{L}(\theta_0) - \mathcal{L}^*\right)}{\eta KT} + \frac{2\alpha_2\log(2T/\delta)G^2}{\sqrt{NT}\epsilon} + \frac{\eta_{\text{local}}\alpha_2 LKG^2}{2\epsilon} + \frac{\sqrt{2}\alpha_2 G\log(2T/\delta)\sigma_s}{\sqrt{NTK}\epsilon}$$

$$+ \max\left\{0, \frac{\alpha_2 G(KG-\tau)}{K\epsilon}\right\} + \frac{\sqrt{2}\alpha_2\log^2(NTd/\delta)G\tau_{K,G}}{\sqrt{bT}K\epsilon} + \frac{\eta_{\text{local}}\alpha_1^2\alpha_2(2+\beta_1)\sqrt{1-\beta_2}G\tau_{K,G}^2}{K\epsilon^2(1-\beta_1)}$$

$$+ \frac{\eta_{\text{local}}\alpha_2(1+2\beta_1)\sqrt{1-\beta_2}\sigma_g^2 G\log^2(2dT/\delta)}{NK\epsilon^2(1-\beta_1)} + \frac{\alpha_2\log^2(2T/\delta)G^2}{\eta TK\epsilon} + \frac{2\eta\alpha_2\log^2(2T/\delta)\sigma_g^2}{NK\epsilon}$$

$$+ \frac{2\eta\alpha_1^2\alpha_2(1+2\beta_1)(1+\beta_1)\mathcal{I}L\tau_{K,G}^2}{K\epsilon^2(1-\beta_1)^2} + \frac{2\eta\alpha_2(1+2\beta_1)(1+\beta_1)\sigma_g^2\mathcal{I}L\log^2(2dT/\delta)}{NK\epsilon^2(1-\beta_1)^2}$$

in which

$$\alpha_1 = 1 + \frac{\log^{1.5}(NTd^2/\delta)}{\sqrt{b}}, \alpha_2 = \eta_{\text{local}}\left(1 + \frac{\log^{1.5}(NTd^2/\delta)}{\sqrt{b}}\right)\tau_{K,G} + \frac{\eta_{\text{local}}\sigma_g\log(2dT/\delta)}{\sqrt{N}} + \epsilon$$

with $\tau_{K,G} = \min\{\tau, KG\}$, then we finish the proof. $\qquad\square$

# F    Additional Experimental Results

We will present more experiment results under different privacy levels. These figures further verifies our observations and conclusions in Section 4.[1]

## F.1    Vision Tasks

### F.1.1    ResNet101 on EMNIST

#### F.1.1.1 Different Privacy Level $\epsilon_p$

Training dynamics and test accuracies of Fed-SGM with ADAM, Fed-SGM with GD, DP-FedAvg and its Adam variant, and DiffSketch training ResNet101 on EMNIST with $\epsilon_p = \{2.75, 0.42, 0.18\}$ are presented in Figure 3-5. Furthermore, in Figure 6, with a fixed privacy level $\epsilon_p = 1.6$, we also show the comparison of training dynamics and test accuracies among sketching dimension $b \in \{4 \times 10^4, 4 \times 10^5, 4 \times 10^6, 4 \times 10^7\}$.

#### F.1.1.2 Ablation Studies

We conduct the following groups of ablation experiments, mirroring the setup of Figure 1 and varying only the clipping threshold $\tau$ and the number of local steps $K$.

- We compare our original setting ($\tau = 1$) with $\tau = 0.5$ and $\tau = 2$, holding all other hyperparameters constant. As shown in Table 3, in our current experimental setup, test accuracy increases as grows, indicating that a looser clipping bound allows larger gradient norms and yields better utility under our noise regime.

Table 3: Test accuracy (%) for different clipping thresholds $\tau$.

|  | GD, Sketching | GD, Non-Sketching | Adam, Sketching | Adam, Non-Sketching |
|---|---|---|---|---|
| $\tau = 0.5$ | 62.91% | 62.89% | 85.03% | 78.42% |
| $\tau = 1$ | 62.98% | 63.34% | 85.09% | 78.60% |
| $\tau = 2$ | 63.18% | 63.57% | 85.33% | 78.75% |

- We compare our original setting ($K = 18$) with $K = 9$ and $K = 36$, keeping all other hyperparameters fixed. Table 4 shows that, in our current experimental setup, increasing allows each client to perform more optimization steps, which leads to better test accuracy.

Table 4: Test accuracy (%) for different local steps $K$.

|  | GD, Sketching | GD, Non-Sketching | Adam, Sketching | Adam, Non-Sketching |
|---|---|---|---|---|
| $K = 9$ | 56.54% | 56.66% | 83.63% | 76.57% |
| $K = 18$ | 62.98% | 63.34% | 85.09% | 78.60% |
| $K = 36$ | 71.81% | 72.18% | 86.72% | 81.04% |

### F.1.2    ResNet50 on MNIST

Training dynamics and test accuracies of Fed-SGM with ADAM, Fed-SGM with GD, DP-FedAvg and its Adam variant training ResNet50 on EMNIST with $\epsilon_p = \{2.75, 0.42, 0.18\}$ are presented in Figure 7-10.

## F.2    Language Tasks

Training dynamics and test accuracies of Fed-SGM with ADAM, Fed-SGM with GD, DP-FedAvg and its Adam variant finetuning Bert on SST-2 with $\epsilon_p = \{2.45, 0.35, 0.12\}$ are presented in Figure 11-13.

---

[1]All experiments in this section and Section 4 are conducted on a computing cluster with an AMD EPYC 7713 64-core processor and an NVIDIA A100 Tensor Core GPU, and the code is provided at https://github.com/lucmonl/mlopt/tree/main.

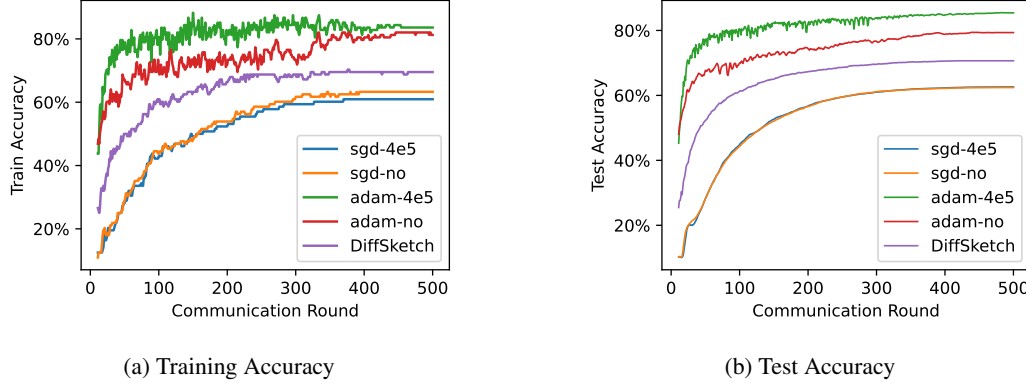

(a) Training Accuracy

(b) Test Accuracy

Figure 3: Vision Task: EMNIST, $\epsilon_p = 2.75$

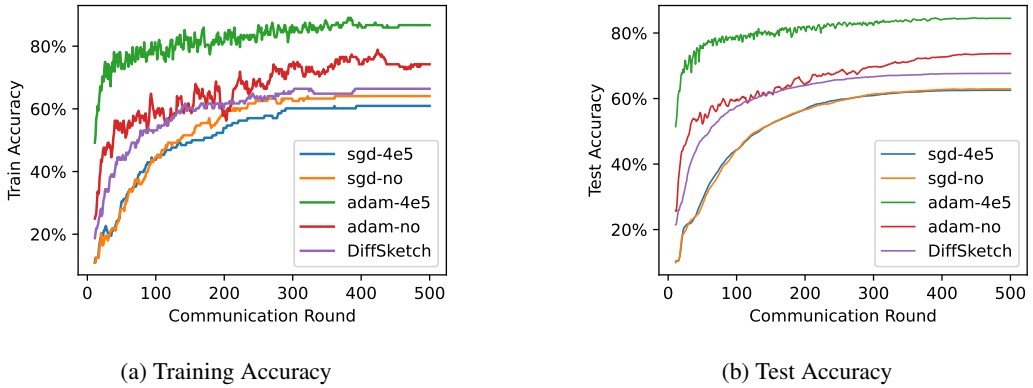

(a) Training Accuracy

(b) Test Accuracy

Figure 4: Vision Task: EMNIST, $\epsilon_p = 0.42$

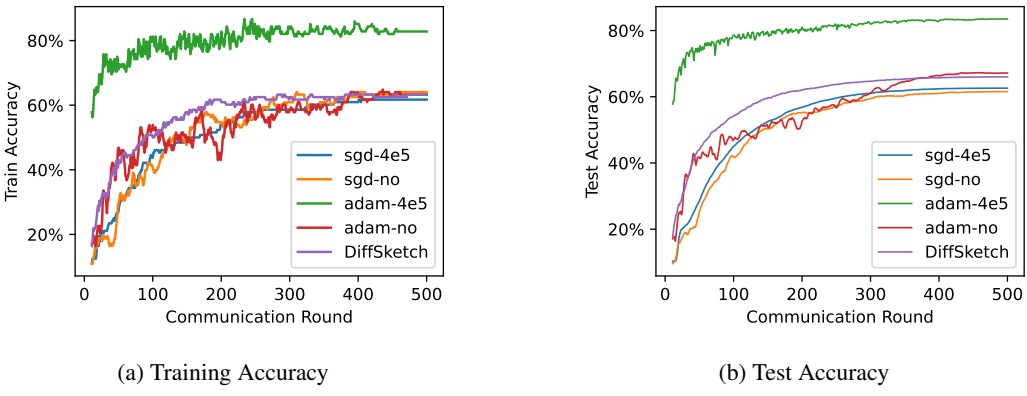

(a) Training Accuracy

(b) Test Accuracy

Figure 5: Vision Task: EMNIST, $\epsilon_p = 0.18$

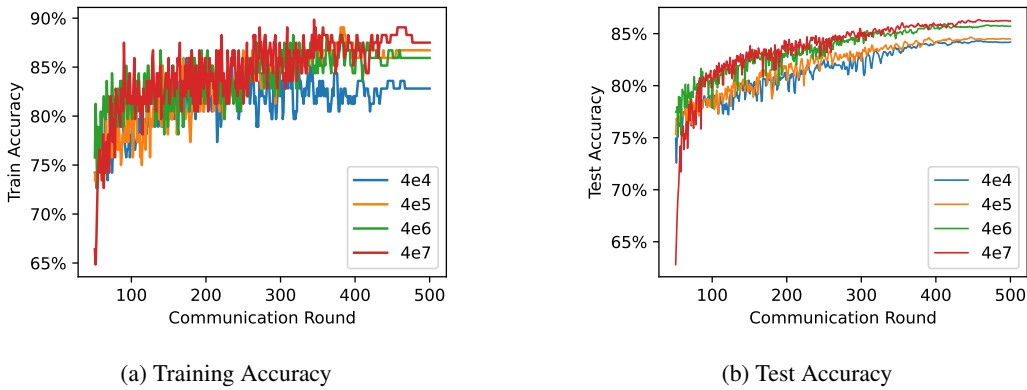

(a) Training Accuracy

(b) Test Accuracy

Figure 6: Vision task: EMNIST, $\epsilon_p = 1,60$, Comparison with Different Sketching Dimensions

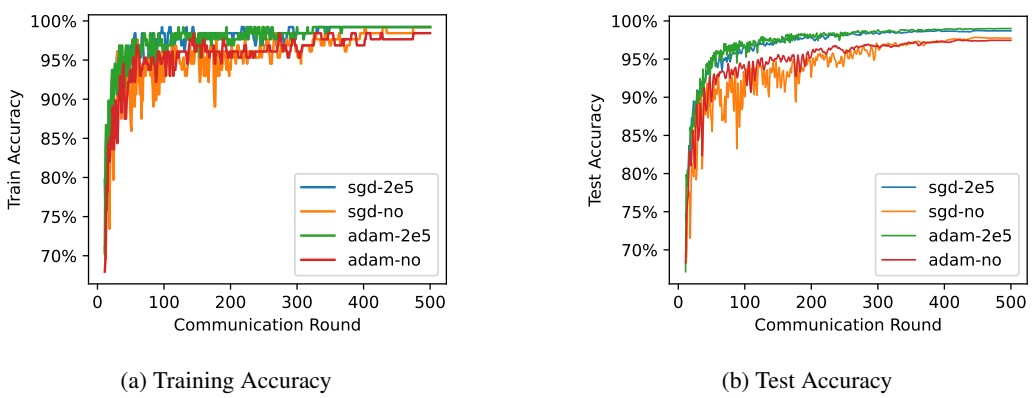

(a) Training Accuracy

(b) Test Accuracy

Figure 7: Vision task: MNIST, $\epsilon_p = 2.75$

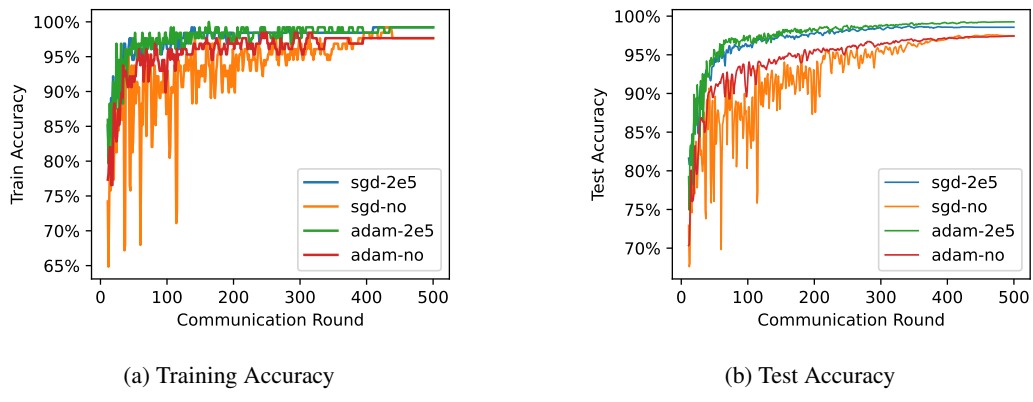

(a) Training Accuracy

(b) Test Accuracy

Figure 8: Vision task: MNIST, $\epsilon_p = 1.60$

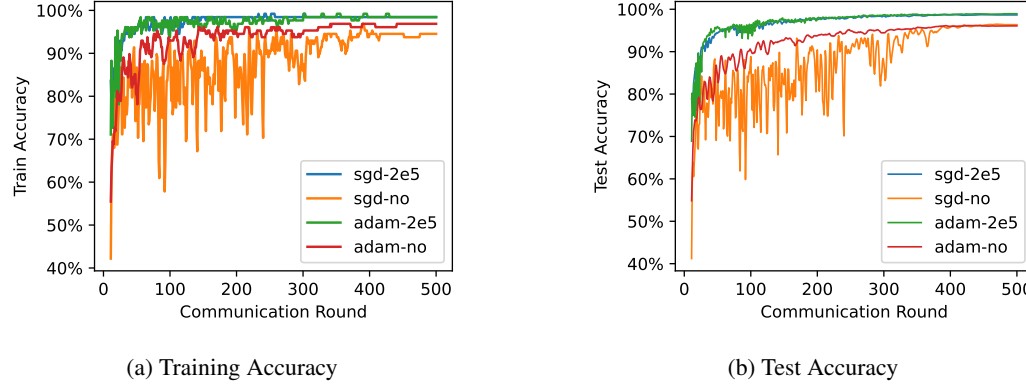

(a) Training Accuracy

(b) Test Accuracy

Figure 9: Vision task: MNIST, $\epsilon_p = 0.42$

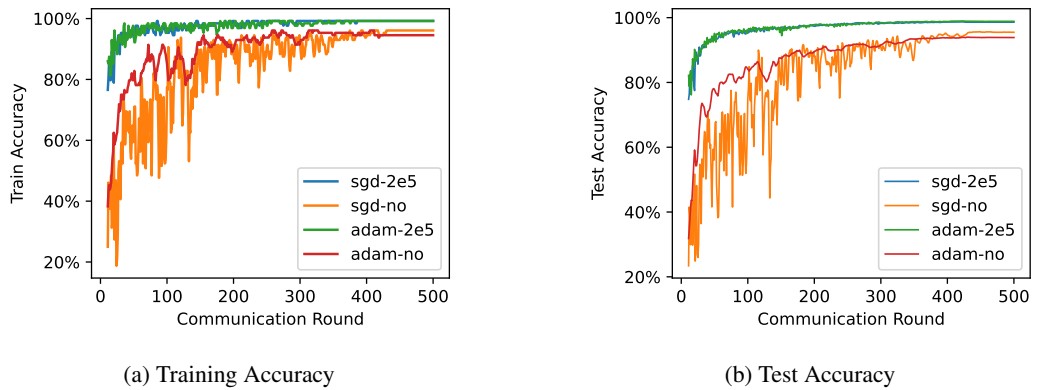

(a) Training Accuracy

(b) Test Accuracy

Figure 10: Vision task: MNIST, $\epsilon_p = 0.18$

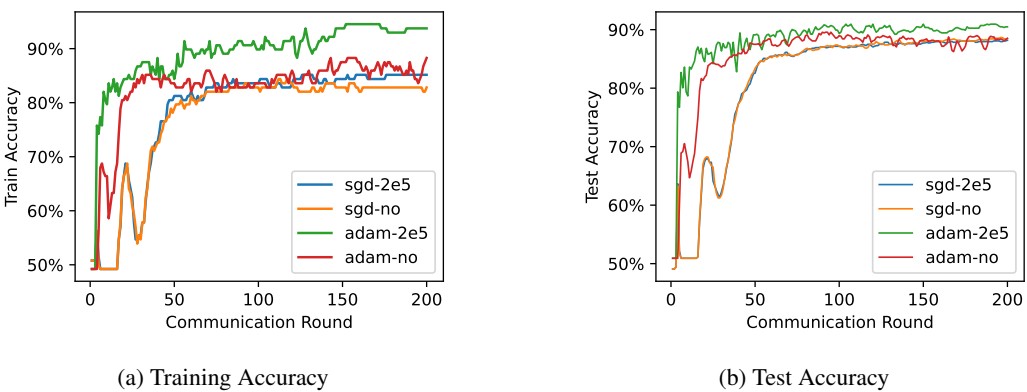

(a) Training Accuracy

(b) Test Accuracy

Figure 11: Language task, $\epsilon_p = 2.45$

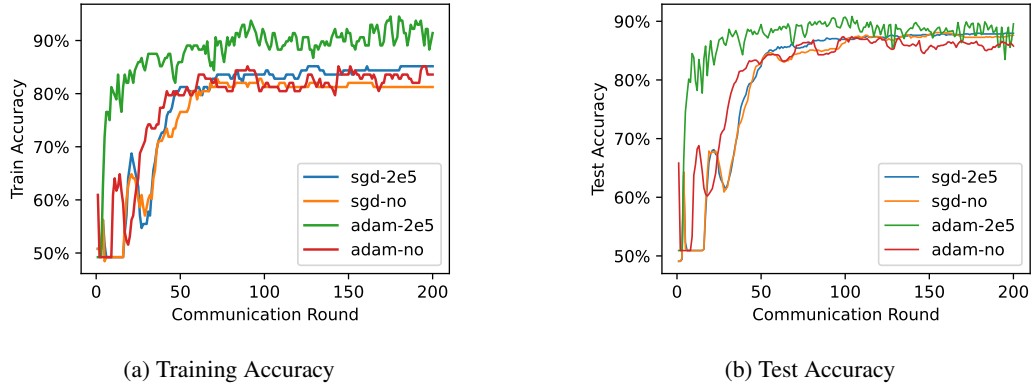

(a) Training Accuracy

(b) Test Accuracy

Figure 12: Language task, $\epsilon_p = 0.35$

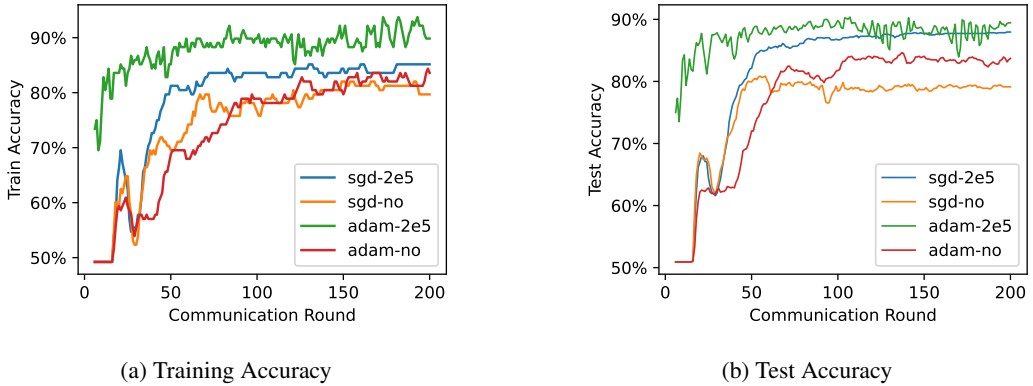

(a) Training Accuracy

(b) Test Accuracy

Figure 13: Language task, $\epsilon_p = 0.12$

