# OpenReview forum: "Sketched Gaussian Mechanism for Private Federated Learning"
_NeurIPS.cc/2025/Conference — NeurIPS 2025 spotlight_

### Official Review · Reviewer_5oY6 · 2025-07-01

**Clarity:** 2
**Significance:** 3
**Originality:** 3
**Rating:** 5
**Confidence:** 3

**Summary:**

This paper proposes the Sketched Gaussian Mechanism (SGM), an approach to jointly address communication efficiency and differential privacy (DP) in federated learning (FL). SGM combines isometric Gaussian sketching with the Gaussian mechanism for DP. The main contribution is a new privacy analysis showing that sketching itself provides DP benefits, proportional to ${b}^{-1/2}$, where $b$ is the sketching dimension. When $b$ is large enough, the established privacy bound is better than the corresponding bound with standard analysis which ignores the effect of sketching on privacy. The paper also proposes an FL algorithm incorporating SGM and provides convergence bounds including for adaptive global optimizers. It also empirically validates its effectiveness across vision and language tasks, showing competitive or better accuracy while requiring less noise.

**Questions:**

Please answer the questions in weaknesses.

**Ethical Concerns:**

["NO or VERY MINOR ethics concerns only"]

**Final Justification:**

The authors have addressed my questions and concerns decently. Overall, I think this is a solid contribution so I decided to raise my score to 5.

**Limitations:**

Limitations have been touched upon briefly in Section 5 but more discussion is needed, especially pertaining to some of the points I mentioned in weaknesses.

**Quality:**

3

**Strengths And Weaknesses:**

**Strengths:** The joint analysis of sketching and differential privacy is significant and interesting (although I have some doubts which I mention in weaknesses). Its integration into FL and derivation of a high-probability convergence bound ( especially when the global optimizer is adaptive) is good. Empirically, the benefits of the proposed scheme are decent for Adam.

**Weaknesses:**

**1.** One thing that is not clear is why should the privacy bound decrease monotonically with $b$ -- especially when the sketching matrix $R_t$ is known globally, as is the case in Algorithm 2 ($R_t$ is used in "Update parameter" step). As $b$ increases and in the case when $R_t$ is known, the amount of information released (in terms of the number of noisy projections of the vector being privatized) increases, so it is not intuitively clear to me why the privacy bounds should improve. Maybe I'm missing something here, but the authors should discuss this. In particular, when $b=d$ and suppose no extra Gaussian noise is added, one should be able to retrieve the vector being sketched precisely ($R_t$ should be invertible w.h.p.). *If this is addressed properly, I can raise my score*.

**2.** In Algorithm 2, in the "Update parameter" step, why is $R_t^\top$ being used instead of the pseudo-inverse of $R_t$? I'd expect least squares estimate to be better(?) Is it for ease of analysis or something else? The authors should discuss this.

**3.** The bound in Theorem 2.2 does not improve with $n$. Can the authors provide some discussion on the tightness of this bound?

**4.** Parsing the three terms together in Theorem 3.1 is a bit hard, especially for different values of $\tau$. I'd write the overall bound (combining the three terms) for different regimes of $\tau$.

**5.** While the authors admit that their analysis is limited to isotropic Gaussian sketching matrices, the proposed sketching method could be inefficient computationally for very large models ($d$ is very large).

**6.** The values of $b$ chosen for the experiments seem somewhat arbitrary. How should one chose $b$ in practice? I didn't see discussion on this. If I missed this, please point me to it.

**7.** The empirical benefits of the proposed scheme for SGD don't seem great. In particular, in Fig. 2, sketching seems to be doing worse.

---

> ### Author Rebuttal · Authors · 2025-07-31
>
> We thank the reviewer for recognizing the significance of our joint sketching and differential privacy analysis, its integration into federated learning with high-probability convergence bounds for adaptive optimizers, and clear empirical benefits.
>
> We will answer the questions in **Weaknesses** and **Questions**.
>
> **Q1: Why does the privacy guarantee get strictly stronger as the sketch dimension $b$ increases—even though releasing more noisy projections with a known $R_t$ should intuitively leak more information (up to exact recovery when $b=d$)?**
>
> Thank you for this important question. We address the question in two parts.
>
> * According to a direct calculation from line 151, $\mathcal{SG}(\gamma_{t}(D);R_{t},\xi_{t})\sim\mathcal{N}\left(0,\left(\frac{\\|\gamma_{t}(D)\\|\_{2}^{2}}{b}+m\sigma_{g}^{2}\right)\mathbb{I}\_{b}\right)$. As $b$ grows, the “signal” term $\frac{\\|\gamma_{t}(D)\\|\_{2}^{2}}{b}$ vanishes and the “noise” term $m\sigma_{g}^{2}$ dominates, making the output distribution increasingly independent of $D$. Consequently, for any two neighboring datasets $D$ and $D'$, the distributions $\mathcal{SG}(\gamma_{t}(D);R_{t},\xi_{t})$ and $\mathcal{SG}(\gamma_{t}(D');R_{t},\xi_{t})$ become increasingly similar, since both are essentially noise–driven. We make this rigorous in Lemma 2.2, where we show that for a fixed noise multiplier $\sigma_{g}/\tau$, the $\alpha$-Renyi divergence between these two distributions is bounded by $\frac{\alpha^{2}\tau^{4}}{(\alpha-1)b\sigma_{g}^{4}}$, so that larger $b$ strictly decreases the divergence. Therefore, increasing $b$ makes it ever harder to distinguish which dataset produced the sketch, which implies a higher privacy level.
> * Recall that for a given (gradient) vector $g$, SGM produces $\tilde{g}=Rg+\xi$, so just doing $R^\top\tilde{g}$ (or $R^\dagger\tilde{g}$) to recover $g$ will not work due to the added noise $\xi$. In fact, our privacy-amplification result hinges critically on the presence of strictly positive Gaussian noise variance $\sigma_g^2$ for $\xi$. As shown in Theorem 3.1, we need $\sigma_{g}^{2}\geq\frac{c_{4}q\tau^{2}\sqrt{T}\log(2qT/\delta_{p})}{\sqrt{b}\epsilon_{p}}$. Hence, only when $\sigma_{g}^{2}>0$ does the privacy level $\epsilon_{p}$ decrease with $b$. In degenerate case of zero added noise with $\sigma_{g}^{2}=0$, the bound in our Theorem 3.1 blows up and no privacy guarantee holds, and this includes the special case of $b=d$ with an invertible sketch. Thus, that added Gaussian noise is essential for any privacy-amplification effect.
>
> We will improve the exposition to make the point clear.
>
>
> ---
> **Q2: Why use $R_{t}^{\top}$ instead of the pseudo-inverse of $R_{t}$ in the parameter update of Algorithm 2?**
>
> Thank you for this insightful question. We chose to “desketch” with the transpose $R_{t}^{\top}$ rather than the pseudo‑inverse $R_{t}^{\dagger}$ for three closely related reasons.
>
> * Since each sketching matrix $R_t\in\mathbb{R}^{b\times d}$ is drawn i.i.d. Gaussian with variance $\frac{1}{\sqrt{b}}$ so that $\mathbb{E}[R_t^\top R_t]=\mathbb{I}\_d$, then for any vector $g$, $\mathbb{E}[R_t^\top R_tg]=g$, i.e., $R_t^\top$ can recover $g$ in expectation. Furthermore, standard concentration results imply $R_t^\top R_t\approx\mathbb{I}\_{d}$ when $b$ is in the usual Johnson–Lindenstrauss regime. Then, $(R_{t}^{\top}R_{t})^{-1}\approx\mathbb{I}\_{d}$, so $R_{t}^{\dagger}=(R_{t}^{\top}R_{t})^{-1}R_{t}^{\top}\approx R_{t}^{\top}$, which shows that $R_t^\top$ suffices.
> * Inverting the $d\times d$ matrix $R_{t}^{\top}R_{t}$ at every round would introduce significant extra computation cost.
> * In the common case where $b\ll d$ (e.g., $b\approx0.01d$ in our experiments), the system $R_tg = y$ is underdetermined. Even the pseudo-inverse $R_{t}^{\dagger}$ recovers only the projection of $g$ onto the row space of $R_t$, losing the complementary subspace. Hence, any desketching operator—whether transpose or pseudo-inverse—cannot recover the unencoded information.
>
> This was an excellent question----we will add a discussion to make the point clear.
>
> ---
> **Q3: The bound in Theorem 2.2 does not improve with $n$.**
>
> The bound in Theorem 2.2 does, in fact, exhibit improvement with respect to the number of clients $n$, though this dependency is implicit. Specifically, the bound is given by $\sigma_{g}^{2}\geq\frac{c_{4}q\tau^{2}\sqrt{T}\log(2qT/\delta_{p})}{\sqrt{b}\epsilon_{p}}$, where $q=\frac{m}{n}$ represents the per‑round sampling ratio, with $m$ being batch size and $n$ being total sample size. It is evident from this formulation that, when all other parameters are held fixed, the bound decreases monotonically as $n$ increases. In this sense, the bound improves with a larger sample population $n$.
>
> ---
> **Q4: Bound in Theorem 3.1 separately for each regime of τ to make it easier to parse.**
>
> Thank you for the suggestion on Theorem 3.1. We agree that clarifying how three components interact—especially with respect to clipping threshold $\tau$—will improve exposition.
>
> We present the bound in the two regimes of $\tau$:
>
> 1. When $\tau\leq KG$: In this regime, clipping may be active, and $\tau$ acts as an upper bound on the norm of each clipped update $\Delta_{c,t}$ according to our algorithm. The bound in this case is $\mathcal{O}\left(\frac{\mathcal{L}(\theta_0)-\mathcal{L}^*}{\eta TK} \right)+\tilde{\mathcal{O}}\left(\frac{1}{\sqrt{NT}}\right)+\tilde{\mathcal{O}}(\eta_{\text{local}}K)+\tilde{\mathcal{O}}\left(\frac{\tau}{\sqrt{bT}K}\right)+\tilde{\mathcal{O}}\left(\frac{(\eta_{\text{local}}+\eta\mathcal{I})\tau^2}{K}\right)+\tilde{\mathcal{O}}\left(\frac{1}{\eta TK}\right)+\max\left\\{0,\frac{\left(\epsilon+\tilde{\mathcal{O}}(\eta_{\text{local}})\right)G(KG-\tau)}{K\epsilon}\right\\}+\tilde{\mathcal{O}}\left(\frac{(\eta_{\text{local}}+\eta\mathcal{I})\sigma_g^2}{NK}\right).$ We observe that $\tilde{\mathcal{O}}\left(\frac{\tau}{\sqrt{bT}K}\right)+\tilde{\mathcal{O}}\left(\frac{(\eta_{\text{local}}+\eta\mathcal{I})\tau^2}{K}\right)$ in the optimization bound of sketching algorithm is monotonically increasing in $\tau$, while $\max\left\\{0,\frac{\left(\epsilon+\widetilde{\mathcal{O}}(\eta_{\text{local}})\right)G(K G-\tau)}{K\epsilon}\right\\}$ caused by clipping is monotonically decreasing in $\tau$. This trade-off highlights the need for a careful choice of $\tau$, as overly aggressive clipping threshold introduces clipping bias, while too loose a threshold amplifies optimization error of sketching algorithm.
>
> 2. When $\tau\geq KG$: In this regime, the clipping operation becomes inactive, as all updates fall within clipping threshold $\tau$. $KG$ effectively bounds norms of each update $\Delta_{c,t}$, and clipping no longer influences computation. Therefore, the total bound becomes $\mathcal{O}\left(\frac{\mathcal{L}(\theta_0)-\mathcal{L}^*}{\eta TK}\right)+\tilde{\mathcal{O}}\left(\frac{1}{\sqrt{NT}}\right)+\tilde{\mathcal{O}}(\eta_{\text{local}}K)+\tilde{\mathcal{O}}\left(\frac{1}{\sqrt{bT}} \right)+\tilde{\mathcal{O}}\left((\eta_{\text{local}}+\eta\mathcal{I})K\right)+\tilde{\mathcal{O}}\left(\frac{1}{\eta TK}\right)+\tilde{\mathcal{O}}\left(\frac{(\eta_{\text{local}}+\eta\mathcal{I})\sigma_g^2}{NK}\right)$. Consequently, the bound becomes independent of $\tau$ and remains constant as $\tau$ increases further.
>
> We believe this two-regime characterization offers a clearer view of how $\tau$ affects the bound and further confirms the intuition behind its tuning.
>
> ---
> **Q5: Sketching with Gaussian matrices could be inefficient computationally for large models.**
>
> We acknowledge that isotropic Gaussian sketching adds computational cost for large $d$, due to matrix–vector operations during sketching and de-sketching at each client. Our key contribution is showing that sketching, beyond dimension reduction, also reduces the noise needed for differential privacy. Future work will extend this to more general, efficient sketching matrices like CountSketch and SRHT.
>
> ---
> **Q6: Choice of $b$**
>
> As stated in Parameter Setting in Section 4, we chose $b$ as a very small fraction of the original parameter dimension $d$ to demonstrate the practicality of sketching. For image classification experiments in Figure 1 (ResNet‑101 on EMNIST), we used a $1\\%$ compression rate, i.e., $b = 0.01d$, reducing the dimension from 42 million down to $4\times10^5$. For the language task in Figure 2(BERT on SST‑2), we applied a $0.2\\%$ compression rate, i.e., $b=0.002d$, shrinking 100 million dimensions to $2\times10^5$.
>
> Remarkably, despite this aggressive reduction in dimensions, the sketching‑based method remains competitive with the non‑sketching baseline and even outperforms it under certain situations.
>
> ---
> **Q7: Empirical benefits of the proposed scheme for SGD**
>
> First, projecting each client’s gradient into a lower-dimensional subspace alters the update direction, introducing some loss relative to the original unsketched update, regardless of using vanilla SGD or adaptive optimizers. This explains why sketching-based curves in Figure 2 may lie slightly below the non-sketching baseline.
>
> Still, the sketching algorithms closely match—and sometimes exceed—the non-sketching baselines in accuracy. This stems from sketching’s dual role in reducing Gaussian noise for privacy: (1) shrinking noise dimensionality from $d$ to $b\ll d$—with $b$ set to $0.2\\%$ of $d$ in Figure 2 (a 500-fold reduction); and (2) requiring significantly lower noise variance per coordinate. Theorem 3.1 shows that, for the same privacy guarantee, sketching needs smaller variance per coordinate compared to non-sketching methods (nearly one-tenth in Figure 2). These two effects drastically lower total noise, offsetting the projection-induced error.
>
> Finally, this holds under extreme sketching compression rates ($0.2\\%$), drastically reducing per-round communication. Achieving strong accuracy with such low communication highlights the scheme’s practical utility in large-scale federated learning.

---

> > ### Comment · Reviewer_5oY6 · 2025-08-05
> > **Response to Authors**
> >
> > I thank the authors for the detailed rebuttal. Regarding Q1, I think I sort of understand now; I didn't consider the effect of noise scaling earlier. I request the authors to include the discussion on this in the next version. Regarding Q3, thanks for the clarification but it seems that Theorem 2.1 is better than Theorem 2.2 by a factor of $1/n$.
> >
> > Overall, I think this is a good contribution so I'll raise my score to 5.

---

> > > ### Author Response · Authors · 2025-08-05
> > >
> > > We are grateful to the reviewer for these positive remarks and for increasing your score to 5.
> > >
> > > Regarding Q1:
> > >
> > > We appreciate your understanding on the role of noise‐scaling, and we will include a comprehensive discussion and explanation on this perspective in the next revision.
> > >
> > > Regarding Q3:
> > >
> > > We agree that, in terms of $n$, Theorem 2.1 establishes $\sigma_{g}^{2}\geq O(n^{-2})$ while Theorem 2.2 yields $\sigma_{g}^{2}\geq O(n^{-1})$. The principal advantage of Theorem 2.2, however, is the explicit incorporation of the sketching dimension $b$ into the noise bound. We will investigate whether the dependence on $n$ can be further tightened within the analysis of Theorem 2.2.
> > >
> > > Once again, thank you for your careful review and constructive suggestions.

---

### Official Review · Reviewer_VjkF · 2025-07-01

**Clarity:** 2
**Significance:** 2
**Originality:** 2
**Rating:** 4
**Confidence:** 3

**Summary:**

The paper proposes Sketched Gaussian Mechanism to provide a joint privacy analysis for both sketching and Gaussian mechanism, which results in a tighter bound while providing detailed theoretical proofs and certain experimental evaluation.

**Questions:**

1. Why do we only look at Renyi mechanism? What not consider zCDP?
2. What are the justifications of fixing the experimental setups to the following: "sampling N = 4 clients uniformly at289
random in each communication round. Each selected client executes K = 18 local SGD updates290
on mini-batches of size 64, with gradient clipping threshold τ = 1." For the privacy part, why is the ${\delta}_p$ fixed?

**Ethical Concerns:**

["NO or VERY MINOR ethics concerns only"]

**Final Justification:**

The usage of Sketched Gaussian Mechanism in this work improves the privacy bound in federate learning by a promising margin.

**Limitations:**

Yes

**Quality:**

3

**Strengths And Weaknesses:**

## Strengths

1. The paper proposes Sketched Gaussian Mechanism to provide a joint privacy analysis for both sketching and Gaussian mechanism, which results in a tighter bound by $\frac{1}{\sqrt{d}}$.

2. The paper includes detailed theoretical proofs for both privacy guarantee and convergence analysis.

## Weaknesses

1. The evaluation is limited to some datasets and specific models. Additionally, the experimental setups are fixed to one dimension. The expandability of the work is unclear. It seems like the experiment was run once, there is no indicator of statistical significance especially considering the random sampling from the Gaussian mechanism.

2. The paper can benefit from a more clarified problem statement with proper background knowledge. Specifically, there is no clear introduction on Renyi DP and sketching but the paper jumped straight to the definition of SGM.

3. The paper is based on the assumption that DP-based solution can solve the privacy challenge for federated learning, but in reality, DP-based solutions fall short when handling a distributed training system with small client sizes. There is no discussion on other privacy-preserving solutions including MPC, HE and hybrid of DP + cryptographic primitives.

4. Minor:
(1) typo in Algorithm 2: "Computer update" -> "Compute update"
(2) Equation at Line 252 is out of bound.

---

> ### Author Rebuttal · Authors · 2025-07-31
>
> We thank the reviewer for noting our Sketched Gaussian Mechanism and our rigorous proofs of both privacy guarantees and convergence.
>
> We will respond to the problems outlined in the **Weaknesses** and **Questions** sections.
>
> **Q1: The evaluation is limited to some datasets and specific models.**
>
> We thank the reviewer for raising concerns about breadth of our evaluation. In the paper, we report results on two different tasks：ResNet-101 on EMNIST for image classification and BERT on SST-2 for sentiment analysis。 In both cases, Adam as the global optimizer outperforms GD, while our sketching variants remain competitive with non-sketching counterparts. We believe this consistent behavior across vision and language benchmarks speaks to the generality of our approach.
>
> To further demonstrate expandability, we add a new experiment using ResNet-50 on MNIST with privacy level $\epsilon_{p}=1.60$. We observe the same qualitative trends: Adam yields higher accuracy than GD, and sketching incurs no significant loss relative to non-sketched baseline. These additional results reinforce that our method extend well beyond the initially reported settings.
>
> |GD, Sketching|GD, Non-Sketching|Adam, Sketching|Adam, Non-Sketching
> |:-:|:-:|:-:|:-:|
> |$82.56\\%$|$83.44\\%$|$90.26\\%$|$89.42\\%$|
>
> ---
> **Q2: Experimental setups are fixed to one dimension.**
>
> We fixed the sampling rate $N=4$, the number of local updates $K=18$, mini‑batch size 64, and clipping threshold $\tau=1$ in order to compare the four algorithmic variants under identical system conditions, so that any observed performance gap then reflects only the choice of global optimizer (SGD vs. AMSGrad) and the presence or absence of sketching.
>
> To verify that these relative comparisons hold more broadly, we ran additional experiments under the setting of Figure 1 (ResNet-101 on EMNIST), varying the clipping threshold $\tau$ and the number of local steps $K$, while keeping all other settings the same. Across different values of $\tau$ and $K$, the four variants exhibit the same pattern: AMSGrad consistently outperforms SGD as the global optimizer, and sketching‑enabled methods remain competitive with or superior to their non‑sketching counterparts.
>
> |          | GD, Sketching |   GD, Non-Sketching    |Adam, Sketching|Adam, Non-Sketching|
> |:-----------:|:---:|:----------:|:---:|:---:|
> | $\tau=0.5$  |  $62.91\\%$ | $62.89\\%$ | $85.03\\%$ | $78.42\\%$ |
> | $\tau=1$      |  $62.98\\%$ | $63.34\\%$  |$85.09\\%$ | $78.60\\%$ |
> | $\tau=2$  |  $63.18\\%$ | $63.57\\%$ | $85.33\\%$ | $78.75\\%$|
>
> |          | GD, Sketching |   GD, Non-Sketching    |Adam, Sketching|Adam, Non-Sketching|
> |:-----------:|:---:|:----------:|:---:|:---:|
> | $K=9$  |  $56.54\\%$ | $56.66\\%$ | $83.63\\%$ | $76.57\\%$ |
> | $K=18$      |  $62.98\\%$ | $63.34\\%$  |$85.09\\%$ | $78.60\\%$ |
> | $K=36$  |  $71.81\\%$ | $72.18\\%$ | $86.72\\%$ | $81.04\\%$|
>
> ---
> **Q3: For the privacy part, why is the $\delta_{p}$ fixed?"**
> We fix the privacy failure probability $\delta_{p}=10^{-5}$ as a constant in all of our experiments, which is standard practice in the differential‑privacy literature。 For example， both Section 4 of [1] and Section 7 of [2] likewise set $\delta_{p}=10^{-5}$. By holding $\delta_{p}$ at this sufficiently small value, it allows us to isolate the impact of $\epsilon$ and our algorithmic choices without conflating changes in $\delta_{p}$.
>
> [1] Zhang, Xinwei, et al. "Understanding clipping for federated learning: Convergence and client-level differential privacy." International Conference on Machine Learning, ICML 2022.
>
> [2] Wang, Lun, Ruoxi Jia, and Dawn Song. "D2P-Fed: Differentially private federated learning with efficient communication." arXiv preprint arXiv:2006.13039 (2020).
>
> ---
> **Q4:** "It seems like the experiment was run once, there is no indicator of statistical significance especially considering the random sampling from the Gaussian mechanism."
>
> We thank the reviewer for pointing out the need for statistical significance. As noted in our NeurIPS Paper Checklist (line 1193), each of our core experiments was in fact repeated 5 times with independent random seeds for the Gaussian noises and sketching matrices, and the curves in the paper represent the mean performance across those 5 runs. To make this explicit, we present the following table with error bars showing the standard deviation over the five trials. These error bars demonstrate that the performance trends we report hold consistently across repetitions, thereby addressing concerns about variability due to random sampling.
>
>  | GD, Sketching |   GD, Non-Sketching    | Adam, Sketching| Adam, Non-Sketching
> |:---:|:----------:|:---:|:---:|
> |$62.98\\%\pm0.76\\%$|$63.34\\%\pm0.37\\%$|  $85.09\\%\pm0.28\\%$ | $78.60\\%\pm0.19\\%$ |
>
> ---
> **Q5: Clarified problem statement with proper background knowledge on Renyi DP and sketching**
>
> Thank you for the suggestion. We agree that a more thorough introduction to both Renyi Differential Privacy (RDP) and sketching would improve clarity and accessibility for a broader audience. We will revise the manuscript to incorporate these foundational elements more explicitly in the main text.
> * **Renyi DP:** Renyi DP is a widely used analytical framework for differential privacy. A randomized mechanism $M$ is said to satisfy $(\alpha,\epsilon(\alpha))$-RDP of order $\alpha$ if $D_{\alpha}(M(D)\|M(D'))\leq\epsilon(\alpha)$, where $D_{\alpha}$ denotes the Renyi divergence of order $\alpha$ (see Definition 2.3). RDP is particularly suitable in our setting for two main reasons. First, it has a well-established connection to $(\epsilon,\delta)$-DP via Lemma 2.3, allowing us to translate RDP bounds into standard privacy guarantees. Second, because both the sketching transformation and the added noise in SGM are Gaussian, the resulting mechanism remains Gaussian, enabling a closed-form and tractable analysis of the Rényi divergence. These properties make RDP a natural choice for our privacy analysis.
> * **Sketching:** Sketching is a classical technique for dimensionality reduction. Given a high-dimensional vector or matrix $X$, sketching applies a projection via a sketching matrix $S$ to produce $SX$, which resides in a lower-dimensional subspace. This reduces communication and computation while approximately preserving key geometric properties of the original data. We briefly introduce sketching in the Introduction section and provide additional applications of sketching in Related Work in appendix.
>
> ---
> **Q6: Why do the authors focus exclusively on DP for federated learning without considering other privacy-preserving approaches like MPC, HE, or DP–cryptographic hybrids?**
>
> We appreciate the reviewer’s suggestion to consider cryptographic approaches such as secure multi-party computation (MPC), homomorphic encryption (HE), and hybrid schemes that combine differential privacy (DP) with cryptographic primitives. We are aware of active line of work in this area and agree that these methods provide strong privacy guarantees at protocol level.
>
> That said, our work is grounded in the well-established line of research that uses differential privacy (DP) as central mechanism for privacy protection in federated learning. DP-based solutions to privacy of federated learning have been extensively studied, particularly for their composability, formal guarantees, and scalability to large client populations. Notable examples include [1, 2, 3], among many others.
>
> Our focus is on improving the utility and communication efficiency of DP-based federated learning while maintaining formal privacy guarantees. If opportunity allows, we are interested in extending our techniques and analyses to hybrid approaches that incorporate cryptographic protections, and we view this as a promising direction for future work.
>
> [1] Geyer, Robin C., Tassilo Klein, and Moin Nabi. "Differentially private federated learning: A client level perspective." arXiv preprint arXiv:1712.07557 (2017).
>
> [2] Zhang, Xinwei, et al. "Understanding clipping for federated learning: Convergence and client-level differential privacy." International Conference on Machine Learning, ICML 2022.
>
> [3] Wang, Lun, Ruoxi Jia, and Dawn Song. "D2P-Fed: Differentially private federated learning with efficient communication." arXiv preprint arXiv:2006.13039 (2020).
>
> ---
> **Q7:** "Why do we only look at Rnyi mechanism? What not consider zCDP?"
>
> We first recall the two definitions. A mechanism $\mathcal{M}$ satisfies $(\alpha,\epsilon(\alpha))$-RDP if $D_{\alpha}\left(\mathcal{M}(D)\|\mathcal{M}(D')\right)\leq\epsilon(\alpha)$, where $D_{\alpha}$ is the Renyi divergence of order $\alpha$. By contrast, $(\xi,\rho)$-zCDP requires that for every $\alpha>1$, $D_{\alpha}\left(\mathcal{M}(D)\|\mathcal{M}(D')\right)\leq\xi+\rho\alpha$. In other words, zCDP can be viewed as a special case of RDP in which the mechanism satisfies $(\alpha,\epsilon(\alpha))$-RDP for all $\alpha>1$, and the corresponding $\epsilon(\alpha)$ grows linearly with the order $\alpha$. In our Lemma 2.4, however, we demonstrates that SGM satisfies $(\alpha,\epsilon(\alpha))$-RDP with $\epsilon(\alpha)=\frac{\alpha^{2}\tau^{4}}{(\alpha-1)b\sigma_{g}^{4}}$ which fails the linear growth condition of $\epsilon(\alpha)$ required by zCDP. Therefore, we cannot claim a zCDP guarantee for any fixed $(\xi,\rho)$. Instead, we work in the full RDP framework, which precisely captures the true $\alpha$-dependence of our privacy bound.
>
> ---
> **Q8:** "Minor: typo in Algorithm 2: "Computer update" $\to$ "Compute update"."
>
> Thank you for pointing out the typo in Algorithm 2. We will thoroughly proofread the entire manuscript and correct all typographical errors in the upcoming revision.
>
> ---
> **Q8:** "Minor: Equation at Line 252 is out of bound."
>
> we will ensure that any equations exceeding the column width are reformatted either by using a smaller font size or by breaking them across multiple lines in the next revision.

---

> > ### Comment · Reviewer_VjkF · 2025-08-05
> >
> > Thanks for the clarification. I will keep my positive rating.

---

> ### Author Response · Authors · 2025-08-05
>
> We are grateful for your positive review. Thank you for your continued support of our work.

---

### Official Review · Reviewer_q7wo · 2025-07-02

**Clarity:** 3
**Significance:** 3
**Originality:** 3
**Rating:** 5
**Confidence:** 4

**Summary:**

While existing works primarily analyze the privacy guarantees of sketching and Gaussian mechanisms in isolation, this paper considers the inherent privacy benefits of the sketching operations and derives tighter privacy bounds for the combined sketched Gaussian mechanism by leveraging Rényi differential privacy. The proposed framework is applied to federated learning with both gradient descent and adaptive optimizers, demonstrating superior empirical performance.

**Questions:**

1. Is it possible to add comparisons with more recent works on privacy-communication-accuracy tradeoff? Comparisons from both theoretical and experimental perspectives will benefit the paper.
2. I do not get the first equality in line 646 of Lemma C.2. Could you please clarify?

**Ethical Concerns:**

["NO or VERY MINOR ethics concerns only"]

**Final Justification:**

After the rebuttal, I remain positive about this paper.

**Limitations:**

The authors have mentioned that the analyses are limited to isotropic Gaussian sketching matrices.

**Paper Formatting Concerns:**

Some equations are not appropriately numbered.

**Quality:**

3

**Strengths And Weaknesses:**

Strengths:
1. The theoretical derivation is comprehensive and rigorously demonstrates a tighter bound on the privacy level of SGM.
2. Beyond the theoretical analysis, the authors apply SGM to Federated Learning (FL) and conduct a thorough convergence analysis, demonstrating its performance in terms of convergence bounds.
3. Both NLP and image classification tasks are considered in the experiments.

Weakness:
1. The derivation is quite complex, and some high-level explanations would help readers better understand and appreciate the analyses.
2. In the experiments, the proposed method is only compared with DiffSketch (which was published in 2019) from the communication overhead perspective. Considering that there are many existing works that consider the tradeoff between privacy, communication overhead, and utility, comparisons with more recent works are expected.
3. Some equations are not appropriately numbered.

---

> ### Author Rebuttal · Authors · 2025-07-31
>
> We’re grateful for the reviewer’s acknowledgement of our novel tighter privacy derivation for SGM, our thorough convergence guarantees in the federated learning setting, and broad empirical validations across NLP and image classification.
>
> We’ll address the points listed under **Weaknesses** and **Questions**.
>
> **Q1: High-level explanations to clarify the complex derivation**
>
> Thank you for the suggestion. We agree that high-level explanations can help improve accessibility of the analysis. Below is a brief overview of key ideas behind our derivations.
>
> **Privacy analysis:** To clarify privacy guarantees of SGM, we include proof sketches in Section 2.2. Our goal is to analyze $(\epsilon,\delta)$-differential privacy (DP) of SGM using Renyi Differential Privacy (RDP) framework. According to the definition, a mechanism $\mathcal{M}$ satisfies $(\alpha,\epsilon(\alpha))$-RDP if $D_{\alpha}(\mathcal{M}(D)\|\mathcal{M}(D'))\leq\epsilon(\alpha)$, where $D_{\alpha}$ is Renyi divergence of order $\alpha$. Leveraging the standard conversion between RDP and $(\epsilon,\delta)$-DP (Lemma 2.3), we focus on bounding RDP of SGM. This requires bounding Renyi divergence between output distributions under neighboring datasets $D$ and $D'$.
>
> Using Lemma 2.1, we reduce the problem to analyzing the ratio sensitivity (Definition 2.4) and $f_{\alpha}$ (Lemma 2.1). By bounding the ratio sensitivity and obtaining properties of $f_{\alpha}$, we derive Lemma 2.2, which shows that the divergence is bounded by $O(1/b)$. This leads to an RDP guarantee and, via conversion, to a $(\epsilon,\delta)$-DP result (Lemma 2.4). Finally, applying standard tools in DP analysis, including post-processing, composition, and sub-sampling, we derive privacy guarantee for Algorithm 1 in Theorem 3.1.
>
> **Optimization guarantee:** We first analyze convergence of Fed-SGM under global GD optimizer. The second-order Taylor expansion (equation (7), line 707) splits the loss difference to two parts: a first-order term $T_{1}$ and a second-order term $T_{2}$.
>
> For $T_{1}$, we use the update rule of $\theta_{t}$ and isolate the contribution of clipped updates (term $S_{1}$) and that of Gaussian noise (term $S_{2}$) in equation (8), line 710. $S_{1}$ is bounded using sketching-based concentration inequalities, and $S_{2}$ via concentration bounds for Gaussian noise.
>
> For $T_{2}$, which involves the loss Hessian matrix, we leverage the special structure of its spectrum stated in Assumption 4. This allows us to express $T_{2}$ as a sum of contributions along each eigenvector direction (equation (20), line 765). Each component is controlled similarly to $T_{1}$. Combining all these pieces yields Theorem D.1.
>
> For Fed-SGM with AMSGrad, we follow a similar strategy, resulting in a more complicated but structurally similar bound, presented in Theorem E.1.
>
> We hope this summary provides better intuition for underlying analyses.
>
> ---
> **Q2: Theoretical and experimental comparisons with recent privacy–communication–utility tradeoff methods beyond DiffSketch.**
>
> We extend our evaluation to DPSFL and DPSFL-AC in [1] from October 2024 which are differentially private variants of the communication-efficient FetchSGD algorithm from [2].
>
> From a theoretical perspective, DPSFL employs a count-sketch matrix for dimension reduction, whereas our approach uses a Gaussian sketch matrix. As shown in Theorem 2 of [1], their privacy analysis does not guarantee that the required noise variance decreases as sketching dimension grows. In contrast, our privacy result (Theorem 3.1) shows that noise variance scales down proportionally to $1/\sqrt{b}$ with sketching dimension $b$. In their convergence analysis (Theorem 6), they restrict to single local step ($K=1$), while our analysis supports multiple local updates. Under specialized learning rate settings, both DPSFL’s Theorem 6 and our Remark 3.2 yield an $1/\sqrt{T}$ dependence on the number of communication rounds $T$.
>
> Experimentally, we compare DPSFL and DPSFL-AC from [1] and variants of our methods under our Figure 1 setup (ResNet101 on EMNIST). According to the table, DPSFL and DPSFL-AC outperform our GD-based variants; this gap follows the fact that FetchSGD (the non-private base of DPSFL) outperforms FedAvg (the non-private base of our method) before privacy [2, Figures 3–5]. Crucially, with Adam as global optimizer, our methods, with or without sketching, surpass DPSFL and DPSFL‑AC, demonstrating that combination of Gaussian sketching and adaptive server optimization yields superior trade‑offs among privacy, communication, and accuracy.
>
> |GD, Sketching|GD, Non-Sketching|Adam, Sketching|Adam, Non-Sketching|DPSFL|DPSFL-AC|
> |:-:|:-:|:-:|:-:|:-:|:-:|
> |$62.98\\%$|$63.34\\%$|$85.09\\%$|$78.60\\%$|$70.13\\%$|$75.64\\%$|
>
> [1] Zhang, Meifan, Zhanhong Xie, and Lihua Yin. "Private and communication-efficient federated learning based on differentially private sketches." arXiv preprint arXiv:2410.05733.
>
> [2] Rothchild, Daniel, et al. "Fetchsgd: Communication-efficient federated learning with sketching." International Conference on Machine Learning. PMLR, 2020.
>
> ---
> **Q3: Clarification of the first equality in line 646 of Lemma C.2**
>
> We apologize for the typo: in line 646, the first term on left‑hand side of the first equation should be $\log\left(\frac{\sqrt{\frac{\\|\gamma_{t}(D')\\|\_{2}^{2}}{b}+m\sigma_{g}^{2}}}{\sqrt{\frac{\\|\gamma_{t}(D)\\|\_{2}^{2}}{b}+m\sigma_{g}^{2}}}\right)^{b}$ instead of $\log\left(\frac{\sqrt{\frac{\\|\gamma_{t}(D)\\|\_{2}^{2}}{b}+m\sigma_{g}^{2}}}{\sqrt{\frac{\\|\gamma_{t}(D')\\|\_{2}^{2}}{b}+m\sigma_{g}^{2}}}\right)^{b}$. All subsequent equations are correct starting from the second equation.
>
> Now we present the derivation of the equation. Denote $\sigma_{1}^{2}=\frac{\\|\gamma_{t}(D)\\|\_{2}^{2}}{b}+m\sigma_{g}^{2}$, $\sigma_{2}^{2}=\frac{\\|\gamma_{t}(D')\\|\_{2}^{2}}{b}+m\sigma_{g}^{2}$, we need to calculate Renyi divergence between $P=\mathcal{N}(0,\sigma_{1}^{2}\mathbb{I}\_{b})$ and $Q=\mathcal{N}(0,\sigma_{2}^{2}\mathbb{I}\_{b})$. According to definition of multivariate Gaussian distribution, for $x\in\mathbb{R}^{b}$, we can write density functions
> \begin{align*}
>     P(x)&=(2\pi\sigma_{1}^{2})^{-\frac{b}{2}}\exp\left(-\frac{\\|x\\|\_{2}^{2}}{2\sigma_{1}^{2}}\right),\\\\
>     Q(x)&=(2\pi\sigma_{2}^{2})^{-\frac{b}{2}}\exp\left(-\frac{\\|x\\|\_{2}^{2}}{2\sigma_{2}^{2}}\right).
> \end{align*}
> So we have:
> \begin{align*}
>     \mathbb{E}\_{x\sim Q}\left[\left(\frac{P(x)}{Q(x)}\right)^{\alpha}\right]
>     &=\int_{\mathbb{R}^{b}}Q(x)\left(\frac{P(x)}{Q(x)}\right)^{\alpha}dx\\\\
>     &=\int_{\mathbb{R}^{b}}P(x)^{\alpha}Q(x)^{1-\alpha}dx\\\\
>     &=\int_{\mathbb{R}^{b}}\left((2\pi\sigma_{1}^{2})^{-\frac{b}{2}}\exp\left(-\frac{\\|x\\|\_{2}^{2}}{2\sigma_{1}^{2}}\right)\right)^{\alpha}\cdot\left((2\pi\sigma_{2}^{2})^{-\frac{b}{2}}\exp\left(-\frac{\\|x\\|\_{2}^{2}}{2\sigma_{2}^{2}}\right)\right)^{1-\alpha}dx\\\\
>     &=(2\pi\sigma_{1}^{2})^{-\frac{\alpha b}{2}}(2\pi\sigma_{2}^{2})^{-\frac{(1-\alpha)b}{2}}\int_{\mathbb{R}^{b}}\exp\left(-\frac{\\|x\\|\_{2}^{2}}{2}\left(\frac{\alpha}{\sigma_{1}^{2}}+\frac{1-\alpha}{\sigma_{2}^{2}}\right)\right)dx\\\\
>     &\overset{(1)}{=}(2\pi\sigma_{1}^{2})^{-\frac{\alpha b}{2}}(2\pi\sigma_{2}^{2})^{-\frac{(1-\alpha)b}{2}}\cdot\left(\frac{\pi}{\frac{1}{2}\left(\frac{\alpha}{\sigma_{1}^{2}}+\frac{1-\alpha}{\sigma_{2}^{2}}\right)}\right)^{\frac{b}{2}}\\\\
>     &=(\alpha\sigma_{2}^{2}+(1-\alpha)\sigma_{1}^{2})^{-\frac{b}{2}}\sigma_{2}^{\alpha b}\sigma_{1}^{(1-\alpha)b}
> \end{align*}
> in which $(1)$ uses that for every $a>0$, $\int_{\mathbb{R}^{b}}\exp(-a\\|x\\|\_{2}^{2})=(\frac{\pi}{a})^{\frac{b}{2}}$. Therefore, according to Definition C.1,
> \begin{align*}
>     D_{\alpha}(P\\|Q)&=\frac{1}{\alpha-1}\log\mathbb{E}\_{x\sim Q}\left[\left(\frac{P(x)}{Q(x)}\right)^{\alpha}\right]\\\\
>     &=\frac{1}{\alpha-1}\log\left((\alpha\sigma_{2}^{2}+(1-\alpha)\sigma_{1}^{2})^{-\frac{b}{2}}\sigma_{2}^{\alpha b}\sigma_{1}^{(1-\alpha)b}\right)\\\\
>     &=\frac{1}{\alpha-1}\left(-\frac{1}{2}\log(\alpha\sigma_{2}^{2}+(1-\alpha)\sigma_{1}^{2})^{b}+\alpha\log\sigma_{2}^{b}+(1-\alpha)\log\sigma_{1}^{b}\right)\\\\
>     &=\frac{1}{2(\alpha-1)}\left(\log(\sigma_{2}^{2})^{b}-\log(\alpha\sigma_{2}^{2}+(1-\alpha)\sigma_{1}^{2})^{b}\right)+(\log\sigma_{2}^{b}-\log\sigma_{1}^{b})\\\\
>     &=\log\left(\frac{\sigma_{2}}{\sigma_{1}}\right)^{b}+\frac{1}{2(\alpha-1)}\log\left(\frac{\sigma_{2}^{2}}{\alpha\sigma_{2}^{2}+(1-\alpha)\sigma_{1}^{2}}\right)^{b}\\\\
>     &=\log\left(\frac{\sigma_{2}}{\sigma_{1}}\right)^{b}+\frac{1}{2(\alpha-1)}\log\left(\frac{\frac{\sigma_{2}^{2}}{\sigma_{1}^{2}}}{\alpha\frac{\sigma_{2}^{2}}{\sigma_{1}^{2}}+1-\alpha}\right)^{b}
> \end{align*}
> Substituting $\sigma_{1}^{2}=\frac{\\|\gamma_{t}(D)\\|\_{2}^{2}}{b}+m\sigma_{g}^{2}$ and $\sigma_{2}^{2}=\frac{\\|\gamma_{t}(D')\\|\_{2}^{2}}{b}+m\sigma_{g}^{2}$, we can obtain that
> \begin{align*}
>     &D_{\alpha}(\mathcal{SG}(\gamma_{t}(D);R_{t},\xi_{t})\\|\mathcal{SG}(\gamma_{t}(D');R_{t},\xi_{t}))\\\\
>     =&D_{\alpha}\left(\mathcal{N}\left(0,\frac{\\|\gamma_{t}(D)\\|\_{2}^{2}}{b}+m\sigma_{g}^{2}\right)\Big\\|\mathcal{N}\left(0,\frac{\\|\gamma_{t}(D')\\|\_{2}^{2}}{b}+m\sigma_{g}^{2}\right)\right)\\\\
>     =&\log\left(\frac{\sqrt{\frac{\\|\gamma_{t}(D')\\|\_{2}^{2}}{b}+m\sigma_{g}^{2}}}{\sqrt{\frac{\\|\gamma_{t}(D)\\|\_{2}^{2}}{b}+m\sigma_{g}^{2}}}\right)^{b}+\frac{1}{2(\alpha-1)}\log\left(\frac{\frac{\frac{\\|\gamma_{t}(D')\\|\_{2}^{2}}{b}+m\sigma_{g}^{2}}{\frac{\\|\gamma_{t}(D)\\|\_{2}^{2}}{b}+m\sigma_{g}^{2}}}{\alpha\frac{\frac{\\|\gamma_{t}(D')\\|\_{2}^{2}}{b}+m\sigma_{g}^{2}}{\frac{\\|\gamma_{t}(D)\\|\_{2}^{2}}{b}+m\sigma_{g}^{2}}+1-\alpha}\right)^{b}.
> \end{align*}
> which is exactly the first equation in line 646. We will add this part of calculation to the original proof later.
>
> ---
> **Q4: Some equations are not appropriately numbered.**
>
> Thank you for noting this. We will conduct a thorough review of all equation numbering and correct any inconsistencies in the revised manuscript.

---

> > ### Comment · Reviewer_q7wo · 2025-08-06
> >
> > Thanks to the authors for the detailed response. I remain positive about this paper.

---

> > > ### Author Response · Authors · 2025-08-06
> > >
> > > We appreciate your positive feedback and thank you for your continued support of our paper.

---

### Official Review · Reviewer_EkNe · 2025-07-03

**Clarity:** 3
**Significance:** 3
**Originality:** 3
**Rating:** 4
**Confidence:** 4

**Summary:**

This paper studies two key challenges (communication efficiency and privacy) simultaneously in FL. The paper introduces the Sketched Gaussian Mechanism, which incorporates sketching into the gaussian mechanism to achieve both communication efficiency and privacy guarantees. The authors provide a strong connection between these components and support their approach with both theoretical analysis and empirical validation.

**Questions:**

* Why does the performance degrade in Figure (a) when gradient sketching is not applied? Is this due to the need for a larger $\sigma_g$ to satisfy the DP guarantee without sketching?

* Why both the bounded gradient assumption and gradient clipping are used in the paper/theoretical analysis? In some adaptive FL papers, such as [1] and [2], the convergence analysis relies on the bounded stochastic gradient assumption. SGD-based FL methods typically do not require this. In my opinion, if Fed-SGM already includes a gradient clipping step for DP, the additional bounded stochastic gradient assumption is naturally satisfied, so we don't need the bounded gradient one anymore. Could the authors clarify this point? Moreover, it would be relevant to include discussions with methods designed for adaptive FL, such as [1][2].﻿

[1] Reddi, S., Charles, Z., Zaheer, M., Garrett, Z., Rush, K., Konečný, J., ... & McMahan, H. B. (2020). Adaptive federated optimization

[2] Wang, Y., Lin, L., & Chen, J. (2022, June). Communication-efficient adaptive federated learning.

* Some other minor issues:

  * The update parameter $\theta_{t+1}$ does not match the final model output $\theta_T$
  * In Theorem 3.1, the summation still runs from $t=0$ to $T-1$
  * What is $z_{c,t}$ in Algorithm 2?

**Ethical Concerns:**

["NO or VERY MINOR ethics concerns only"]

**Final Justification:**

After the rebuttal, I would like to maintain my positive score of 4.

**Limitations:**

Yes, they discussed the limitations.

**Quality:**

3

**Strengths And Weaknesses:**

Strengths:
* The paper is well-structured, well-written, and easy to follow.
* DP guarantee is provided, supporting the privacy claims. The paper also includes a convergence analysis.
* I find the conclusion that “Fed-SGM usually requires a strictly lower Gaussian noise variance than the unsketched mechanism to attain the same privacy level” quite interesting, and the related analysis to support this claim is solid.

Weaknesses:
* It seems that Fed-SGM is used with Adam in the experiments, but the theoretical analysis presents results based on the global AMSGrad optimizer.
* Does Theorem 3.1 account for the stochasticity (i.e., stochastic noise) in gradient computation? It seems that the authors assume bounded stochastic noise in their assumptions, but the term $\sigma_s$ does not appear in Theorem 3.1.
* I think it is not clear enough to figure out the result of DP-FedAvg.  It seems that the SGD-no and Adam-no curves in Figures 1 and 2 correspond to DP-FedAvg, but this is not made explicit and could be confusing to readers.
* How do the authors partition the data across clients? Is the data split in an IID or non-IID manner? Given that the participation ratio is only 4 out of 625, it seems challenging to achieve good performance under heterogeneous data distributions. Moreover, the theoretical analysis does not appear to account for data heterogeneity either.

* Some ablation studies are missing:

  * As mentioned earlier, an ablation on data partitioning (e.g., IID vs. non-IID) would be helpful.
  * It would also be helpful to include ablations on the clipping threshold $\tau$ and/or the number of local steps $K$, to evaluate how these factors influence convergence behavior.

---

> ### Author Rebuttal · Authors · 2025-07-31
>
> We thank the reviewer for highlighting the strengths of the work. We appreciate that the reviewer found a key conclusion of the work 'quite interesting' and that the analysis to support the claim is solid.
>
> We first address the points mentioned under **Weaknesses** and then the points under **Questions**.
>
> **Weaknesses:**
>
> **W1: AMSGrad in theory vs. Adam in experiments**
>
> Thanks for highlighting the mismatch between the optimizer in experiments (Adam) and the one in our theory (AMSGrad). Below we address this point along two perspectives:
>
> 1. Theoretically, AMSGrad and Adam share the same update rules of first-order momentum $m_{t}$ and second-order momentum $\hat{v}\_{t}$, with AMSGrad using $v_{t}=\max(\hat{v_{t}},v_{t-1})$. As a result, our proof for AMSGrad can be modified to get a similar bound for Adam as well—we will add relevant details.
> 3. Experimentally, we have rerun our ResNet101 experiments on EMNIST using AMSGrad in addition to Adam under the same condition as Figure 1. The results demonstrate that the experiment results with AMSGrad are similar to Adam.
>
> ||Sketching|Non-Sketching|
> |:-:|:-:|:-:|
> |Adam|$85.09\\%$|$78.60\\%$|
> |AMSGrad|$85.04\\%$|$78.10\\%$|
>
> ---
> **W2: Accounting for stochastic noise $\sigma_s$ in Theorem 3.1**
>
> Yes, Theorem 3.1 does account for the stochasticity in gradient computation.
>
> The reason $\sigma_{s}$ does not appear in Theorem 3.1 is that Theorem 3.1 is an informal version (as noted in the paper), where $\sigma_{s}$ is treated as a constant and omitted in order notation. In the formal version (Theorem E.1 in appendix), $\sigma_{s}$ appears in the term $\frac{\sqrt{2}\alpha_{2}G\log(2T/\delta)\sigma_{s}}{\sqrt{NTK}\epsilon}$. We will make this clear in the final version.
>
> ---
> **W3: Does SGD-no and Adam-no curves in Figures 1 and 2 correspond to DP-FedAvg?**
>
> Apologies for lack of clarity—yes, "SGD-no" corresponds to DP-FedAvg.
>
> As we noted in captions of Figures 1 and 2, each curve label has two parts: the first part, “SGD” or “Adam”, indicates the global optimizer in Algorithm 2; and the second part is either a number, denoting the sketching dimension $b$, or “no,” indicating the non‑sketching version. Consequently, “SGD‑no” corresponds to the original DP‑FedAvg method from [1], and “Adam‑no” denotes the same method with Adam substituted for SGD. We will provide a clearer exposition in the final version.
>
> [3] Zhang, Xinwei, et al. "Understanding clipping for federated learning: Convergence and client-level differential privacy." ICML 2022.
>
> ---
> **W4: Data partition across clients (IID vs non-IID), handling heterogeneity**
>
> For experiment results in the paper, we used an IID split across all 625 clients.
>
> To assess robustness under heterogeneity, we have rerun our image‐classification experiments with a non‑IID partition (using a Dirichlet distribution with concentration parameter 0.05). The table below compares test accuracy for models with and without sketching, under both SGD and AMAGrad optimizers. As expected, overall accuracy decreases under non‑IID; however, AMSGrad still outperforms SGD, and sketching methods remains competitive with non‑sketching ones.
>
> ||GD, Sketching|GD, Non-Sketching|AMSGrad, Sketching|AMSGrad, Non-Sketching|
> |:-:|:-:|:-:|:-:|:-:|
> |IID|$62.98\\%$|$63.34\\%$|$85.04\\%$|$78.10\\%$|
> |Non-IID|$40.71\\%$|$42.31\\%$|$52.50\\%$|$51.27\\%$|
>
> For theoretical analysis, according to our data setting in line 189, we optimize over a fixed, already‐sampled dataset, i.e., finite sum or empirical loss, and make no assumption that each client’s local data share the same distribution. Consequently, our convergence bounds (for empirical loss) hold equally for both homogeneous (IID) and heterogeneous (non‑IID) partitions.
>
> ---
> **W5: Ablations on the clipping threshold $\tau$ and/or the number of local steps $K$**
>
> We have added the following groups of ablation experiments, mirroring the setup of Figure 1 (ResNet‑101 on EMNIST) and varying only the clipping threshold $\tau$ and the number of local steps $K$:
>
> 1. We compare our original setting ($\tau = 1$) with $\tau = 0.5$ and $\tau = 2$, holding all other hyperparameters constant. As shown in the table below, in our current experimental setup, test accuracy increases as $\tau$ grows, indicating that a looser clipping bound allows larger gradient norms and yields better utility under our noise regime.
>
> ||GD, Sketching|GD, Non-Sketching|AMSGrad, Sketching|AMSGrad, Non-Sketching|
> |:-:|:-:|:-:|:-:|:-:|
> |$\tau=0.5$|$62.91\\%$|$62.89\\%$|$84.97\\%$|$78.06\\%$|
> |$\tau=1$|$62.98\\%$|$63.34\\%$|$85.04\\%$|$78.10\\%$ |
> |$\tau=2$|$63.18\\%$|$63.57\\%$|$85.26\\%$|$78.23\\%$|
>
> 2. We compare our original setting ($K = 18$) with $K = 9$ and $K = 36$, keeping all other hyperparameters fixed. This table shows that, in our current experimental setup, increasing $K$ allows each client to perform more optimization steps, which leads to better test accuracy.
>
> ||GD, Sketching|GD, Non-Sketching|AMSGrad, Sketching|AMSGrad, Non-Sketching|
> |:-:|:-:|:-:|:-:|:-:|
> |$K=9$|$56.54\\%$|$56.66\\%$|$83.81\\%$|$76.21\\%$|
> |$K=18$|$62.98\\%$|$63.34\\%$|$85.04\\%$|$78.10\\%$|
> |$K=36$|$71.81\\%$|$72.18\\%$|$86.43\\%$|$80.61\\%$|
>
> ---
>
> Next we answer points raised under **Questions**.
>
> **Questions:**
>
> **Q1: Reason for performance degradation without sketching; is this due to the need for a larger $\sigma_{g}$ to satisfy the DP guarantee without sketching?**
>
> Yes, without sketching, we need larger $\sigma_g$ to satisfy DP which degrades the performance.
>
> As discussed in the first point starting from line 310 in the paper, gradient sketching reduces per‑round Gaussian noise for privacy in 2 ways:
> 1. **Reduced noise dimensionality:** Instead of adding noise in the original $d$‑dimensions, sketching projects each update to a $b$‑dimensional space with $b\ll d$, so that every round incurs only $b$‑dimensional noise. In our experiments, we set $b = 0.01~d$.
> 2. **Lowered per‑dimension noise variance:** Within an appropriate range of $b$, our privacy analysis (Theorem 3.1) far smaller variance noise suffices to meet the same privacy level.In our experiments, our FedSGM needed only $\approx 1/10$ of the noise variance used by the non‑sketching version.
>
> These two effects sharply decrease the total noise needed by Fed-SGM to reach the same privacy level, compensating for performance loss due to the projection itself. As a result, the sketching methods achieve performance that is competitive with, and in some cases superior to, the non‑sketching baseline, as shown in Figure 1 and Figure 2.
>
> ---
> **Q2: Need for both bounded gradient assumption and gradient clipping**
>
> In analyses of private FL that incorporate gradient clipping, it is standard to also impose a uniform bound on the true stochastic gradient norm [3] [4].
>
> This assumption serves a crucial purpose: by ensuring every gradient lies within a known radius $G$ _before clipping_, it guarantees that clipping introduces only bounded bias rather than uncontrolled distortion. Without this bound, one could not rule out arbitrarily large clipping errors.
>
> In our analysis, Theorem 3.1 in Section 3.2.2 decomposes convergence error into 3 terms; the term $E_{c}^{\text{AMSGrad}}$ captures the bias due to clipping. Controlling $E_{c}^{\text{AMSGrad}}$ relies directly on the bound $G$ from Assumption 2, as noted in Remark 3.3.
>
> [3] Zhang, Xinwei, et al. "Understanding clipping for federated learning: Convergence and client-level differential privacy." ICML 2022.
>
> [4] Lun Wang, et al. “D2p-fed: Differentially private federated learning with efficient communication,” arXiv:2006.13039, 2020.
>
> ---
> **Q3: Include discussions with methods designed for adaptive FL, such as [1][2].**
>
> [1] Reddi, S., et al (2020). Adaptive federated optimization.
>
> [2] Wang, Y., et al (2022). Communication-efficient adaptive federated learning.
>
> Thanks for the suggestion—we agree that deeper discussion of these works will strengthen the paper’s context. We will expand this discussion in the revised manuscript.
> * While FedAvg is effective in many settings, it inherits SGD’s limitations since FedAvg aggregates local SGD updates. As a result, several works have indeed adopted adaptive methods into federated learning, including [1][2][5][6].
> * For communication-cost, server-side adaptivity imposes no extra per-round communication on clients but accelerates convergence. Combined with sketching, we get a significant reduction in overall communication cost.
> * For privacy, the choice of server optimizer does not affect the privacy guarantee, since any optimizer to the output of SGM in each round is post‑processing, hence does not alter the privacy level.
>
> [5] Tang, H., et al. "1-bit adam: Communication efficient large-scale training with adam’s convergence speed." ICML, 2021.
>
> [6] Chen, C., et al. "Efficient-adam: Communication-efficient distributed adam." IEEE Transactions on Signal Processing, 2023.
>
> ---
> **Minor issues**
>
> **M1:** "The update parameter $\theta_{t+1}$ does not match the final model output $\theta_{T}$."
>
> Thank you for identifying the typo. We will revise the range of $t$ to run from 0 to $T-1$ to ensure proper index alignment in Algorithm 2.
>
> ---
> **M2:** "In Theorem 3.1, the summation still runs from $t=0$ to $T-1$".
>
> Our analysis yields an upper bound on the average squared gradient norm at $\theta_{t}$ throughout the learning process; the same bound holds when averaging over $t=0,...,T$. We will clarify this expression and ensure it is fully consistent with the notation in Algorithm 2.
>
> ---
> **M3:** "What is $\mathbf{z}_{c,t}$ in Algorithm 2?"
>
> Per Definition 2.1, SGM takes three inputs: the target parameter, the Gaussian projection matrix, and Gaussian noise. In Algorithm 2, $\mathbf{z}_{c,t}$ is the third input, indicating that it corresponds to the Gaussian noise. We will clarify this in Algorithm 2’s description in a future revision.

---

> > ### Comment · Reviewer_EkNe · 2025-08-05
> >
> > Thank you to the authors for the detailed response. It is clear, I have no further concerns and will maintain my positive score.

---

> > > ### Author Response · Authors · 2025-08-05
> > >
> > > We sincerely appreciate your careful review of our responses and your continued support of our paper.

---

### Public Comment · ~Moshe_Shenfeld1 · 2025-11-04
**Missing reference?**

I apologize in advance if I have misunderstood the claim, but it seems like a similar privacy guarantee directly results from the celebrated insight of [1] (which was further improved by [2]), that the JL transform itself preserves DP. In fact, this result holds even when the sketching matrix is sampled once per batch, rather than separately per example.

[1] Blocki, J., Blum, A., Datta, A., & Sheffet, O. (2012, October). The johnson-lindenstrauss transform itself preserves differential privacy. In 2012 IEEE 53rd annual symposium on foundations of computer science (pp. 410-419). IEEE.

[2] Sheffet, O. (2015). Private approximations of the 2nd-moment matrix using existing techniques in linear regression. arXiv preprint arXiv:1507.00056.

---

### Decision · Program_Chairs · 2025-09-17

**Decision:**

Accept (spotlight)

**Comment:**

This paper introduces the Sketched Gaussian Mechanism (SGM), a novel method that combines sketching and differential privacy for Federated Learning. The paper was accepted because it makes a significant and well-validated contribution to a hot area. The reviewers liked the joint privacy analysis, leading to a tighter privacy bound than prior work, and the comprehensive theoretical and empirical validation. The authors provided a great rebuttal, including new experiments and ablation studies. This led one reviewer to raise their score, leading to unanimous support for the submission.